# Widespread position-dependent transcriptional regulatory sequences in plants

Yoav Voichek [1] ✉, Gabriela Hristova[1], Almudena Mollá-Morales [1], Detlef Weigel[2] & Magnus Nordborg [1] ✉

Much of what we know about eukaryotic transcription stems from animals and yeast; however, plants evolved separately for over a billion years, leaving ample time for divergence in transcriptional regulation. Here we set out to elucidate fundamental properties of *cis*-regulatory sequences in plants. Using massively parallel reporter assays across four plant species, we demonstrate the central role of sequences downstream of the transcription start site (TSS) in transcriptional regulation. Unlike animal enhancers that are position independent, plant regulatory elements depend on their position, as altering their location relative to the TSS significantly affects transcription. We highlight the importance of the region downstream of the TSS in regulating transcription by identifying a DNA motif that is conserved across vascular plants and is sufficient to enhance gene expression in a dose-dependent manner. The identification of a large number of position-dependent enhancers points to fundamental differences in gene regulation between plants and animals.

Eukaryotes have diverged for more than a billion years[1]. Despite their immense diversity, our basic knowledge of eukaryotic transcription is mainly based on observations in yeast and a few animals. The knowledge gap is even more striking when we consider transcriptional regulation in the context of multicellularity, which requires regulatory mechanisms to enable cell type-specific gene expression. Complex multicellularity arose independently at least six times, including once in animals and once in plants[2,3]. These independent inventions require specialized mechanisms of transcriptional regulation to allow each cell type to express a different set of genes. The mechanisms that support this complexity likely evolved with the emergence of multicellularity[4]. However, we already know that plants and animals solved many of the challenges associated with multicellularity very differently, such as cell communication or adhesion. In particular, there is no reason to believe that what is true for animals is also true for plants when it comes to principles that go beyond the basic transcriptional machinery[5,6]. Notably, plants and animals exhibit differences, such as

distinct core promoter DNA motifs[7], expanded[8] and novel[9] families of specific and general transcription factors (TFs), and different features of long-range enhancers[10–12]. Yet, how regulatory sequences function is widely assumed to be similar to animals and yeast[13]. Here, we show that the basic property of the majority of animal enhancers, position independence, does not hold for plants.

## Results

### *Arabidopsis* expression quantitative trait loci are enriched downstream of the transcription start site

We first set out to determine the typical locations of regulatory regions near genes in *Arabidopsis* by large-scale mapping of variants underlying expression quantitative trait loci (eQTL). Therefore, we analyzed genotypic and rosette transcriptomic data from the Arabidopsis 1001 Genomes Project to identify *cis*-eQTL within 10 kb of each gene[14,15]. While we expected to find most eQTL in proximal promoters upstream of the transcription start site (TSS), we discovered a similar proportion

[1]Gregor Mendel Institute, Austrian Academy of Sciences, Vienna BioCenter, Vienna, Austria. [2]Department of Molecular Biology, Max Planck Institute for Biology Tübingen, Tübingen, Germany. ✉e-mail: yoav.voichek@gmi.oeaw.ac.at; magnus.nordborg@gmi.oeaw.ac.at

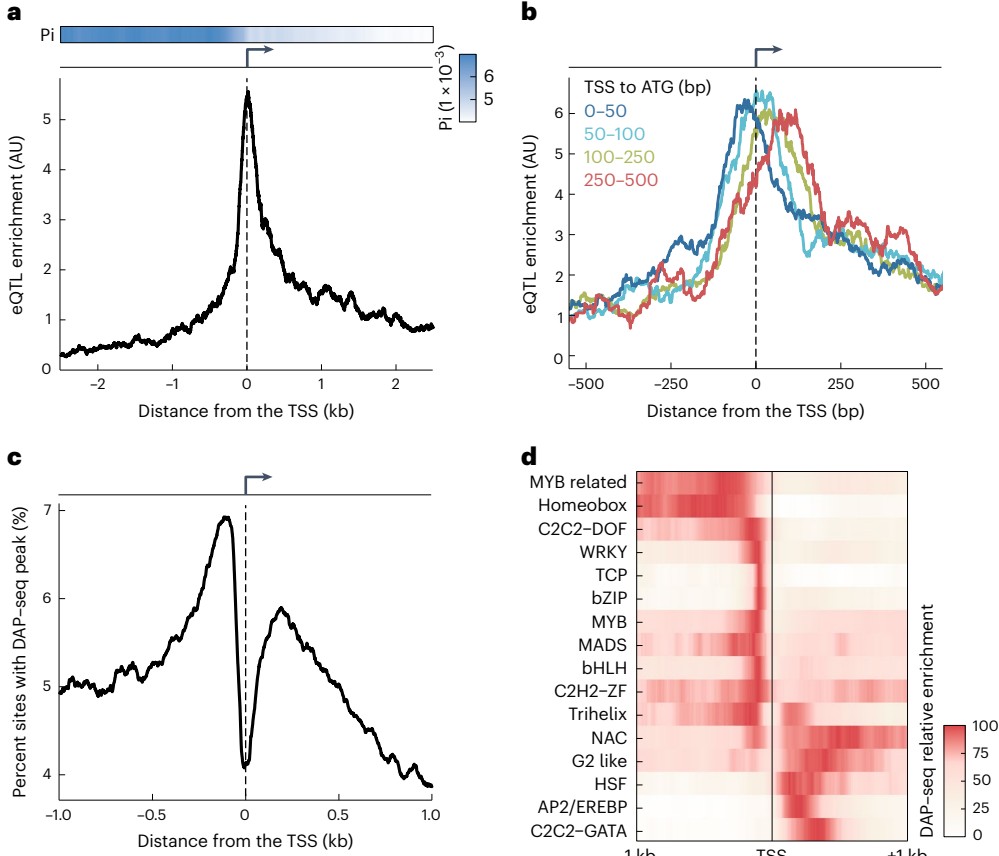

**Fig. 1 | Evidence for transcriptional regulatory sequences downstream of the TSS. a**, eQTL enrichment (below) and nucleotide diversity (Pi, above) near TSSs. **b**, eQTL enrichment for genes with different TSS-to-ATG distances. Group counts: 1,088 (0–50 bp, dark blue), 968 (50–100 bp, light blue), 1,122 (100–250 bp, green) and 577 (250–500 bp, red). **c**, Proportion of sites with a DAP–seq[19] peak center, based on data for 529 TFs. **d**, DAP–seq peak enrichment, as in **c**, consolidated for each TF family with at least ten members in the dataset[19]; the maximum signal for each TF family was scaled to 100. In **a**–**c**, data are smoothed using a 100-bp rolling window. AU, arbitrary units.

of eQTL downstream of the TSS (Fig. 1a). As eQTL are more likely to occur where the density of single nucleotide polymorphisms is higher, the lower sequence diversity downstream of TSSs made the downstream eQTL enrichment even more unexpected (Fig. 1a). This pattern was consistent across multiple gene expression datasets, and accounting for linkage between single nucleotide polymorphisms only intensified it (Supplementary Fig. 1). Our eQTL analysis pointed to a potential regulatory region of previously unappreciated importance downstream of the TSS in *Arabidopsis*.

We next examined possible explanations for the observed eQTL distribution. If causal eQTL variants within transcripts are in sequences controlling messenger RNA (mRNA) stability, these should be more frequent in exons than in introns, which are removed by splicing. While this is what is seen for human eQTL[16], we observed no such preference for exons in *Arabidopsis* (Extended Data Fig. 1a–f). eQTL were also not enriched toward the end of transcripts (Supplementary Fig. 2), even though 3′ untranslated regions (3′ UTRs) have known roles in controlling mRNA stability[17,18]. Finally, we asked whether eQTL were more likely to occur outside coding regions, which have strong sequence constraints. Indeed, eQTL tended to be most frequent just downstream of the TSS for genes with longer TSS-to-ATG distances (including 5′ UTRs and introns; Extended Data Fig. 1g) and just upstream of the TSS for genes with shorter TSS-to-ATG distances (Fig. 1b and Extended Data Fig. 1h). These findings suggest that downstream regulatory regions are enriched between the TSS and the start codon and affect transcription rather than mRNA stability.

If the location of eQTL variants in the proximity of genes results from variation in transcription rate, as opposed to mRNA stability, then chromatin and TFs are likely to be involved. The first observation in agreement with this was that histone H3.1 and H3.3 enrichment downstream of the TSS moves away from the TSS with increasing TSS-to-ATG distances (Supplementary Fig. 3). Second, binding sites of 529 TFs, as measured by DNA affinity purification and sequencing (DAP–seq)[19], have two prominent peaks, upstream as well as downstream of the TSS (Fig. 1c). Individual TFs have a preference for binding on only one side of the TSS, with similar preferences for members of the same TF family (Fig. 1d and Supplementary Fig. 4). In vivo chromatin immunoprecipitation followed by sequencing data of three TFs binding confirmed the preference of TFs to bind on either side of the TSS (Supplementary Fig. 5). We do not think that inaccuracies in the TSS annotations (Extended Data Fig. 1c) greatly affect our results, given that the clear dip in TF binding sites is centered on annotated TSSs. These analyses support the notion that many *Arabidopsis* genes have a transcriptional regulatory region downstream of the TSS.

## Massively parallel reporter assay in four species

To systematically investigate the role of sequences downstream of the TSS in controlling gene expression, we designed a massively parallel reporter assay[20] (MPRA; Fig. 2a). We synthesized 12,000 160-bp-long fragments, derived from regions 40–200 bp upstream or 40–360 bp downstream of the TSSs of highly expressed *Arabidopsis* genes, excluding 80 bp around the TSS (−40 bp to 40 bp), which contains the core

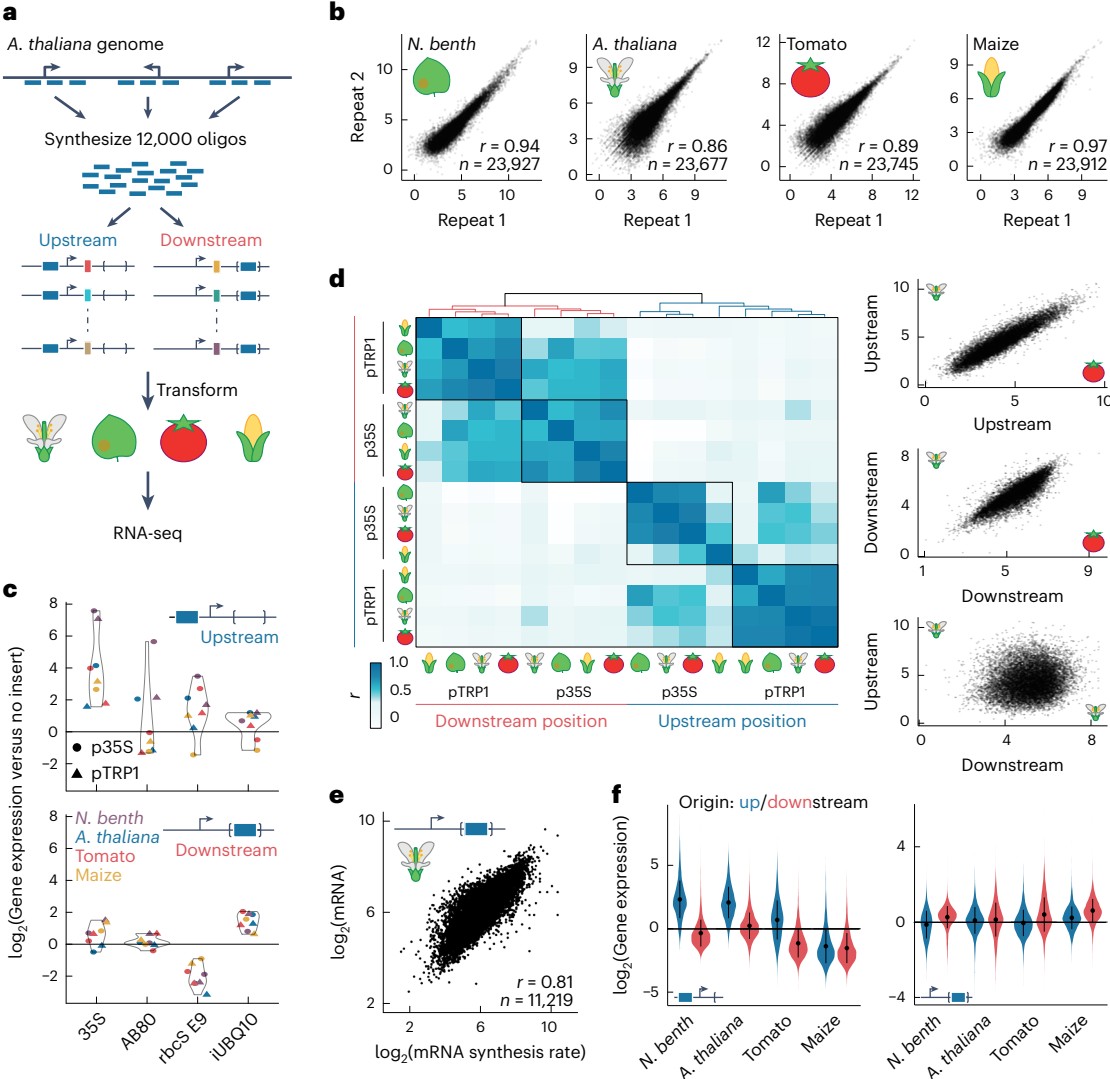

**Fig. 2 | Position-dependent enhancers reside inside transcribed regions.**
**a**, MPRA overview: 12,000 fragments (160 bp), originating from upstream
or downstream of the TSS of *Arabidopsis* genes, were synthesized, pooled
and inserted upstream of the TSS or within the intron of a reporter gene and
tagged with barcodes. Following transient transformation into one of four
species, barcoded RNA sequencing was used to quantify expression. **b**, High
reproducibility in MPRA experiment replicates, demonstrated here for *CaMV* 35S
minimal promoter-based libraries, plotted as log₂(gene expression). Pearson's
correlation coefficient *r* and number of fragments *n* are indicated. **c**, Comparison
of construct expression with control enhancer fragments and no-insert
constructs for upstream (top) and downstream (bottom) insertions. Depicted
for *N. benthamiana* (*N. benth*; purple), *A. thaliana* (blue), tomato (red) and maize
(yellow); construct backgrounds are coded by symbol shape. **d**, Left: Pearson's

correlation coefficients across all libraries, hierarchically clustered. Construct
background and insertion position are indicated. Right: activity of pTRP1-based
constructs, as log₂(gene expression), compared between upstream (top, *r* = 0.92,
*n* = 11,966) and downstream (middle, *r* = 0.85, *n* = 11,817) libraries of *Arabidopsis*
and tomato and between upstream and downstream libraries of *Arabidopsis*
(bottom, *r* = 0.13, *n* = 11,813). **e**, Comparison of mRNA steady-state levels and
synthesis rates in the downstream MPRA with pTRP1 constructs. Correlation
and fragment counts are indicated as in **b**. **f**, Comparison of activity of upstream-
derived (blue, 3,966 fragments) and downstream-derived (red, 7,928 fragments)
fragments relative to no-insertion constructs when fragments are positioned
upstream (left) or downstream (right) of the TSS. Constructs are p35S based.
Error bars depict mean ± 1 s.d.

promoter. Downstream-derived fragments included exons and introns,
except for donor and acceptor splicing sites. We inserted these frag-
ments in their original orientation on either side of the TSS of a green
fluorescent protein (GFP) reporter gene. Insertion-free constructs
served as controls. The downstream insertion site was located in an
intron of the reporter gene to rule out effects due to altered mRNA
sequence on mRNA stability. For robust quantification, multiple vari-
ants were generated for each insertion, with a 15-bp random barcode
within the transcript. Barcodes and tested regulatory fragments were
linked by DNA sequencing, and transcriptional activity was read out
by RNA sequencing.

We used two different GFP reporter constructs to provide differ-
ent promoter contexts (Extended Data Fig. 2): the 46-bp Cauliflower
Mosaic Virus (*CaMV*) 35S minimal promoter, commonly used to test
plant enhancers, in combination with a short synthetic 5' UTR, and
a 700-bp *Arabidopsis TRP1* promoter fragment including its 5' UTR,
previously used to study the effect of introns on gene expression[21].
To derive conclusions with broad applicability to flowering plants, we
quantified activity of the libraries in four different species: in *Arabi-
dopsis*, tomato and maize using transfection of leaf protoplasts and
in *Nicotiana benthamiana* using leaf infiltration of *Agrobacterium
tumefaciens* into mesophyll cells. We reasoned that the use of two

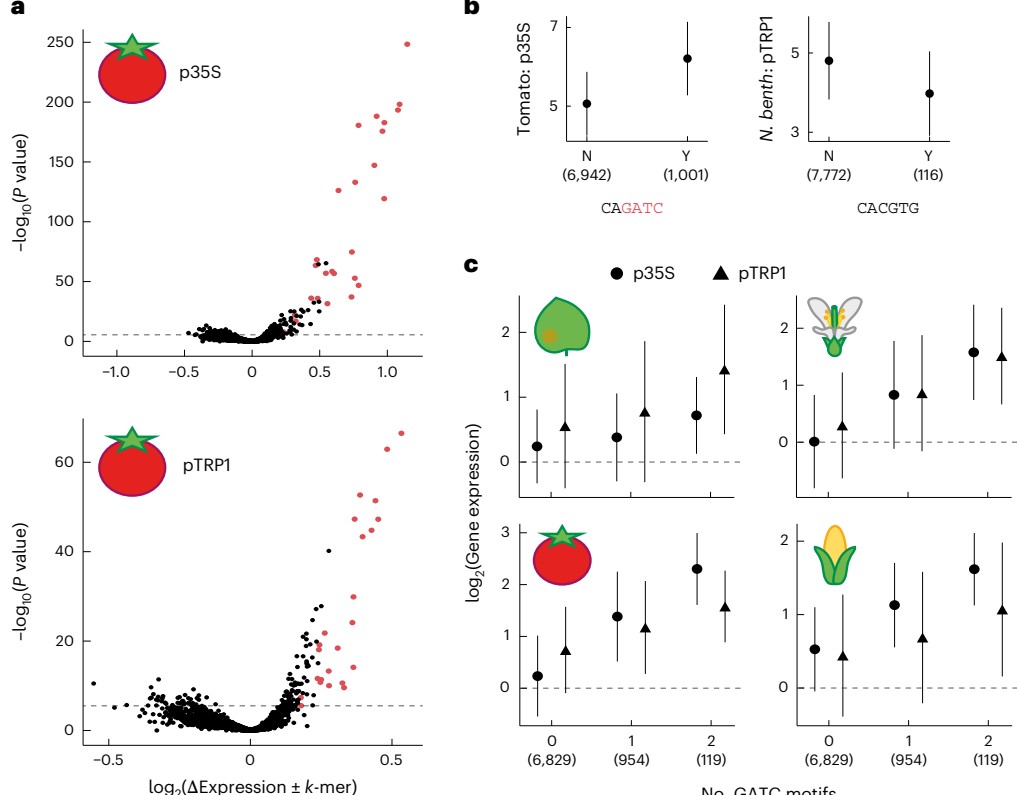

**Fig. 3 | The GATC motif is associated with high expression when positioned downstream of the TSS. a**, Association of 6-mers with downstream MPRA activity. Each 6-mer's presence (or absence) in downstream-derived fragments was compared based on the difference in average $\log_2$(gene expression) ($x$ axis) and a two-sided Mann–Whitney $U$-test $P$ value ($y$ axis). Displayed for p35-based (top) and pTRP1-based (bottom) constructs in tomato. Red points denote 6-mers with GATC; dashed line marks the 5% Bonferroni multiple-testing threshold. **b**, Activity distribution for downstream-derived fragments with (Y) or without

(N) 6-mers when inserted downstream. Plotted as $\log_2$(gene expression) in the MPRA for CAGATC in tomato with p35S-based constructs (left) and for CACGTG (G-box[54]) in *N. benthamiana* with pTRP1-based constructs (right). **c**, Relative activity of downstream fragments when inserted downstream as a function of the number of YVGATCBR consensus motifs in the tested fragments. Group sizes: 6,829 (no motif), 954 (one motif) and 119 (two motifs). Error bars in **b** and **c** represent mean ± 1 s.d. Backbone and species are indicated. The numbers in parentheses in **b** and **c** indicate the number of fragments.

different transformation methods would increase the robustness of our conclusions. Reproducibility was ensured through three to four replicated experiments (Fig. 2b and Supplementary Fig. 6).

### Position-dependent regulatory elements in plants

As a control, a small fraction of the synthesized fragments were from known enhancers, previously examined in the MPRA[20]. In most cases, these fragments increased expression when placed upstream of the TSS (Fig. 2c and Supplementary Fig. 7a) but not when placed inside introns, in agreement with previous results[20]. Segments of the *UBQ10* intron, known to enhance expression[22], were also included in the library. These intron-derived fragments drove higher expression when inserted into the intron of the reporter gene rather than when inserted upstream of the TSS (Fig. 2c).

The position-dependent effects observed for the known enhancers seem to be representative for the majority of tested fragments. We found that fragments had similar activity independent of species, promoter or how they were introduced into the host cell (Fig. 2d and Supplementary Fig. 7b,c). By contrast, the relative activity greatly changed when the same fragment was inserted either upstream or downstream of the TSS, even when using the same backbone and species. We were surprised by this lack of correlation; one possibility is that downstream insertion mainly affects mRNA stability. To test this, we modified our MPRA setup to measure mRNA synthesis directly (Extended Data Fig. 3a). We transformed *Arabidopsis* leaf protoplasts with pTRP1-based constructs containing the fragment library downstream of the TSS

and added 5-ethynyl uridine (5-EU) 20 min before RNA collection, followed by purification of 5-EU-containing mRNA[23]. Sequencing of these newly synthesized mRNA species revealed that the main effect of inserting fragments downstream is on transcription rate (Fig. 2e and Extended Data Fig. 3b). These results suggest that, unlike the position independence seen for animal enhancers, the activity of flowering plant enhancers is strongly dependent on their position relative to the TSS.

The original genomic location of the fragment played a substantial role as well. Generally, fragments increased expression when positioned in their original position relative to the TSS, but the extent varied between backbone and species (Fig. 2f and Supplementary Fig. 8). Enhancers were relatively more effective in the *CaMV* 35S promoter than in the *TRP1* promoter construct, in which fragment insertions disrupted the *TRP1* genomic sequence (Supplementary Fig. 8). In maize, fragments often reduced reporter expression when inserted upstream, regardless of genomic origin, in agreement with previous observations[24]. This finding, along with the strong correlation among the relative activity of fragments across all libraries, suggests that, while absolute levels are strongly influenced by backbone and species, the relative effects of different fragments in the same position are similar across species.

### GATC motifs enhance transcription from downstream to the TSS

An immediate question that arises from our observation is how sequences downstream of the TSS control transcription activity.

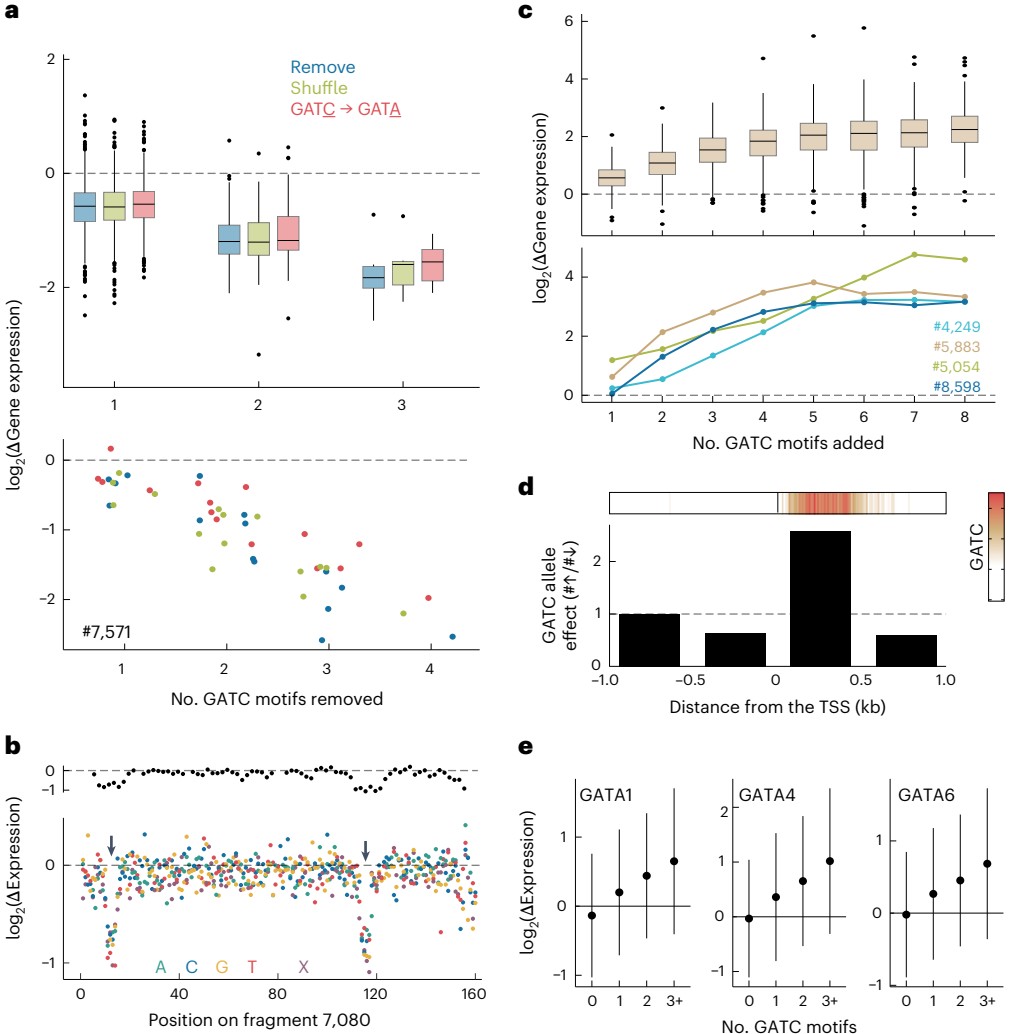

**Fig. 4 | The GATC motif is sufficient to increase expression when positioned downstream of the TSS. a–c**, Differences in activity of the p35S-downstream library between original and mutated fragments. **a**, Effect of removing GATC motifs on the activity of 823 fragments initially containing the motif (top) and a specific example (fragment 7,571) with four motifs (bottom). Motifs were removed by deletion, 8-bp shuffle or GATC-to-GATA mutation. **b**, Deep mutagenesis of fragment 7,080: effects of 10-bp deletions (top), 1-bp deletions (X) or 1-bp mutations (bottom). Arrows highlight GATC motifs. **c**, Effects of adding GATC motifs for 221 fragments with incremental motif additions (top) and four specific examples (bottom). **d**, GATC motif-gain or motif-loss alleles in the 1001 Genomes population of accessions linked to nearby gene expression.

The bar graph showcases the ratio of the number of significant associations with higher versus lower expression in the GATC motif allele, grouped by distance to the TSS (bottom). Top, GATC motif's enrichment in proximity to the TSS for all *Arabidopsis* genes. **e**, Gene expression response to overexpression of GATA TFs (individual graphs) versus GFP, plotted for genes with 0, 1, 2 or ≥3 GATC motifs within 500 bp downstream of the TSS. The groups contain 11,435, 4,039, 944 and 343 genes, respectively. Overexpression of GATA TFs or GFP was driven by double 35S promoters in *Arabidopsis* protoplasts, which were collected 8 h after transformation. Error bars represent mean ± 1 s.d. Box plots in **a** and **c** display the median (center line), the interquartile range (IQR; box bounds), whiskers (minimum and maximum within 1.5 IQR) and outliers (points beyond whiskers).

Given the results in Fig. 1d, we suspected that TFs promote transcription in this region. Although TFs often work in concert, we hypothesized that even the DNA-binding motifs of single TFs will be more abundant in strong downstream enhancers. Thus, we searched for 6-bp sequences (6-mers) for which the presence downstream of the TSS was associated with increased or decreased expression (Fig. 3a,b and Supplementary Fig. 9). We found more 6-mers that promoted expression than 6-mers that repressed expression when found downstream. These 6-mers are thus potentially part of sequence motifs bound by TFs downstream of the TSS.

Across species and backbones, 6-mers including a GATC sequence had the strongest effect (Fig. 3a and Supplementary Figs. 9 and 10). To quantify the GATC effect, we combined the six 6-mers with the strongest effect into an 8-bp YVGATCBR motif (Y = CT, V = ACG, B = CGT, R = AG; Extended Data Fig. 4), referred to as the 'GATC motif'. Transcriptional

activity increased with the number of GATC motifs in the fragment, with each copy associated with an average increase in expression of nearly 50% (Fig. 3c and Extended Data Fig. 3b). The effect was minimal with fragments inserted upstream of the TSS (Supplementary Fig. 11).

To investigate the effects of the GATC motif further, we synthesized 18,000 additional oligonucleotides, each a variant of a fragment from our initial pool as described below. These fragments were inserted downstream of the TSS in both backbones, and their effects on gene expression were measured in *Arabidopsis* protoplasts across three replicates (Extended Data Fig. 5). First, we tested the requirement of the motif by focusing on 841 downstream-derived fragments containing a GATC motif. By deleting, shuffling or modifying the core GATC to GATA, we effectively removed these motifs. We found that such removal led to an average 50% decrease in gene expression, regardless of mutation type (Fig. 4a and Extended Data Fig. 6a).

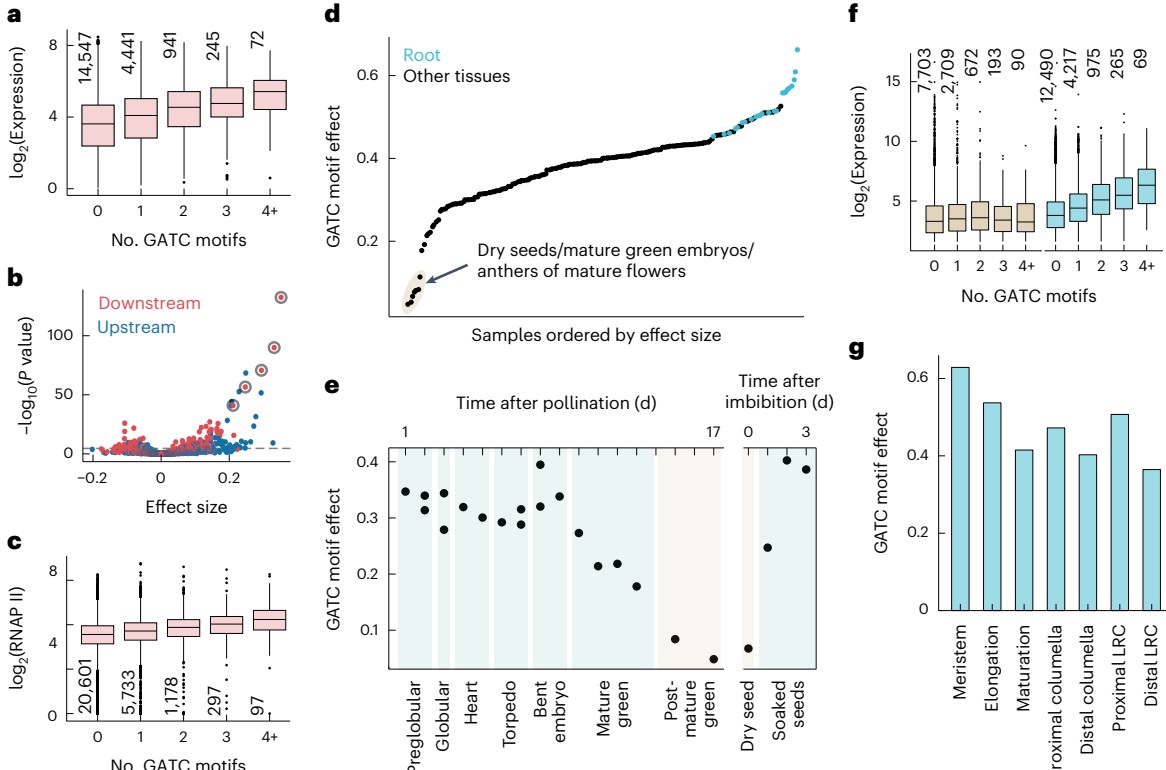

**Fig. 5 | Cell type specificity of the GATC motif effect on gene expression.**
**a**, In aerial parts of *Arabidopsis* seedlings[55], gene expression correlates with the number of GATC motifs within 500 bp downstream of the TSS. Expression values are depicted for various motif counts, with '4+' representing four to nine motifs. A linear fit reveals a GATC motif effect size of 0.4 ($P$ value = $5 \times 10^{-128}$), indicating the average expression increase for each added motif. **b**, For all 6-mers within 500 bp upstream (blue points) or downstream (red points) of the TSS, effect size and $P$ value are determined as in **a**. The five most significant downstream 6-mers, all containing the GATC sequence, are highlighted with circles. A 5% Bonferroni threshold is indicated by a dashed line. **c**, Average RNA polymerase (RNAP) II occupancy at genes plotted as in **a**, with an effect size of 0.17 ($P$ value = $10^{-107}$).

**d**, GATC motif effect sizes from a compendium of 200 tissue-specific gene expression datasets[30–32], as determined in **a**. Samples with the lowest effect sizes are shaded and detailed. **e**, Chart of GATC effect size during embryo and seed development and upon imbibition[31,32,56]. **f**, Expression values in dry seeds (brown) and seedling roots[31] (blue), plotted as in **a**, with effect sizes of 0.07 ($P$ value = $2.6 \times 10^{-3}$) and 0.57 ($P$ value = $4 \times 10^{-400}$), respectively. **g**, GATC effect sizes across different root developmental stages, averaged from single-cell expression data[33]. LRC, lateral root cap. Box plots in **a**, **c** and **f** display the median (center line), the IQR (box bounds), whiskers (minimum and maximum within 1.5 IQR) and outliers (points beyond whiskers); the number of genes per category is also indicated. $P$ values in **a**–**c** and **f** were calculated using a two-sided $t$-statistic.

To supplement the GATC-focused mutation analysis, we conducted a deep mutational scan of 13 downstream-derived fragments. For each, we (1) deleted every set of ten consecutive base pairs and (2) either mutated each nucleotide to its three alternatives or deleted it. This resulted in 736 derivatives from each original fragment. Any change to the core 4 bp of the GATC motif decreased activity, underscoring the motif's strict constraints (Fig. 4b and Supplementary Figs. 12 and 13). As expected, these analyses also revealed additional sequences that do not include GATC motifs as important for enhancing the activity of the tested fragments (Supplementary Fig. 14).

We next explored the sufficiency of GATC motifs for enhancing gene expression. We started with a random set of 221 fragments from our initial set (166 downstream-derived and 55 upstream-derived fragments) and incrementally added one to eight GATC motifs to the fragments. Expression consistently increased with each added copy, even for upstream-derived fragments (Fig. 4c and Extended Data Fig. 6b,c). Remarkably, 97% of these fragments enhanced expression as soon as at least four GATC motifs were added. The enhancement was a function of the basal activity of each fragment, with the increased activity of highly active fragments becoming saturated after a single addition and the activity of the initially least active fragments remaining unsaturated even after adding eight GATC motifs (Extended Data Fig. 6d). This finding suggests that the GATC motif and other activity-enhancing sequences

may act by the same mechanism to increase expression of the reporter constructs.

Finally, to confirm the inferences from our synthetic MPRA mutational analysis of the GATC motif, we explored the effects of natural variation in the GATC motif by returning to the Arabidopsis 1001 Genomes Project data[14,15]. We identified gains and losses of GATC motifs near TSSs and asked how these correlated with expression of the affected genes. We categorized significant associations based on whether the allele with the GATC motif had higher or lower expression. Consistent with our MPRA findings, an enrichment of higher expression was observed exclusively in the GATC motif allele situated downstream of the TSS, particularly within the initial 500 bp (Fig. 4d and Supplementary Fig. 15). Intriguingly, this is also where the GATC motif is predominantly found (Fig. 4d), reinforcing its role in enhancing gene expression when located downstream of the TSS.

What might be the mechanisms underlying the observed effects of GATC motifs? In plants, the GATC motif is recognized by GATA TFs[19] (Supplementary Note 1 and Supplementary Fig. 16), which are linked to diverse biological functions[25]. The *Arabidopsis* genome encodes 30 of these TFs. Available DAP–seq data[19] reveal GATA factor-binding enrichment within 500 bp downstream of the TSS (Fig. 1d). In this region, regardless of genomic context, 7,397 genes have at least one GATC motif (Supplementary Fig. 17a,b). Transient overexpression of three different GATA TFs in *Arabidopsis* leaf protoplasts followed by

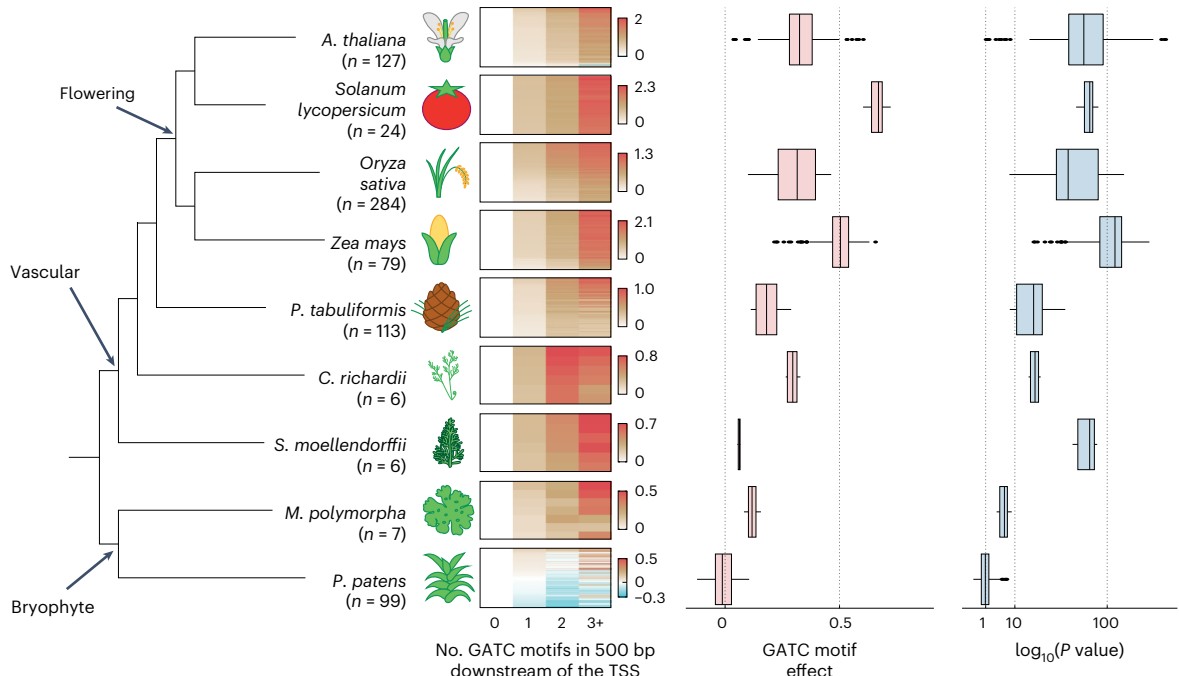

**Fig. 6 | Downstream GATC motif correlates with gene expression in vascular plants.** Heatmap of average log₂(expression change) of multiple transcriptome datasets for genes, categorized by 0, 1, 2 or ≥3 GATC motifs within 500 bp downstream of the TSS, across land plant species. Number of datasets (*n*) is indicated. Expression is normalized to the 0-motif group, with species-specific color scales. The effect on expression (slope) and the significance of association (two-sided *t*-statistic *P* value) of the GATC motif, as in Fig. 5, are presented. Box plots show the median (center line), the IQR (box bounds), whiskers (minimum and maximum within 1.5 IQR) and outliers (points beyond whiskers). The right *x* axis is square-root scaled.

RNA sequencing confirmed that these genes are direct targets of GATA TFs (Fig. 4e, Extended Data Fig. 7 and Supplementary Fig. 18). Gene ontology analysis showed these genes to be enriched in processes related to the Golgi apparatus, the endoplasmic reticulum, endosomes and vesicle-mediated transport (Supplementary Table 7). Given its prevalence, association with the secretion system and the evidence for conservation between species (Supplementary Fig. 19), the GATC motif likely acts as a widespread and conserved regulatory signal in diverse biological functions.

Enhancer sequences typically consist of multiple DNA motifs that are targeted by specific TFs, of which *Arabidopsis* has more than 1,500 (ref. 26). Given this diversity, individual regulatory motifs have generally limited power to predict absolute levels of gene expression. We found nevertheless a strong positive relationship between the occurrence of the GATC motif within 500 bp downstream of the TSS and gene expression (Fig. 5a). This relationship was driven by motifs in all genomic contexts, with motifs in introns and UTRs showing a stronger association (Supplementary Fig. 20). Analysis of all 6-mer counts, both downstream and upstream of the TSS, showed that the GATC motif has the strongest association with gene expression (Fig. 5b). This identifies the downstream GATC motif as an especially potent regulatory sequence.

As the effects of the GATC motif are strong enough to be observed in genome-wide gene expression measurements, we can investigate its function using the many other resources available for *Arabidopsis*. As one example, if the GATC motif indeed works primarily through transcription and not mRNA stability, we expect it to affect chromatin measurements. Indeed, the occurrence of GATC motifs is correlated with the active marks histone 3 lysine 4 trimethylation (H3K4me3) and histone 3 lysine 36 trimethylation (H3K36me3)[27] as well as RNA polymerase II occupancy[28] (Fig. 5c and Extended Data Fig. 8a,b). Moreover, we observed a correlation with genome-wide measurements of mRNA synthesis but not mRNA half-life[29] (Extended Data Fig. 8c–f). These results further support an effect through transcription, in accordance with our mRNA synthesis measurements in the MPRA (Extended Data Fig. 3b).

### GATC motifs tune transcription across tissues

Our MPRA inferences came only from enhancer activity in leaf cells; therefore, we were curious whether the GATC motif was also effective in other tissues. Analyzing a compendium of gene expression in different tissues and developmental stages verified once more the potent activity of the GATC motif in increasing expression yet also revealed a roughly threefold fluctuation in the impact of the GATC motif[30–32] (Fig. 5d). Its influence was smallest in specific seed developmental stages: decreasing from mature green embryo stages through seed drying and then rebounding upon germination[32] (Fig. 5e,f).

Conversely, the strongest effects were seen in roots (Fig. 5d). Single-cell expression data from *Arabidopsis* roots[33] pinpointed the meristem as the region most associated with the GATC motif, with decreasing effects through the elongation and maturation zones (Fig. 5g). This trend held true across various root cell types (Extended Data Fig. 9). Similarly, in the vegetative shoot apex[34], the GATC motif's impact diverged between cell types: for example, mesophyll cells showing muted effects compared to the pronounced effects in epidermal cells (Supplementary Fig. 21). Overall, the GATC motif's regulatory role spans the entire body plan of the plant, being modified by tissue and cell type. In addition, the expression of GATA TFs, especially from subfamily A[25,35], correlates with the effect of the GATC motif (Extended Data Fig. 10). This suggests that the GATC motif functions like a general rheostat, modulating gene expression of thousands of genes across plant cell types, likely through GATA TFs.

### The GATC motif effect is conserved in vascular plants

To evaluate the conservation of the GATC motif's influence on gene expression, we correlated the number of GATC motifs in the 500-bp downstream region with gene expression across various

land plants[31,36–43]. Consistent with our MPRA findings in four flowering plants, GATC motif count correlated with gene expression in all flowering plants examined (Fig. 6 and Supplementary Fig. 22). This conservation extended to the gymnosperm *Pinus tabuliformis* and the fern *Ceratopteris richardii*, albeit with lower effect size than in flowering plants. In the lycophyte *Selaginella moellendorffii*, the association of the GATC motif with gene expression was markedly weaker, although still significant. Among bryophytes, there was a modest effect in *Marchantia polymorpha*, with weak statistical support, and there was no clear effect in *Physcomitrium patens*. Overall, the impact of the GATC motif and, by extension, of downstream regulatory sequences is conserved in vascular plants, with a weaker influence outside flowering plants.

In summary, we have identified the 500-bp region downstream of the TSS as a prominent site for transcription regulation for a large fraction of plant genes. We demonstrate that the function of regulatory sequences near the TSS is dependent on their position relative to the TSS, making them distinct from animal enhancers. We further examined a specific downstream GATC motif that modulates transcription in a dose-dependent manner through GATA TFs. In our analysis, the effect size of the GATC motif surpassed that of any other short DNA motif, even those located upstream of the TSS. The motif apparently acts as a regulatory module, operating much like a rheostat in tuning gene expression between cell types throughout vascular plants.

## Discussion

Our findings are consistent with previous observations of differences in transcriptional regulation between plants and animals[7–12]. Specifically, plant introns in close proximity to the TSS have been frequently identified as drivers of gene expression[44–46]. In particular, research into the role of introns in controlling gene expression has highlighted a motif similar to the GATC motif that was also conserved in natural populations of *Arabidopsis thaliana*[21,47,48].

Our observations on the dependency of enhancer position relative to the TSS are consistent with several previous reports based on individual genes: intron-derived regulatory sequences became inactive when moved upstream of the TSS[21] and strong upstream enhancers lost activity when moved into the transcribed region[20]. More generally, our results show that regulatory sequences function differently on either side of the TSS in plants, rather than exclusively on one side, as indicated by the lack of, rather than negative, correlation between the effects of the same fragment on either side (Fig. 2d). This may explain why testing enhancers by positioning them in the 3′ UTR of plants results in a strong enrichment of regions from transcribed regions[49,50]. Although this contradicts the common view of the role of the upstream region in controlling expression, the different ways in which enhancers are 'read' on either side of the TSS may account for these contrasting results.

One might expect that intragenic enhancers impede RNA polymerase II due to recruitment of DNA-binding TFs to the transcribed region. While the presence of nucleosomes at genes and intronic enhancers in animals indicate that RNA polymerase can navigate proteins obstructing its path[51], it remains unclear how enhancers might function differently depending on their positioning relative to the TSS. We propose that the distinct three-dimensional genome architecture in plants, characterized by densely packed genes compared to what has been shown in animals[52,53], might create different local environments on either side of the TSS, but many other scenarios can be imagined as well.

Finally, the GATC motif regulatory program exerts a widespread influence, modulating the gene expression of a substantial proportion of genes throughout the plant body. The adaptive advantages this mechanism offers and how it has evolved across different lineages promise to be a fertile ground for future exploration.

## Online content

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

## Methods

Additional methods section can be found in Supplementary Information.

### Construction of MPRA backbone plasmids

Plasmids pPSup_iGFP (149420) and pPSint (149421) were obtained from Addgene and were modified to generate six new plasmids. Initially, BbsI sites were replaced with BsmBI sites, and a 1-bp mutation was introduced to eliminate an extra BsmBI site in both plasmids. A *ccdB* lethal cassette was inserted between the two BsaI sites. These alterations produced pPSup_iGFP_v2 and pPSint_v2. Next, and pPSint_v2_rmBsaI was created by removing the BsaI insertion sites with the *ccdB* cassette from pPSint_v2. Subsequently, a 633-bp genomic region was cloned into both pPSup_iGFP_v2 and pPSint_v2. This region was lifted from the *TRP1* gene's promoter, extending up to 40 bp upstream of the main TSS. The pTRP1 was positioned upstream of the BsaI site in pPSup_iGFP_v2 and upstream of the BsmBI site in pPSint_v2, generating pPSup_iGFP_v2_pTRP1 and pPSint_v2_pTRP1, respectively. Lastly, the BsaI site and the *ccdB* cassette were removed from pPSint_v2_pTRP1 to create pPSint_v2_pTRP1_rmBsaI.

### Amplification of oligonucleotide pools

Two oligonucleotide pools were obtained from Twist Bioscience, consisting of 12,000 and 17,996 oligonucleotides 200 bp in length, referred to as OP1 and OP2, respectively. The design of these pools is detailed in the 'Design of oligonucleotide pools' section in Supplementary Information. Two separate amplification reactions were carried out for (1) OP1 and (2) an equimolar mix (in the level of single fragments) of OP1 and OP2 (2:3 ratio of full libraries) as follows: 20 reactions were performed, each containing 1 µl of OP1 or the OP1 + OP2 mix (1 ng µl$^{-1}$), 5 µl primer P1 (10 µM; primers are listed in Supplementary Table 1), 5 µl primer P2 (10 µM), 25 µl KAPA HiFi HotStart ReadyMix (KK2601) and 14 µl double-distilled water (DDW). The reactions were run using the following thermal cycling protocol: 95 °C, 3 min; (98 °C, 20 s; 60 °C, 30 s; 72 °C, 20 s) × 15; 72 °C, 1 min. The amplified oligonucleotide pool was then purified using 1.4× AMPure XP (A63880) beads, and amplification was verified with an Agilent Fragment Analyzer.

### Cloning oligonucleotide pools and barcodes into plasmid backbones

Cloning of OP1 into pPSup_iGFP_v2, pPSint_v2, pPSup_iGFP_v2_pTRP1 or pPSint_v2_pTRP1 as well as the OP1 + OP2 mixture into pPSint_v2 or pPSint_v2_pTRP1 was performed using 20 Golden Gate reactions. Each reaction contained 1 µl backbone plasmid (75 ng µl$^{-1}$), 1 µl of the amplified oligonucleotide pool (5 ng µl$^{-1}$), 2.5 µl T4 DNA ligase (M0202T), 1.5 µl BsaI-HFv2 (R3733S) and 18.5 µl DDW. A control reaction was also carried out, with DDW replacing the oligonucleotide pool. The reactions were incubated in a thermal cycler using the following protocol: (37 °C, 5 min; 16 °C, 5 min) × 30; 60 °C, 5 min. The reactions were then cleaned with 1.5× AMPure XP beads. The resulting reactions were transformed into MegaX DH10B T1R Electrocomp Cells (C640003, *Escherichia coli*), according to ref. [57]. A dilution series was plated on LB–spectinomycin medium for both reaction and control to estimate cloning complexity. The rest of the transformation was grown overnight and purified with the Qiagen Plasmid Plus Midi Kit (12943).

A second set of Golden Gate reactions was performed with the libraries containing the cloned oligonucleotide pools to add a minimal promoter, a 5′ UTR and a 15-bp barcode (VNN × 5). Two different inserts were used. The first insert contained a *CaMV* 35S minimal promoter and the SynJ synthetic 5′ UTR, as described[20,58], and was amplified using P3–P5 primers, diluted to 3.13 ng µl$^{-1}$ and inserted into pPSup_iGFP_v2, pPSint_v2 or pPSint_v2_rmBsaI. A second insert contained the minimal promoter and the 5′ UTR of the *TRP1* gene and was amplified using P6–P8 primers, diluted to 5 ng µl$^{-1}$ and inserted into pPSup_iGFP_v2_pTRP1, pPSint_v2_pTRP1 or pPSint_v2_pTRP1_rmBsaI. Fifteen Golden Gate

reactions and one control reaction were carried out for pPSup_iGFP_v2, pPSint_v2, pPSup_iGFP_v2_pTRP1 and pPSint_v2_pTRP1 as described above, substituting the restriction enzyme with BsmBI-v2 (R0739L) and adding the corresponding insert to each reaction, with the thermocycler protocol (42 °C, 5 min; 16 °C, 5 min) × 30; 60 °C, 5 min. Transformation into *E. coli* and efficiency calculation followed the same procedure. For pPSint_v2_rmBsaI and pPSint_v2_pTRP1_rmBsaI, two modifications were made: only ten Golden Gate reactions and one control were performed, and in-house-prepared *E. coli* competent cells were used.

This procedure resulted in eight plasmid libraries: (1) pPSup_iGFP_v2 (p35S.SynJ-OP1), (2) pPSint_v2 (p35S.SynJ-OP1), (3) pPSint_v2_rmBsaI (p35S.SynJ), (4) pPSup_iGFP_v2_pTRP1 (pTRP1-OP1), (5) pPSint_v2_pTRP1 (pTRP1-OP1), (6) pPSint_v2_pTRP1_rmBsaI (pTRP1), (7) pPSint_v2 (p35S.SynJ-OP1 + OP2) and (8) pPSint_v2_pTRP1 (pTRP1-OP1 + OP2). Extended Data Fig. 2 depicts the structure of the libraries.

### Mixing of input MPRA libraries

Three mixes of libraries were made. The labeled MIX1 contained six libraries: pPSup_iGFP_v2 (p35S.SynJ-OP1), pPSint_v2 (p35S.SynJ-OP1), pPSup_iGFP_v2_pTRP1 (pTRP1-OP1), pPSint_v2_pTRP1 (pTRP1-OP1), pPSint_v2_pTRP1_rmBsaI (pTRP1) and pPSint_v2_rmBsaI (p35S.SynJ) in 50:50:50:50:1:1 proportions, respectively. The second was labeled MIX2, with four libraries: pPSint_v2 (p35S.SynJ-OP1 + OP2), pPSint_v2_pTRP1 (pTRP1-OP1 + OP2), pPSint_v2_rmBsaI (p35S.SynJ) and pPSint_v2_pTRP1_rmBsaI (pTRP1) in 100:100:1:1 proportions, respectively. The third, labeled MIX3, contained two libraries: pPSint_v2_pTRP1 (pTRP1-OP1) and pPSint_v2_pTRP1_rmBsaI (pTRP1) in 100:1 proportions, respectively. These three mixes were transformed into MegaX DH10B T1R Electrocomp Cells with an efficiency of >10$^8$ and then purified with the Qiagen Plasmid Plus Giga Kit (12191).

### Sequencing of input MPRA libraries

To connect barcodes to tested fragments and libraries, the relevant region from the cloned libraries was sequenced with next-generation sequencing. To avoid the same initial bases in reads 1 and 2 in the Illumina run, due to amplification using the constant sequence around the barcode and the enhancer, which is detrimental to the imaging analysis of sequence signal, primers were chosen to create variation in read start. Every forward or reverse primer used was a combination of four primers that each had a different, 0–3 nucleotides of shift before the constant sequence. Primers P9–P12 and P13–P16 were used for pPSup_iGFP_v2 (p35S.SynJ-OP1), P17–P20 and P21–24 were used for pPSint_v2 (p35S.SynJ-OP1) and pPSint_v2 (p35S.SynJ-OP1 + OP2), P17–P20 and P25–P28 were used for pPSint_v2_rmBsaI (p35S.SynJ), P13–P16 and P29–P32 were used for pPSup_iGFP_v2_pTRP1 (pTRP1-OP1), P33–P36 and P21–P24 were used for pPSint_v2_pTRP1 (pTRP1-OP1) and pPSint_v2_pTRP1 (pTRP1-OP1 + OP2), and P33–P36 and P25–P28 were used for pPSint_v2_pTRP1_rmBsaI (pTRP1). To estimate exact mixes' ratios, additional shotgun Tn*5*-based DNA-seq libraries, using an in-house Tn*5* enzyme, were prepared from each of the library mixes and sequenced in 50-bp or 100-bp paired-end mode on an Illumina NovaSeq 6000 machine.

### Plant material and growth conditions

*A. thaliana* Col-0 seeds were sterilized with 70% (vol/vol) ethanol and 6.5% (vol/vol) bleach and sown on circular plates containing 0.5× MS, 0.05% (wt/vol) MES and 0.8% (wt/vol) agar. The plants were grown under long-day conditions (21 °C, 85 µmol m$^{-2}$ s$^{-1}$) for 23–26 d. Tomato (*Solanum lycopersicum*) cv. M82 (sp$^-$/sp$^-$) seeds were sterilized with 70% ethanol and 3.25% bleach and sown in Magenta boxes (6 cm × 6 cm × 9.5 cm) containing 62.5 ml 0.217% (wt/vol) Nitsch medium (Duchefa Biochemie, N0224.0050), 2% (wt/vol) sucrose and 0.9% agar. Tomato plants were grown under long-day conditions (21 °C, 85 µmol m$^{-2}$ s$^{-1}$) for 20 d. Maize (*Zea mays*) cv. B73 seeds were grown in soil (4:1 Klasmann Substrate 2:perlite) in the greenhouse (long-day

photoperiod, 23 °C, 150 µmol m$^{-2}$ s$^{-1}$) until 1–2-cm shoots were visible. The pots were covered and grown in the dark for 7–9 d. *N. benthamiana* cv. LAB seeds were sown on soil (4:1 Gramoflor 2006:perlite), stratified for 4 d in the dark (4 °C) and transferred to short-day conditions (21 °C, 60% humidity) and grown for 23 d.

## MPRA assay in tomato and *Arabidopsis* protoplasts

The protocol was adapted from ref. 59. Briefly, five to six leaves from 30–35 *Arabidopsis* seedlings or the true leaves from 20–25 tomato seedlings were cut into strips 0.5–1 mm wide using a razor blade and immediately submerged in 15 ml enzyme solution (0.4 M mannitol, 20 mM MES, pH 5.7, 20 mM KCl, 1.5% (wt/vol) Cellulase R-10 (Duchefa Biochemie, C8001.0010), 0.4% (wt/vol) Macerozyme R-10 (Duchefa Biochemie, M8002.0005), 10 mM CaCl$_2$, 0.1% (wt/vol) BSA). The enzyme solution was incubated overnight at 25 °C in the dark with gentle agitation (25 rpm). The solution was strained through a 100-µm filter and centrifuged at 100*g* for 10 min at room temperature. This and all other centrifugation steps on protoplasts were performed with a soft start and end. The pellet was resuspended in 3 ml W5 solution (154 mM NaCl, 125 mM CaCl$_2$, 5 mM KCl, 2 mM MES, pH 5.7), and healthy protoplasts were isolated using a sucrose gradient (23% (wt/vol)) by centrifugation at 450*g* for 3 min at room temperature. The protoplast fraction was resuspended in 14 ml W5 solution and counted on a hemocytometer. The suspension was centrifuged at 100*g* for 10 min at room temperature, and the pellet was resuspended in MMG solution (0.4 M mannitol, 15 mM MgCl$_2$, 4 mM MES, pH 5.7) to a concentration of 1 million cells per ml.

From the suspension, two separate samples of 200 µl were taken, each containing 200,000 protoplasts. These were placed into two distinct 1.5-ml tubes. To the first tube, 10 µg of a plasmid, which codes for the Clover protein with a nuclear localization signal sequence expressed with the pUBI promoter, was added as a positive control. To the second tube, 10 µl elution buffer was added, serving as the negative control. The rest of the suspension was split into 50-ml Falcon tubes with 4–6 ml suspension each and mixed with 50 µg plasmid library per million cells. A volume of PEG solution (0.2 M mannitol, 0.1 M CaCl$_2$, 40% (wt/vol) polyethylene glycol, MW 4000) equal to that of the protoplast–DNA suspension was added to each tube, and the protoplasts were incubated for 20 min in the dark at room temperature. W5 solution (0.95 ml) was added to the controls, and 4.75 ml per million cells was added to the samples. After 15 min of incubation at room temperature, the protoplasts were centrifuged at 450*g* for 5 min at room temperature, and the pellets were resuspended in 1 ml W1 solution (0.5 M mannitol, 20 mM KCl, 4 mM MES, pH 5.7) for the controls and 5 ml per million cells for the samples. Each protoplast suspension was transferred to a separate sterile Petri dish and incubated at 25 °C under constant light (85 µmol m$^{-2}$ s$^{-1}$) for 6 h (samples) or overnight (controls). After 6 h, the samples were centrifuged at 450*g* for 5 min at room temperature, and the pellets were flash frozen in liquid nitrogen. The samples were stored at −70 °C until RNA extraction. The controls were imaged under a microscope to check the transformation efficiency: typically around 70–80% in the positive control were successfully transformed, for both *Arabidopsis* and tomato. Testing the efficiency of transformation on a large scale, as performed for the libraries, gave 60% efficiency. This experiment was carried out with four replicates for *Arabidopsis*, yielding 10, 14, 10 and 14 million protoplasts and with 3 replicates for tomato, yielding 9, 24 and 16 million protoplasts.

## MPRA assay in maize protoplasts

The protocol was adapted from ref. 60. The middle parts (6–8 cm) of the second leaf from 30 maize plants were used. Each leaf was cut in half, and both halves were placed on top of each other, followed by cutting into 0.5–1-mm-wide strips perpendicular to the veins using a razor blade. Strips were immediately submerged in 60 ml enzyme solution (0.6 M mannitol, 10 mM MES, pH 5.7, 1.5% (wt/vol) Cellulase R-10

(Duchefa Biochemie, C8001.0010), 0.3% (wt/vol) Macerozyme R-10 (Duchefa Biochemie, M8002.0005), 1 mM CaCl$_2$, 0.1% (wt/vol) BSA, 5 mM β-mercaptoethanol) split into four Petri dishes with 15 ml solution each. The enzyme solutions were covered and vacuum infiltrated for 1 h at room temperature and then incubated for 2 h in the dark at room temperature with gentle agitation (40 rpm). The protoplasts were released by shaking at 80 rpm for 10 min. The solutions were strained through a 100-µm filter and combined in two tubes with 30 ml solution each. The tubes were centrifuged (all centrifugations with protoplasts were carried out with a soft start and end) at 70*g* for 3 min at room temperature, and the pellets were resuspended in 10 ml 0.6 M mannitol. The two protoplast suspensions were combined in one tube and counted on a hemocytometer. The suspension was centrifuged at 70*g* for 3 min at room temperature, and the pellet was resuspended in MMG solution (0.6 M mannitol, 15 mM MgCl$_2$, 4 mM MES, pH 5.7) to a concentration of 1 million cells per ml.

The protoplast suspension was split into several 50-ml Falcon tubes with 4–6 ml suspension each and mixed with 200 µg plasmid library per million cells. A volume of PEG solution (0.6 M mannitol, 0.1 M CaCl$_2$, 40% (wt/vol) polyethylene glycol, MW 4000) equal to that of the protoplast–DNA suspension was added to each tube, and the protoplasts were incubated for 15 min in the dark at room temperature. W5 solution (5 ml per million cells) (the same as for *Arabidopsis* and tomato protoplasts) was added. The samples were centrifuged at 70*g* for 3 min at room temperature, and the pellets were resuspended in 5 ml incubation solution (0.6 M mannitol, 4 mM KCl, 4 mM MES, pH 5.7) per million cells. Each suspension was transferred to a separate sterile Petri dish and incubated at 25 °C in the dark for 12 h. After 12 h, a 1-ml aliquot was saved for imaging, and the rest of the samples were centrifuged at 70*g* for 3 min at room temperature. The pellets were flash frozen in liquid nitrogen and stored at −70 °C until RNA extraction. The 1-ml aliquot was imaged under a microscope to check the transformation efficiency: 77% and 93% of protoplasts were successfully transformed in the first and second replicates, respectively, while the third replicate was not quantified. The protoplast counts for the experiments were 21, 18 and 20 million.

## MPRA assay in *N. benthamiana*

MIX1 libraries were transformed into *A. tumefaciens* (ACC-110, GV3101 (pSoup), Lifeasible). In a cold room, 25 µl ACC-110 competent cells was mixed with 1 µl plasmid, transferred to a prechilled 0.1-cm cuvette (Bio-Rad) and electroporated (1,800 V, 25 µF, 200 Ω). Immediately after, 1 ml of prewarmed (30 °C) LB medium was added, and the mixture was incubated at 30 °C for 160 min at 200 rpm. *Agrobacterium* was then plated in a dilution series on LB plates containing spectinomycin, gentamicin and rifampicin to estimate transformation efficiency. The remaining *Agrobacterium* was grown overnight in 50 ml LB with the same antibiotics. The next day, *Agrobacterium* was centrifuged for 5 min at 3,000*g*, resuspended to an OD$_{600}$ of 0.5 in infiltration medium (50 mM NaH$_2$PO$_4$, 2 mM MES, 0.5% (wt/vol) glucose, 150 µM acetosyringone, pH 5.6–5.7) and incubated for 1–2 h in the dark at room temperature with gentle agitation. *Agrobacterium* was then infiltrated into 17–20 plants (two leaves per plant, using a needleless syringe). Plants were not watered for 2 d before infiltration and were watered immediately after. To increase humidity, plants were covered with a transparent cover for 24 h, and infiltrated leaves were harvested in liquid nitrogen 48 h after infiltration. This experiment was carried out with four replicates.

## RNA extraction from protoplasts

Total RNA from *Arabidopsis*, tomato and maize protoplasts was extracted using the Monarch Total RNA Miniprep Kit (New England Biolabs, T2010S). The frozen pellets were thawed shortly on ice, and then the protocol for 'Cultured Mammalian Cells' in part 1 of the kit manual was followed by resuspending each sample with 400–600 µl

lysis buffer and proceeding to part 2. The *Arabidopsis* and tomato samples were treated with DNase I as recommended in the kit manual. Due to the larger amount of plasmid used for transformation of maize protoplasts, DNase I treatment was repeated three times for maize samples.

### RNA extraction from *N. benthamiana* tissue

Harvested samples were ground using a mortar and pestle in liquid nitrogen and then mixed with TRIzol LS (up to a 1:3 tissue:TRIzol volume ratio), vortexed for 10 min at room temperature and centrifuged for 5 min at 12,000*g* and 4 °C. The clear fraction was transferred to a new tube, and 0.2 ml chloroform per 1 ml TRIzol LS was added for lysis. Samples were mixed by shaking for 15 s, incubated for 3 min at room temperature and then centrifuged for 15 min at 16,000*g* and 4 °C. The aqueous phase was combined (1:1) with chloroform, shaken, incubated for 3 min and centrifuged for 15 min at 16,000*g* and 4 °C. The resulting aqueous phase was mixed with 0.5 ml isopropanol per 1 ml TRIzol used for lysis, vortexed, incubated for 10 min on ice and centrifuged for 60 min at 21,000*g* and 4 °C. The supernatant was discarded, and the sample was washed twice with 70% ethanol (1:1 ratio of TRIzol used for lysis), vortexed and centrifuged for 5 min at 7,500*g* and 4 °C. Ethanol was removed, and residual ethanol was allowed to evaporate for 7 min at room temperature. Finally, RNA was eluted by adding 500 µl DDW and incubating for 10 min at 42 °C.

### mRNA, polyA and RNA isolation with Dynabeads Oligo(dT)$_{25}$

mRNA was isolated from total RNA of protoplasts and *N. benthamiana* tissue with Dynabeads Oligo(dT)$_{25}$ (Thermo Fisher Scientific, 61005) following the STARR-seq protocol of ref. [57]. Briefly, total RNA was heated to 65 °C for 7 min, followed by incubation on ice for 3 min and at room temperature for 1 min. Two volumes of beads were used for each volume of total RNA. For preparation, the beads were placed on a magnetic separator and washed twice with the same volume of 2× binding buffer[57], followed by resuspension in 0.5× volume of 2× binding buffer. Total RNA was mixed with the washed beads and incubated for 10 min on a rolling shaker. The tubes were placed on a magnetic separator and washed twice with the same volume of washing buffer as the starting volume of the beads. For mRNA elution, the beads were resuspended in 60 µl 10 mM Tris-HCl (pH 7.5) and incubated at 80 °C for 3 min at 750 rpm. The tubes were placed immediately on a magnetic separator and incubated for >1 min. The eluted mRNA was transferred to a new RNase-free tube. The beads were re-eluted with new 30 µl 10 mM Tris-Cl (pH 7.5) buffer, and the two eluates were pooled together.

### MPRA library construction and sequencing

The protocol was adapted from refs. [20,57]. From each replicate, ten reactions with 11 µl mRNA each and a construct-specific primer (P37–P40) were prepared for complementary DNA (cDNA) synthesis using SuperScript IV reverse transcriptase (RT; Thermo Fisher Scientific, 18090010). One of the reactions was used as a no-reverse transcription control, in which the RT enzyme was replaced with RNase-free water. After cDNA synthesis, 1 µl RNase A (200 µg ml$^{-1}$) was added to each reaction, and the samples were incubated at 37 °C for 1 h. The nine RT reactions were pooled and purified with 1.8 volumes of AMPure XP beads (Beckman Coulter, A63880) for each volume of cDNA. The no-RT control was processed separately in the same way as the RT reactions. In the final step, the RT reactions and the no-RT control were eluted in 146 µl and 49 µl 10 mM Tris-HCl (pH 8) buffer, respectively. The purified cDNA was split into two 72-µl aliquots, and each aliquot was used for the preparation of three PCR reactions amplifying the p35S-based transcripts (P17–P20 and P41) and the pTRP1-based transcripts (P33–P36 and P41), respectively. The no-RT control was also split in two 24-µl aliquots, and each aliquot was used for one PCR reaction of the p35S and pTRP1 transcripts, respectively. All PCR reactions were prepared with 24 µl template, 25 µl KAPA HiFi HotStart ReadyMix (Roche Molecular

Systems, KK2601), 0.5 µl forward primer (100 µM) and 0.5 µl reverse primer (100 µM). The samples from *N. benthamiana* tissue were amplified with 21 cycles, whereas the ones from *Arabidopsis*, tomato and maize protoplasts were amplified with 24 cycles. The three reactions from each library (p35S and pTRP1) were pooled and purified with an equal volume of AMPure XP beads. The two no-RT reactions (p35S and pTRP1) were processed separately in the same way. Finally, all samples were analyzed with the 5200 Fragment Analyzer (Agilent, M5310AA) and sequenced on an Illumina NovaSeq 6000 or NextSeq 2000 system in paired-end configuration.

### mRNA synthesis rate measurements in the MPRA assay

The protocol was adapted from ref. [23] and the manual of the Click-iT Nascent RNA Capture Kit (Invitrogen). *Arabidopsis* protoplasts were transformed with MIX3 following the same procedure as for the MPRA in *Arabidopsis* protoplasts. After transformation, the samples were incubated for 5 h and 40 min at 25 °C under constant light (85 µmol m$^{-2}$ s$^{-1}$). A 200 mM stock solution of 5-EU (Click-iT Nascent RNA Capture Kit, Invitrogen, C10365) was added to each sample to a final concentration of 200 µM, and the samples were incubated for an additional 20 min before collection. Total RNA was extracted from the frozen pellets as described above. DNase I treatment was repeated three times for each sample. mRNA was isolated from the total RNA as described above. In the elution step, Dynabeads Oligo(dT)$_{25}$ (Thermo Fisher Scientific, 61005) were resuspended in 55 µl 10 mM Tris-HCl (pH 7.5) and then re-eluted with the first eluate. Five to eight microliters of mRNA was set aside for preparation of libraries from total mRNA, whereas the remaining mRNA was split into three aliquots and each aliquot was used for one Click reaction with 0.25 mM biotin azide following the Click-iT kit manual. Biotinylated mRNA was precipitated from each Click reaction following the manual and resuspended in 25 µl RNase-free water. A bead suspension (3 µl) was added to each aliquot of biotinylated mRNA, and samples were incubated for 30 min at room temperature with rotation at 30 rpm. After incubation, the three aliquots were pooled and washed five times with wash buffer 1 and five times with wash buffer 2 (wash buffers from the Click-iT kit). The bead suspension was resuspended in 50 µl wash buffer 2 and used immediately for cDNA synthesis.

Libraries were prepared from the total mRNA and 5-EU-labeled mRNA bead suspension samples following the procedure for MPRA libraries with a few modifications. The total mRNA samples were diluted to a final volume of 50 µl with RNase-free water. From each sample, four reactions with 11 µl mRNA each and a construct-specific primer (P37–P40) were prepared for cDNA synthesis using SuperScript IV RT (Thermo Fisher Scientific, 18090010). An additional reaction with the remaining mRNA (5.5 µl) was prepared as a no-reverse transcription control, in which the RT enzyme was replaced with RNase-free water. The cDNA reactions with 5-EU-labeled mRNA were incubated on a shaker at 1,500 rpm to prevent settling of the beads on the bottom of the tube, whereas the reactions with total mRNA were incubated without mixing. Following the final step of cDNA synthesis, the reactions with 5-EU-labeled mRNA were immediately placed on a magnetic rack and the supernatants were transferred to new tubes. After this step, the total mRNA and 5-EU-labeled mRNA samples were handled identically. RNase A treatment and purification with AMPure XP beads was performed as described above. In the final step, the RT reactions and the no-RT control were eluted in 73 µl and 24.5 µl 10 mM Tris-HCl (pH 8) buffer, respectively. The purified cDNA from the RT reactions was used to prepare three PCR reactions amplifying the pTRP1-based transcripts (P33–P36 and P41). The no-RT control was used for one PCR reaction of the pTRP1 transcripts. All PCR reactions were incubated for 24 cycles. The second purification with AMPure XP beads was performed as described above. Finally, all samples were analyzed with the 5200 Fragment Analyzer (Agilent, M5310AA) and sequenced on an Illumina NovaSeq 6000 or NextSeq 2000 system in paired-end

configuration. This experiment was carried out with two replicates, yielding 19 and 15 million protoplasts.

To estimate the specificity of the Click-iT Nascent RNA Capture Kit in our system, *Arabidopsis* protoplasts were transformed with a plasmid encoding the Clover protein under the control of the *Petroselinum crispum* ubiquitin (PcUbi) promoter. In the final step, the protoplast suspension was split into three samples of 4.3 million protoplasts, and the samples were incubated at 25 °C under constant light (85 µmol m$^{-2}$ s$^{-1}$) for 6 h. After 4 h, 200 mM 5-EU stock solution was added to one of the samples ('2 h 5-EU') to a final concentration of 200 µM. After 5 h and 40 min, the same amount of 5-EU stock solution was added to the second sample ('20 min 5-EU'), and an identical volume of DMSO was added to the third sample ('no 5-EU'), which served as a negative control. After 6 h of incubation, all three samples were collected and stored at −70 °C. Total RNA was extracted from the samples as described above. RNA labeled with 5-EU was isolated from the total RNA as described above. Biotin azide (0.5 mM) was used for each Click reaction. cDNA was prepared from both the total RNA and 5-EU-labeled RNA samples using the SuperScript VILO cDNA Synthesis Kit (Invitrogen, 11754050). No-reverse transcription controls were prepared for each sample, in which the enzyme mix was heat inactivated at 65 °C for 10 min following the kit manual. qPCR reactions were prepared from all samples using an in-house qPCR mix and oligonucleotides specific for *ACTIN2* (*AT3G18780*) mRNA and *Clover* mRNA. The results were analyzed using the LightCycler 96 system (Roche Diagnostics). Using the 'no 5-EU' samples, the estimated amount of nonlabeled mRNA in the '20 min 5-EU' and '2 h 5-EU' samples was less than 9% and 2% for *ACTIN2* mRNA, respectively, and less than 6% and 1% for *Clover* mRNA, respectively.

### Transient overexpression of GATA TFs in *Arabidopsis* protoplasts

Four plasmids were constructed using the GreenGate reaction[61] by combining the following sequences: the double 35S promoter, N-terminal tag dummy sequence (Addgene ID 48821), the coding sequence of one of the GATA1 (AT3G24050), GATA4 (AT3G60530) and GATA6 (AT3G51080) TFs from *A. thaliana* or the sequence for GFP with a nuclear localization signal (Addgene ID 48826), the C-terminal tag dummy sequence (Addgene ID 48834), the *rbcS* terminator (Addgene ID 48839), the selection cassette containing the sequence for the Venus protein under the seed-specific *At2S3* (*AT4G27160*) promoter[62] and a destination vector. The plasmids containing the coding sequences of the three GATA TFs were synthesized by Twist Bioscience. The CDS of GATA1 was modified to remove the internal BsaI restriction site with a synonymous mutation at the sequence for Gly46 (GGT → GGA). The four final plasmids were transformed into DH5α competent *E. coli* and purified using the Qiagen Plasmid Plus Midi Kit (12943).

A plasmid (10 µg) encoding the GATA TF or GFP was transformed into 200,000 *Arabidopsis* leaf protoplasts. Each construct was transformed in two replicates into two independent protoplast preparations, resulting in four replicates per construct. An additional two samples per protoplast production were transformed with 10 µg GFP control vector and 10 µl elution buffer, serving as positive and negative imaging controls. After 8 h of incubation at 25 °C under constant light (85 µmol m$^{-2}$ s$^{-1}$), the samples were centrifuged at 450*g* for 5 min at room temperature, and the pellets were flash frozen in liquid nitrogen and stored at −70 °C. The controls were imaged shortly before collection under a microscope to check the transformation efficiency: 60% and 70% of protoplasts were successfully transformed in the first and second replicates, respectively.

### RNA sequencing

Total RNA from the overexpression experiment and polyA-selected RNA from the MPRA assay in *Arabidopsis* using MIX1 were used to construct RNA sequencing libraries. These libraries were prepared using the Smart-seq3 protocol[63], with each library constructed in multiple technical replicates and sequenced on a NovaSeq X or NovaSeq 6000 system with paired-end configuration.

### Statistics and reproducibility

All experiments were performed in at least triplicate as described in Methods. No data were excluded from analysis. The statistical tests used for data analysis are described in the main text or in Methods. No statistical method was used to predetermine sample size. The experiments were not randomized, and the investigators were not blinded to allocation during the experiments and outcome assessment.

### Reporting summary

Further information on research design is available in the Nature Portfolio Reporting Summary linked to this article.

### Data availability

Sequencing data have been deposited in the SRA database with accession number PRJNA1009032. Processed data are available in Supplementary Tables 3–5 and 8.

### Code availability

The processing code is available on Zenodo (https://doi.org/10.5281/zenodo.13170729) (ref. 64).

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

## Acknowledgements

We thank J. Neuhold, M. Clavel and V. Nizhynska for technical assistance; A. Levy and D. Ben-Tov for help establishing the protoplast system; Y. Eshed and J. Bindics for sharing seeds; and the Plant Sciences and Next Generation Sequencing facilities at the Vienna BioCenter Core Facilities. We also thank F. Berger, L. Dolan, A. Stark, R.K. Papareddy, B.P. de Almeida, K. Hanada, C.H. Cho and F.K. Lorbeer for fruitful discussions. This work was supported by core funding to M.N. from the Gregor Mendel Institute, ERA-CAPS grant 1001 G+ to M.N. and D.W., and postdoctoral fellowships to Y.V. from EU Horizon 2020 via the VIP² program and Marie Skłodowska-Curie individual fellowships (101028014).

## Author contributions

Y.V., G.H. and A.M.-M. performed the experiments. Y.V. performed the data analysis. Y.V., G.H., A.M.-M., D.W. and M.N. conceived and designed the experiments and wrote the paper.

## Competing interests

The authors declare no competing interests.

## Ethics

This study did not require any specific ethical approval.

## Additional information

**Extended data** is available for this paper at https://doi.org/10.1038/s41588-024-01907-3.

**Correspondence and requests for materials** should be addressed to Yoav Voichek or Magnus Nordborg.

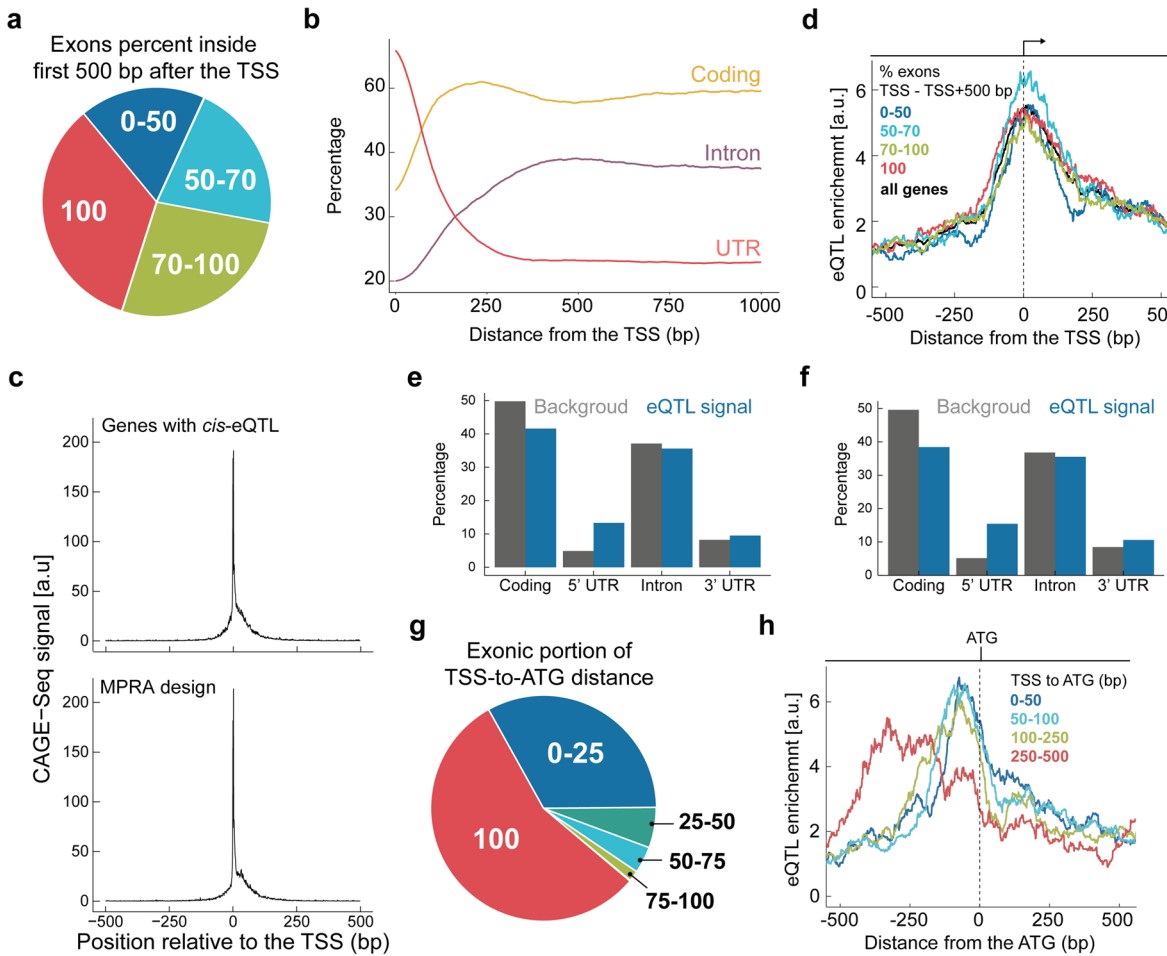

**Extended Data Fig. 1 | No association between exon-intron structure and eQTL enrichment near TSS. (a)** Genes grouped according to fraction of exonic sequence in the 500 bp following the TSS for *A. thaliana* genes. Four groups represent gene portions with different exon content: up to 50%, 50%–70%, 70%–100%, and 100%. This grouping is the same as the grouping in **d**. **(b)** Percentage of genomic sequence coverage by different genomic features as a function of distance from the TSS **(c)** Experimental support for the accuracy of TSS annotations for genes with eQTLs shown in Fig. 1a (top panel) and genes used to design the oligo pool for the massively parallel reporter assay (MPRA, bottom panel). The figure shows cap analysis gene expression followed by sequencing (CAGE-Seq) data from ref. 65 plotted around the TSS of genes. CAGE-Seq reads for each gene are normalized to a total score of 1, and reads from three experimental replicates are summed for each position relative to the TSS across all genes in that group. The peak signal at the TSS confirms the accuracy of the TSS annotations

used in the analysis. **(d)** eQTL enrichment near TSS for genes with varying exonic fraction within the first 500 bp after TSS, shown as in Fig. 1a. Gene counts per group: 914 (0%–50%), 1,044 (50%–70%), 1,179 (70%–100%), 1,102 (100%). **(e-f)** The proportion of eQTL signals within transcripts, as determined by the posterior inclusion probability across various genomic features, compared to the total length of these features in genes where significant associations have been found. Plotted for first **(e)** or second **(f)** batch from ref. 14. **(g)** Genes grouped according to fraction of exonic (5′ UTR) sequence in the TSS-to-ATG regions for *A. thaliana* genes. Five groups represent gene portions with different exonic fractions: 0%–25%, 25%–50%, 50%–75%, 75%–100%, and 100%. These analyses highlight the genomic composition of the TSS-to-ATG region, which is the focus of the analysis, for example in **h** and Fig. 1b. **(h)** eQTL enrichment for genes with different TSS-to-ATG distances, as in Fig. 1b, with data aligned to the ATG and not the TSS. In **a, d**, and **g**, groups exclude the upper limit, that is, A%-B% represents A%≤x < B%.

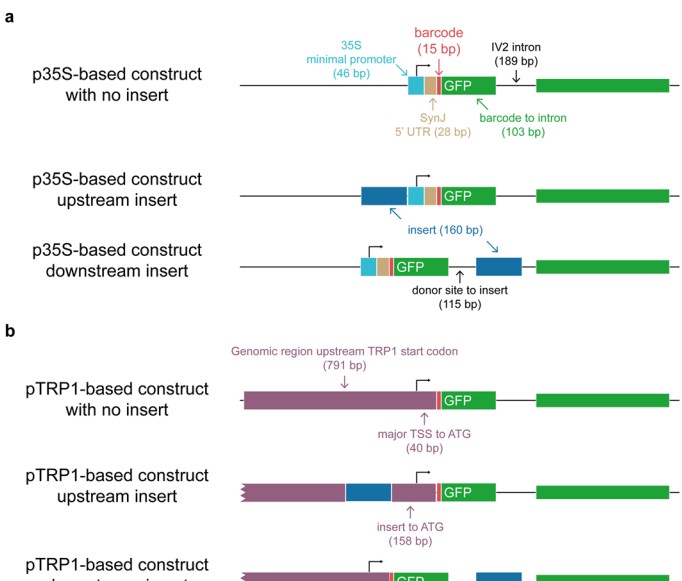

**Extended Data Fig. 2 | MPRA construct sequence layout.** The layout of the constructs used in the MPRA assay, based on a published design[20], for both p35S-based **(a)** and pTRP1-based **(b)** versions. 'No insertion' control constructs are displayed at the top, followed by constructs with upstream and downstream insertions (dark blue). Each construct includes a GFP coding region (green) with an IV2 intron, as well as a 15 bp barcode (red). Distances and lengths of genomic attributes are provided. **(a)** The p35S-based construct incorporates the minimal CaMV 35S core promoter (light blue), followed by the SynJ synthetic 5' UTR (light brown)[58], with the upstream insertion placed just before the minimal core promoter. **(b)** The pTRP1-based construct consists of the 791 bp genomic sequence preceding the coding region of the TRP1 gene (purple), which includes upstream proximal promoter, the core promoter, and 5' UTR. The upstream insertion site is situated within this sequence, 118 bp upstream of the major TSS and 40 bp upstream of the TSS annotation in the TAIR10 database.

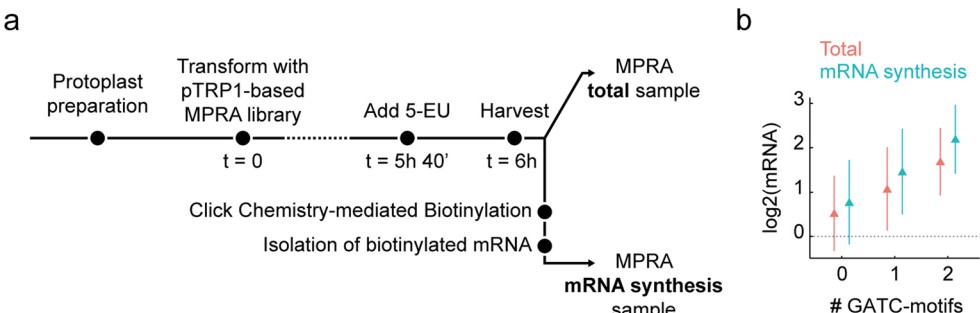

**Extended Data Fig. 3 | mRNA synthesis rate with MPRA. (a)** Measurements of mRNA synthesis rate with MPRA, experimental setup: Arabidopsis protoplasts were transformed with pTRP1-based libraries containing 12,000 fragments positioned downstream of the TSS and incubated for 5 h 40 min at room temperature with constant light. After addition of 5-ethynyl uridine (5-EU) and 20 min incubation[23], total RNA was extracted. Total mRNA was used for regular MPRA. Newly synthesized mRNA was isolated by click reactions followed by selection of biotinylated mRNA using beads. The isolated mRNA was used to generate MPRA libraries. The experiment was performed in two repeats. **(b)** Relative activity of downstream fragments inserted into pTRP1-based constructs as a function of the number of YVGATCBR consensus motifs, as in Fig. 3c. Group sizes: 6,855 (no motif), 956 (1 motif), 119 (2 motifs). Data represent average signal from two replicates of mRNA synthesis MPRA experiments for total mRNA and newly synthesized mRNA. Error bars represent the mean ± 1 standard deviation.

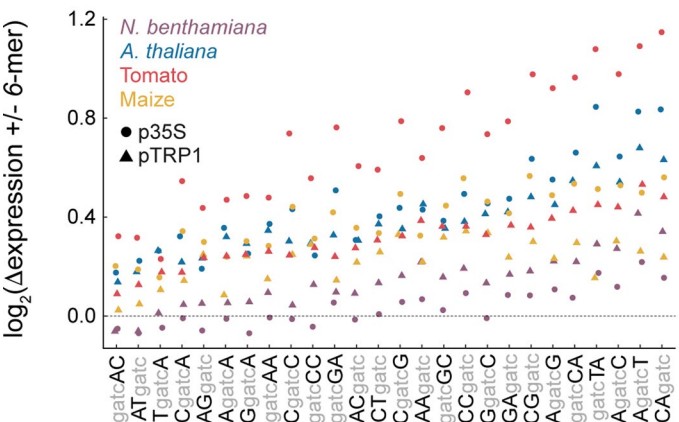

**Extended Data Fig. 4 | Influence of GATC containing-sequences on downstream-MPRA expression.** Examination of the effect of GATC motif's impact on MPRA expression levels using 6-mers. For all 26 unique 6-mer sequences containing a GATC, including their reverse complements, the average $\log_2$ expression difference between sequences having the 6-mer and those lacking the 6-mer is plotted. 6-mers are ordered based on their median effect across the eight experimental setups. Shape indicates the construct backbone (p35S or pTRP1), and color represents the host species. The six 6-mers (CAgatc, AgatcT, AgatcC, gatcTA, gatcCA, and AgatcG) with the highest median effect were used to establish the consensus YVGATCBR sequence defined as the GATC motif, which was then used in downstream analyses.

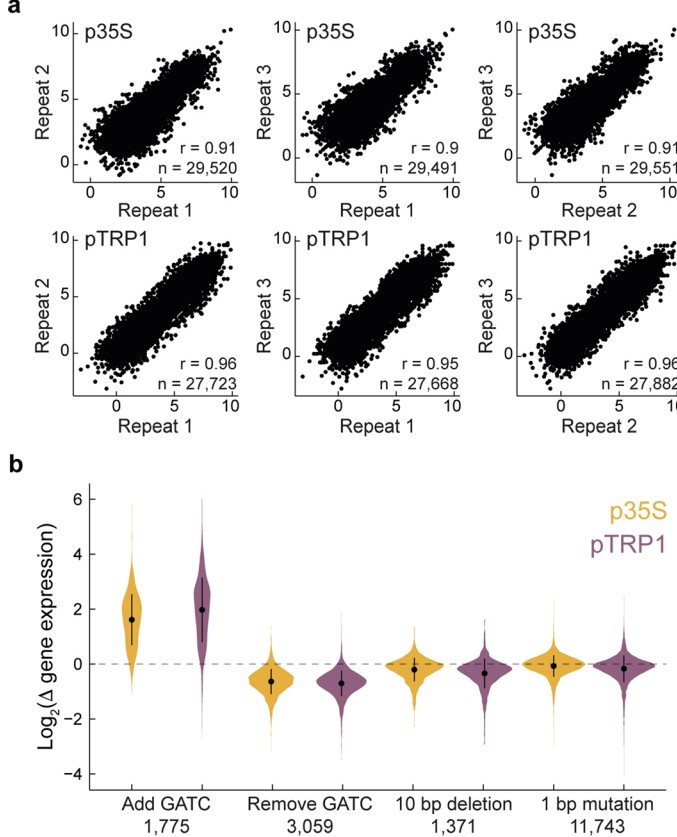

**Extended Data Fig. 5 | MPRA of mutated sequences.** MPRA experiments were performed in Arabidopsis protoplasts using synthetically mutated sequences inserted only in the downstream position. Both p35S- and pTRP1-based libraries were used. The libraries contained 30,000 fragments: 12,000 from the initial pool and an additional 18,000 fragments, each being a variant of one of the original fragments. **(a)** Comparison between each pair from the three replicates, displaying results with the p35S-based library at the top and the pTRP1-based library at the bottom. Pearson's correlation coefficients (*r*) and numbers of compared fragments (*n*) are indicated on each graph. **(b)** $Log_2$ expression ratio between mutated fragments and their original fragment. Mutations encompass: addition of a GATC motif, removal of a GATC motif, 10 bp deletions, and 1 bp changes, which include both deletions and nucleotide substitutions. Color represents the library type, either p35S- or pTRP1-based; numbers below the x-axis indicate the number of fragments in each category. Error bars depict the mean with ±1 standard deviation.

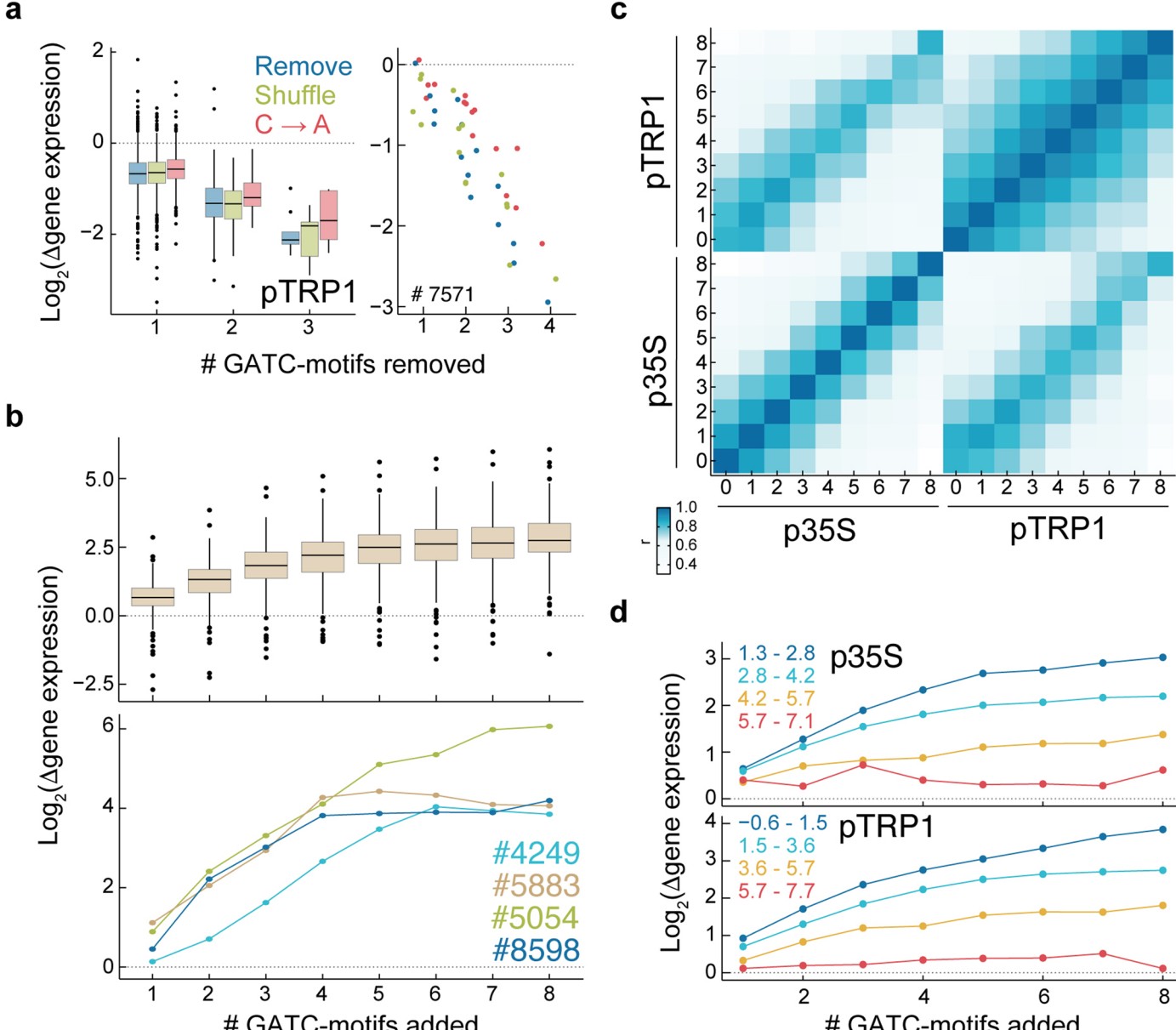

**Extended Data Fig. 6 | Impact of GATC-motif mutations on gene expression in MPRA. (a)** Expression change after removing GATC motifs in a pTRP1-based library, plotted as in Fig. 4a. Illustrated effects across 823 fragments (left) and an example fragment with originally 4 motifs (right). **(b)** Expression changes due to GATC-motifs additions in a pTRP1-based library, mirroring Fig. 4c. Effects across 221 fragments (top) and 4 specific examples (bottom). **(c)** Pearson's correlation coefficients between expression of 221 fragments with varying GATC-motif additions in p35S- and pTRP1-based libraries. **(d)** Influence of GATC-motif additions depends on initial sequence expression. Depicted is the average $\log_2$ expression difference upon motif addition relative to original fragments, for different copy numbers of the added motif. Color indicates original fragment ($\log_2$) expression level ranges, as indicated. Plotted for p35S- (top) and pTRP1-based libraries (bottom). Boxplots in **a,b** represent the median (center line), IQR (box bounds), whiskers (min and max within 1.5 IQR), and outliers (points beyond whiskers).

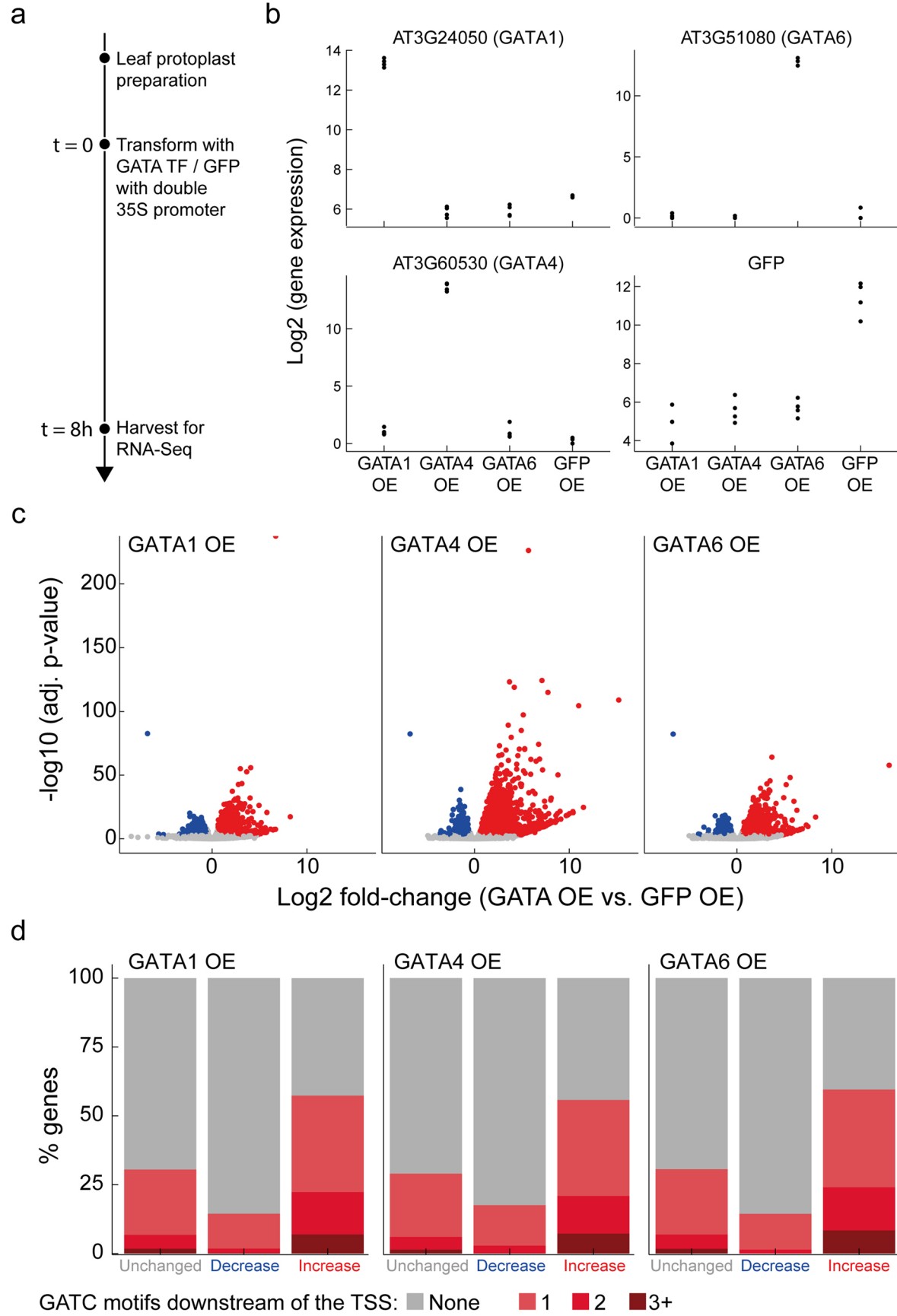

**Extended Data Fig. 7 | See next page for caption.**

**Extended Data Fig. 7 | Transient overexpression of GATA TFs promotes the expression of genes containing the GATC motif. (a)** Experimental setup: Arabidopsis leaf protoplasts were isolated and transformed by PEG-mediated delivery with plasmids carrying GFP, GATA1, GATA4, or GATA6 under the control of a double 35S promoter. RNA was extracted for RNA-Seq analysis eight hours after transformation. Two biological repeats of protoplast isolation were done, with each plasmid transformed twice per isolation replicate, resulting in four replicates per plasmid. In each set of protoplast isolation, an additional sample was transformed with the GFP plasmid to estimate transformation efficiency, which was determined by imaging to be 59% and 69.7% at eight hours post-transformation. **(b)** Verification of overexpression (OE): Gene expression levels of GATA1, GATA4, GATA6, and GFP (as indicated in the panel title) are shown. Expression levels are log2-transformed and plotted for the four plasmids across all four replicates. Note the low level of GFP detection in the GATA TF samples is due to leaky expression of a YFP selection gene driven by a seed coat promoter also found in the plasmids; due to sequence similarities between YFP and GFP, there is cross-alignment of sequencing reads. **(c)** Differential expression analysis: A volcano plot shows the -log10 adjusted Wald test p-values relative to the log2 fold changes in gene expression following OE of each GATA TF compared to GFP OE. Calculations were performed using DESeq2 based on four replicates per experiment[66]. Genes showing at least a 50% change in expression with an adjusted p-value < 0.001 were defined to be differentially expressed and are shown in blue (downregulated) and red (upregulated); all other genes are shown in gray. **(d)** The percentage of genes with a GATC motif within 500 bp downstream of the TSS is shown. These are categorized based on increased, decreased, or unchanged expression (labeled 'Unchanged') as defined in **c**. Further separation is provided for genes with 1, 2, or 3 or more GATC motifs for each GATA TF OE experiment.

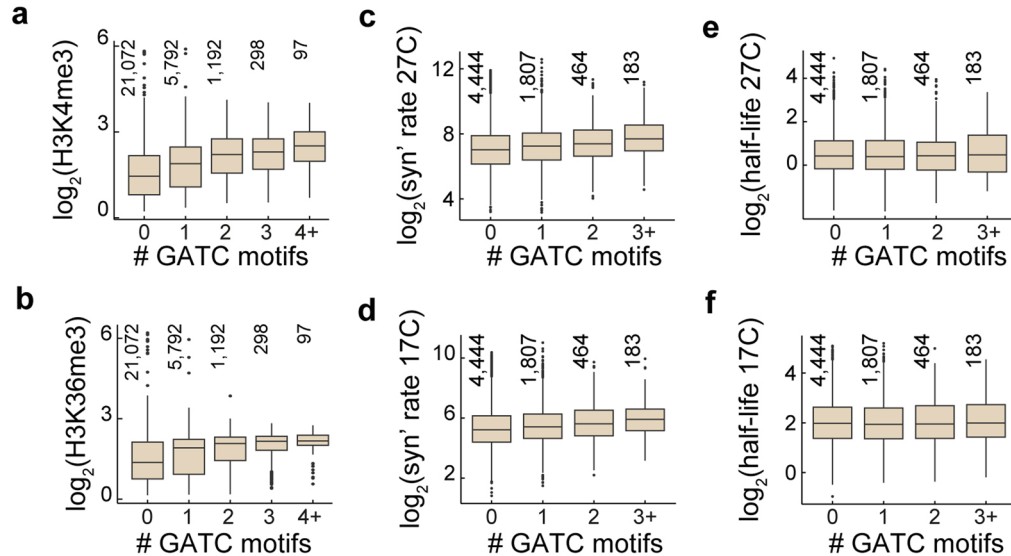

**Extended Data Fig. 8 | Association between GATC-motifs count and histone modification, mRNA synthesis rate, and mRNA half-life. (a-b)** Average log$_2$ enrichment of H3K4me3 **(a)** and H3K36me3 **(b)** across genes[27,67–69] categorized by GATC-motif counts within 500 bp downstream of the TSS, as in Fig. 5a. H3K4me3 and H3K36me3 effect sizes are 0.27 (p-value: 3×10$^{-300}$) and 0.2 (p-value: 4.4×10$^{-194}$), respectively. **(c-d)** mRNA synthesis rates for 7,291 genes at 27 °C **(c)** and 17 °C **(d)**, plotted by GATC-motif counts[29]. Genes with motif counts >=3 are grouped. Effect sizes are 0.17 (p-value: 2.9×10$^{-16}$) at 17 °C and 0.18 (p-value:

8.8×10$^{-19}$) at 27 °C. **(e-f)** mRNA half-lives at 27 °C **(e)** and 17 °C **(f)**, plotted similarly to **c-d**. No significant associations were found between GATC-motif and mRNA half-lives[29], with effect sizes of 0.01 (p-value: 0.4) at 17 °C and −0.01 (p-value: 0.6) at 27 °C. Boxplots display the median (center line), IQR (box bounds), whiskers (min and max within 1.5 IQR), and outliers (points beyond whiskers). Number of genes per box plot is indicated. p-values are calculated using a two-sided t-statistic.

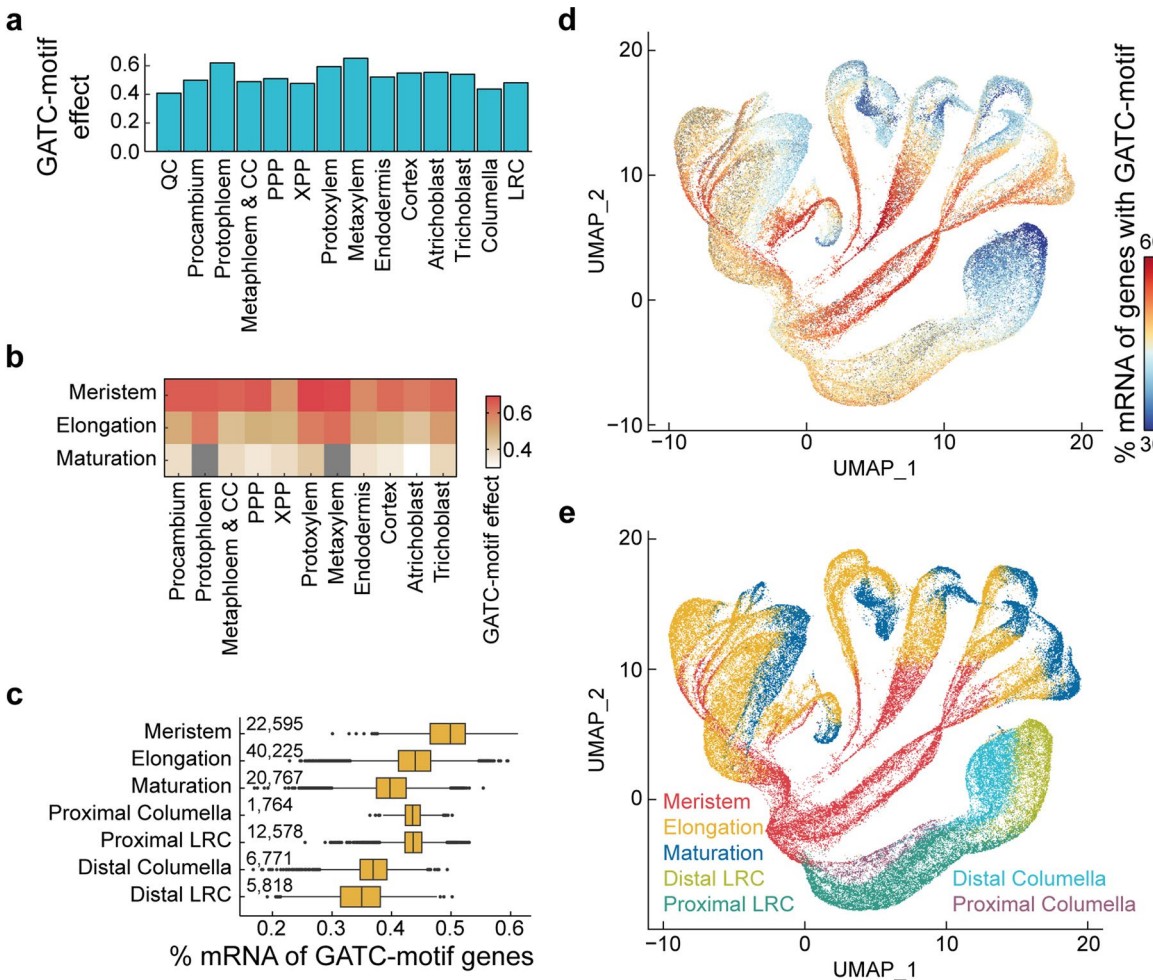

**Extended Data Fig. 9 | Effect of GATC motif on expression across root cell types. (a)** GATC-effect sizes across root cell types, as quantified in Fig. 5g. Gene expression was averaged from scRNA-seq data based on published cell annotations[33]. QC - Quiescent center; LRC - Lateral root cap; CC - companion cell; PPP - phloem pole pericycle; XPP - xylem pole pericycle. **(b)** GATC-effect sizes visualized for different root cell types (x-axis) across root developmental stages (y-axis). Analysis uses cases with at least 150 cells intersecting the two definitions. Gray rectangles indicate cases below this threshold. The quiescent center, defined only as meristematic, is excluded. **(c)** Given that each single cell only provides data on a subset of genes, we used an alternative method to quantify

the GATC-motif effect at the single-cell level. For each cell, mRNA counts from genes with a GATC motif were divided by the total mRNA count. This approach is more robust for sparse data. The quantities plotted reveal significant changes in mRNA species across root developmental stages. Boxplots show the median (center line), IQR (box bounds), whiskers (min and max within 1.5 IQR), and outliers (points beyond whiskers). Number of cells per category is indicated. **(d)** The mRNA percentage from GATC-motif-containing genes is depicted for each of the 110,000 cells in the single-cell Arabidopsis root atlas map[33] **(e)** Root developmental stage definitions from ref. 33, presented as in their original study for comparison.

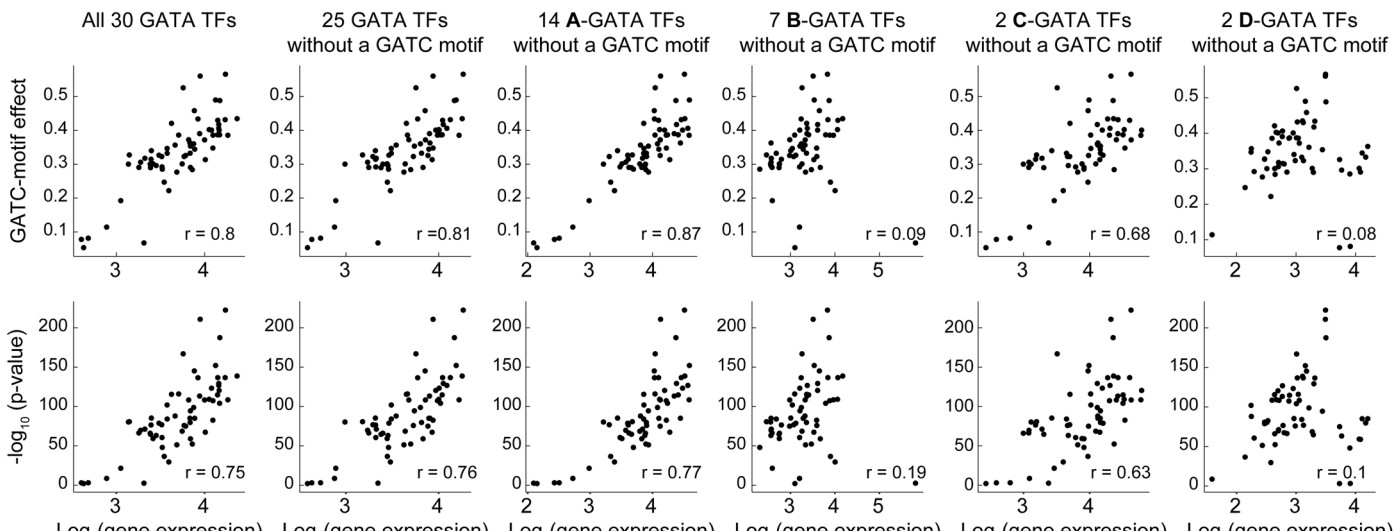

**Extended Data Fig. 10 | Correlation between GATC-motif effect and expression of GATA transcription factors.** Shown is the relationship between the GATC-motif effect (upper row) and the associated fit p-value (lower row), as determined in Fig. 5d, across various tissues part of ref. 31 dataset vs. the mean expression levels of GATA TFs within the same tissues. The average expression is calculated across different groups of GATA TFs: all 30 GATA TFs (1st column), the subset of 25 GATA TFs lacking the GATC motif within 500 bp downstream of the TSS (2nd column), and GATA TFs excluding those with the GATC motif and categorized into subfamilies A (3rd column), B (4th column), C (5th column),

and D (6th column), according to classifications by refs. 25,35. The number of GATA TFs included in the average for each plot is noted in the column titles. The analysis of GATA TFs lacking the motif serves as a control to ensure that the correlation is not influenced by GATA TFs regulated by the GATC motif themselves. Pearson's correlation coefficients are displayed on each scatter plot (*r*). Notably, DAP-Seq analysis by ref. 19 has demonstrated that the A subfamily of GATA TFs binds to the GATC motif downstream of the TSS (Supplementary Fig. 16).

# Reporting Summary

## Statistics

For all statistical analyses, confirm that the following items are present in the figure legend, table legend, main text, or Methods section.

| n/a | Confirmed | |
|---|---|---|
| ☐ | ☒ | The exact sample size (*n*) for each experimental group/condition, given as a discrete number and unit of measurement |
| ☐ | ☒ | A statement on whether measurements were taken from distinct samples or whether the same sample was measured repeatedly |
| ☐ | ☒ | The statistical test(s) used AND whether they are one- or two-sided<br>*Only common tests should be described solely by name; describe more complex techniques in the Methods section.* |
| ☒ | ☐ | A description of all covariates tested |
| ☐ | ☒ | A description of any assumptions or corrections, such as tests of normality and adjustment for multiple comparisons |
| ☐ | ☒ | A full description of the statistical parameters including central tendency (e.g. means) or other basic estimates (e.g. regression coefficient) AND variation (e.g. standard deviation) or associated estimates of uncertainty (e.g. confidence intervals) |
| ☐ | ☒ | For null hypothesis testing, the test statistic (e.g. *F*, *t*, *r*) with confidence intervals, effect sizes, degrees of freedom and *P* value noted<br>*Give P values as exact values whenever suitable.* |
| ☒ | ☐ | For Bayesian analysis, information on the choice of priors and Markov chain Monte Carlo settings |
| ☒ | ☐ | For hierarchical and complex designs, identification of the appropriate level for tests and full reporting of outcomes |
| ☐ | ☒ | Estimates of effect sizes (e.g. Cohen's *d*, Pearson's *r*), indicating how they were calculated |

*Our web collection on statistics for biologists contains articles on many of the points above.*

## Software and code

Policy information about availability of computer code

| Data collection | No software was used for data collection |
|---|---|
| Data analysis | Trim Galore (v0.6.2), bcftools (v1.16), STAR (v2.7.9), vcftools (v0.1.16), Plink v1.9, GEMMA v0.98.5, MASS R library, SusieR (v0.12.35), misha R package, tidyverse R package, MACS2 v2.2.7.1, Bowtie2, Nextflow v22.10.7.5854, nf-core RNA-Seq pipeline v3.6, Salmon v1.10, Seurat v4.3.0, GffRead v0.11.8, OrthoFinder version 2.5.4, BBMap suite, Trim Galore, Salmon v1.5.2, R's 'lm' function. Custom code: https://zenodo.org/records/12968287 |

For manuscripts utilizing custom algorithms or software that are central to the research but not yet described in published literature, software must be made available to editors and reviewers. We strongly encourage code deposition in a community repository (e.g. GitHub). See the Nature Portfolio guidelines for submitting code & software for further information.

## Data

Policy information about availability of data

All manuscripts must include a data availability statement. This statement should provide the following information, where applicable:
- Accession codes, unique identifiers, or web links for publicly available datasets
- A description of any restrictions on data availability
- For clinical datasets or third party data, please ensure that the statement adheres to our policy

Sequencing data have been deposited in the SRA database with accession number PRJNA1009032. Processed data are available in Supplementary Tables 3-5 and 8.

# Research involving human participants, their data, or biological material

Policy information about studies with human participants or human data. See also policy information about sex, gender (identity/presentation), and sexual orientation and race, ethnicity and racism.

| | |
|---|---|
| Reporting on sex and gender | n/a |
| Reporting on race, ethnicity, or other socially relevant groupings | n/a |
| Population characteristics | n/a |
| Recruitment | n/a |
| Ethics oversight | n/a |

Note that full information on the approval of the study protocol must also be provided in the manuscript.

# Field-specific reporting

Please select the one below that is the best fit for your research. If you are not sure, read the appropriate sections before making your selection.

☒ Life sciences ☐ Behavioural & social sciences ☐ Ecological, evolutionary & environmental sciences

For a reference copy of the document with all sections, see nature.com/documents/nr-reporting-summary-flat.pdf

# Life sciences study design

All studies must disclose on these points even when the disclosure is negative.

| | |
|---|---|
| Sample size | MPRA were done in 3 (Maize, Tomato), 4 (Arabidopsis, N. Benthamiana), or 2 (synthesis rate) repeats  Overexpression experiments were done in 4 repeats. |
| Data exclusions | No data was excluded |
| Replication | Each of the MPRA or O.E experiments were done in 3 or 4 repeats, all replicated experiments were used. |
| Randomization | Not applicable, as the experiments conducted in this study were performed one at a time. |
| Blinding | Not applicable, as the experiments conducted in this study were performed one at a time. |

# Reporting for specific materials, systems and methods

We require information from authors about some types of materials, experimental systems and methods used in many studies. Here, indicate whether each material, system or method listed is relevant to your study. If you are not sure if a list item applies to your research, read the appropriate section before selecting a response.

### Materials & experimental systems

| n/a | Involved in the study |
|---|---|
| ☒ | ☐ Antibodies |
| ☒ | ☐ Eukaryotic cell lines |
| ☒ | ☐ Palaeontology and archaeology |
| ☒ | ☐ Animals and other organisms |
| ☒ | ☐ Clinical data |
| ☒ | ☐ Dual use research of concern |
| ☐ | ☒ Plants |

### Methods

| n/a | Involved in the study |
|---|---|
| ☒ | ☐ ChIP-seq |
| ☒ | ☐ Flow cytometry |
| ☒ | ☐ MRI-based neuroimaging |

## Plants

| | |
|---|---|
| Seed stocks | We used reference accessions. M82 (sp▬/sp▬), B73, WT N. benthamiana, and Col-0 |
| Novel plant genotypes | n/a |
| Authentication | n/a |

