## [Peer Review File · Nature Genetics]

Peer Review Information

Manuscript Title: Widespread position-dependent transcriptional regulatory sequences in plants

Corresponding author name(s): Yoav Voichek, Magnus Nordborg

Reviewer Comments & Decisions:

Decision Letter, initial version:

Message: 29th Jan 2024

Dear Dr Voichek,

Your Letter, "Widespread position-dependent enhancers in plants" has now been seen by 3 referees. You will see from their comments copied below that while they find your work of considerable potential interest, they have raised quite substantial concerns that must be addressed. In light of these comments, we cannot accept the manuscript for publication, but would be very interested in considering a revised version that addresses these serious concerns.

In brief, the three reviewers span the full range of support for your study, but highlight a range of conceptual and technical issues that preclude publication at this stage.

Referee #1 is enthusiastic and supportive of publication. They've made a few requests for additional analyses to deepen the novel insight, which seem quite doable.

Reviewer #2 says your work is a "tour de force", but provides a nuanced and thoughtful review examining various important technical aspects of the MPRA and analysis (e.g. core promoter-enhancer compatibility). They also highlight that it's not clear whether the regulatory elements assayed are truly enhancers.

Referee #3 is straightforwardly negative, but goes into much detail explaining why they are,

citing much past literature that reduces the novelty of your study, and a distinct set of technical/analytic concerns; they do, however, provide detailed guidance for each issue raised. Again, they ask whether these are enhancers, as opposed to another type of regulatory element with a distinct mechanism.

In our reading of these reports, we think there is a potential path to publication, but it is clear that Referees #2 and - especially - #3 will need to see major improvements to be supportive. It seems to us that the central question that remains to be answered is: are these really enhancers? We think that doing so will require all the *in silico* requests to be fully addressed, and we suspect that Reviewer #3 will also expect the gold-standard genetic manipulation experiment providing "direct evidence of enhancer-like function".

We hope you will find the referees' comments useful as you decide how to proceed. If you wish to submit a substantially revised manuscript, please bear in mind that we will be reluctant to approach the referees again in the absence of major revisions.

To guide the scope of the revisions, the editors discuss the referee reports in detail within the team, including with the chief editor, with a view to identifying key priorities that should be addressed in revision and sometimes overruling referee requests that are deemed beyond the scope of the current study. We hope that you will find the prioritised set of referee points to be useful when revising your study. Please do not hesitate to get in touch if you would like to discuss these issues further.

If you choose to revise your manuscript taking into account all reviewer and editor comments, please highlight all changes in the manuscript text file. At this stage we will need you to upload a copy of the manuscript in MS Word .docx or similar editable format.

*2) If you have not done so already please begin to revise your manuscript so that it conforms to our Letter format instructions, available here. Refer also to any guidelines provided in this letter.

Please be aware of our guidelines on digital image standards.

[Redacted]

If you wish to submit a suitably revised manuscript we would hope to receive it within 6 months. If you cannot send it within this time, please let us know. We will be happy to consider your revision so long as nothing similar has been accepted for publication at Nature Genetics or published elsewhere. Should your manuscript be substantially delayed without notifying us in advance and your article is eventually published, the received date would be that of the revised, not the original, version.

Nature Genetics is committed to improving transparency in authorship. As part of our efforts in this direction, we are now requesting that all authors identified as 'corresponding author' on published papers create and link their Open Researcher and Contributor Identifier (ORCID) with their account on the Manuscript Tracking System (MTS), prior to acceptance. ORCID helps the scientific community achieve unambiguous attribution of all scholarly contributions. You can create and link your ORCID from the home page of the MTS by clicking on 'Modify my Springer Nature account'. For more information please visit please visit www.springernature.com/orcid.

Thank you for the opportunity to review your work.

Sincerely,

Michael Fletcher, PhD
Senior Editor, Nature Genetics

ORCID: 0000-0003-1589-7087

Referee expertise: plant genetics and regulatory genomics.

Reviewers' Comments:

Reviewer #1:

Remarks to the Author:

I found this to be a fascinating manuscript and enjoyed reading it. The authors wove together a compelling set of observations to demonstrate the role of sequences downstream of the TSS in influencing gene transcription and in particular highlight the role of a specific motif. The combination of using natural variation, large scale screens and deletion analyses are very useful in pulling together the observations. This work highlights unique aspects of cis-regulatory control of gene expression in plants and will open new avenues for research into cis-regulatory variation. I did not have major issues with the findings that were presented but found a number of details were hard to follow or missing. These are detailed below in the minor comments but the major ones for my understanding were as follows:

More clear description of the MPRA assay to document whether orientation of the synthesized oligos was controlled or not. I am quite curious if the motif has any orientation effects when present downstream of the sequence or not.

Clearer description of the "GATC motif". Is this a 4-mer, a 6-mer or an 8-mer. It became difficult to follow this.

It is likely beyond the scope of this paper but I was left quite curious about some comparative genomics aspects of this work. Do orthologs in different species show any level of conservation for 5' UTR length or for GATC motif presence?

The frequency of the GATC motif in 5' UTR, coding sequences and introns was not all that clear. I am curious if the analyses in later portions of the paper would find similar effects for GATC motifs in all these contexts across species or not.

Specific comments:

Lines 37-39; I know this analysis is using previously published datasets but it could be useful to mention the tissue type being used.

Lines 48-55: the authors do several tests to examine the potential bias in the locations of cis eQTL to assess the potential that these cis eQTL may influence transcript stability. It might be helpful to perform a test of the enrichment (or depletion) of cis eQTL within specific regions (ie 5' UTR, coding exon, intron and 3'UTR). By assessing the frequency of cis eQTL in these regions relative to the total length accounted for by these regions it should be possible to directly examine the contributions of variation in different types of regions. I am not saying this should replace the current analyses but it would be helpful for the reader to see the actual distribution.

Lines 54-59: I thought this was a very interesting finding. The propensity for having CREs downstream of the TSS seems highly influenced by the length of the 5' UTR. Just a quick clarification could be useful - how well characterized are the actual TSS sequences in Arabidopsis and do these exhibit any natural variation among accessions? This might be helpful to non-Arabidopsis focused researchers.

One other question - given the observation that genes substantially vary for the position of CREs relative to the TSS given and that some TFs prefer to bind upstream of the TSS while others prefer downstream binding this suggests the UTR length may be a selected factor for the potential for downstream TFs to act on a gene. Is there much evidence that 5' UTR length has any conservation between plant species?

Line 73: What happens to the plots in Figure S4 (or in 1B) if you center the plot on the ATG (rather than the TSS)?

Lines 92-92: I could not tell from the description of the assay in the text whether the fragments were directionally inserted or whether the fragments could be in the sense and antisense orientation. If the latter is true did you assess whether orientation had equivalent effects for upstream and downstream sequences?

Line 157-161: Do the sequences identified in this analysis match any of the predicted motifs for the TFs that have biased DAP-seq peaks for downstream regions (ie Figure 1D)?

Line 163 - on. I got a bit confused about how the authors were using the term "GATC motif". Does this truly refer to the 4-mer GATC or are they using this for the 8-mer defined in line 159. This was not clear through the analyses and figures as the current usage seemed to imply a 4-mer but I wasn't sure that was how this was being used.

Line 204-207 - when you added these GATC motifs to these upstream and downstream fragments were they only tested in the downstream location or did this get test in both upstream and downstream locations?

Lines 238-241 - why test 6-mers rather than the defined 8-bp GATC motif defined earlier?

Line 251-284: What is the overall expression of GATA TF family members in these tissues? Is there any correlation between overall GATA expression and the tissues with more activity for the GATC motif?

Reviewer #2:

Remarks to the Author:

This manuscript is a tour de force study to investigate the important question of whether plant enhancers work similarly as animal enhancers, independently of orientation and position. The study used massively parallel reporter assays (MPRA) to demonstrate the regulatory role of sequences downstream of the transcription start site (TSS) in plant activation of transcription. The study identifies a specific motif (GATC) that functions in a dose-dependent fashion and appears to be associated with activation of transcription driven by downstream sequences and which doesn't function upstream of the TSS. MPRA studies were complemented with the re-analysis of published results to demonstrate that the effects of the GATC motif is not associated with mRNA stability, and that it is conserved across species. The hypothesis that the GATC motif acts as a regulatory module, operating as a rheostat in tuning gene expression between cell types, is attractive but largely derived from correlations (nevertheless, worth mentioning it).

Comments

1. The title sounds good as it suggests a generality in the findings, but the reality is that the study only focuses on putative enhancers containing a "GATC" motif and presumably recognized by GATA TFs. A more honest title (even if less spicy) would provide a better service to the community.

2. How much of the effect observed regarding the conclusion of position with regards to the TSS of the sequences tested could be an effect of interactions between the putative enhancers and core promoter elements? Perhaps the idea that the core promoter is “universal and largely exchangeable between genes” requires revisiting? For example, in Fig. 2C, how is the comparison of each enhancer with the two minimal promoters? For example, does 35S enhancer work better with a minimal 35S?

3. The value of conducting the MPRA experiments in four species is questionable – it certainly complicates a straight-forward story and does not really add conceptually much. Moreover, it is concerning how the results should be interpreted in the light of what was found in the study by Jores et al (2021 Nature Plants) where they clearly show that core promoters behave differently in monocots and dicots, depending on the origin.

4. How many of the sequences tested in the Jores et al manuscript contained GATC motifs? It would be interesting to see if the hypothesis proposed here are supported by those experiments (expectation is that GATC should be significantly less represented in active core promoters).

5. (Lines 148 – 155 and elsewhere) The idea that binding of a TF to a 6-mer sequence is sufficient to drive gene expression, particularly in a system in which the TF is not over-expressed is, in my opinion, a bit out of date. While true that 6 bp are often sufficient in vitro for TF binding, there is an extensive literature that demonstrates that in vivo, TFs recognize much longer sequences. Surely more activity is possible if repeats of the sequence are included, but that is not a very common situation for most TFs. Perhaps of even greater concern is that no TF works on its own with large complexes of TFs forming what is believed to be an active TF complex. I recommend some re-writing to make it clear that the situation is not as simple as written.

6. In Fig. S5, the bimodal distribution can easily be explained by nucleotide distribution bias around the TSS – indeed, Inr elements usually have a pretty conserved motif. Was this considered? And if it is, how does the result/interpretation change?

7. Since the authors analyzed the published Arabidopsis DAP-seq results – when they identified GATC motifs downstream of the TSS, were they preferentially recognized methylated or unmethylated (irrespective of the TF)? Clearly GATC methylation would not fall into either GC or GNC methylation, might the A be methylated? And if so, would that be more suggestive of a role in transcription or post-transcription?

8. Whether GATA TFs are responsible for recognizing the GATC motif described here is in many ways irrelevant. It is correct that some GATA TFs have been shown to bind GATC better than GATA, but the authors don't focus specifically on this aspect. Hence, I would recommend that the discussion of whether GATA TFs are responsible for binding the GATC motif should be eliminated (together with Note S1). As a word of caution, Zn-finger proteins (such as GATA TFs) are very abundant in plants and they often encode nucleic acid binding proteins (beyond TFs). GATC happens to be palindromic, also raising the question of whether it could be the core of a larger motif recognized by dimeric TFs.

9. The title of Fig. 2 should read: "Position-dependent enhancers reside inside transcribe regions". What are the y-axis units for Figs. 2B and 2D (right inset)?

10. I found the activity of the known enhancers shown in Fig. 2C to be pretty modest. Is this because the activity of the minimal promoters was already high? If they were, it might be worth mentioning and comparing the two "minimal" promoters used (I understand that the TRP1 is not formally a minimal promoter). Unfortunately, only ratios are shown.

11. In Fig. 3A, what is the correspondence of sequences when comparing the two core promoters? In other words, are the same sequences providing strong activation regardless of the core promoter used?

12. The authors should reexamine whether the sequences that they are analyzing (containing one or more GATC motifs) truly fit the definition of enhancers, and whether they have enough information to call them enhancers. While it is clear that plant and animal enhancers are probably different (several recent papers have discussed this, one of several that come to my mind includes <https://pubmed.ncbi.nlm.nih.gov/34918159/>, but there are certainly others).

Reviewer #3:

Remarks to the Author:

The authors are asking an interesting question: What is the role of eQTL downstream of the transcription start site (TSS) in plants and how do the underlying sequences affect RNA levels? Unfortunately, they use a previously developed MPRA design that is ill suited to answer this question. The MPRA tests sequences for activity in only two ways: either as potential enhancers upstream of the TSS or inserted into a reporter gene intron at a fixed distance downstream of the TSS. This design fails to capture the contextual origin of the

tested sequences (i.e. 5'UTR, intron in 5'UTR, exon, intron, etc.), ignoring the wealth of prior knowledge on how 5'UTRs, intron, introns in 5'UTRs and other transcribed sequences can affect RNA levels (Chung et al. 2006; Alan B. Rose and Beliakoff 2000; Alan B. Rose 2004; A. B. Rose and Last 1997; Laxa et al. 2016; Laxa 2016; Wang, Liu, and Guo 2023; Callis, Fromm, and Walbot 1987). Given this assay design, it is not surprising that the authors identify a known sequence motif that affects RNA levels when residing in introns. Although the authors present a large amount of carefully generated data from different expression systems, their strong claim of widespread position-dependent plant enhancers is not well supported by the presented evidence and the expression-enhancing motif GATC that is discussed as a major finding is not novel.

I am most puzzled by the insistence of the authors that they have identified a fundamental regulatory phenomenon specific to plants. eQTL in human also reside downstream of the TSS in about equal (or higher!) numbers – see figure reproduced from a published study (Strunz et al. 2018; Figure 2) in the review attachment. Although the scale differs on the x-axis compared to this manuscript's Figure 1A, the majority of signal in human is visible close to the TSS and gene size differs profoundly between human and Arabidopsis. Most assuredly, if the authors had conducted this experiment with human sequences, they would have found that these sequences also affect expression differently depending upon their position upstream or downstream of the TSS. While it is meaningful to consider regulatory features that fundamentally differ between plants and animals – e.g., the presence of CTCF-dependent topological associated domains in animals (Schmitz, Grotewold, and Stam 2022) – the phenomenon investigated here is shared.

The authors' claim that their study identifies widespread position-dependent enhancers in plants is particularly irksome. Enhancers are defined as position-independent sequence elements that enhance transcription of their target gene in a condition- or tissue-specific manner, mediated by transcription factor binding at both enhancer and promoter, often involving looping that allows for long-distance interactions (Schmitz, Grotewold, and Stam 2022). Are these position-dependent elements bound by TFs in the respective genomes and is this binding relevant for their function? The DAP-seq data (O'Malley et al. 2016) presented in Figure 1 are an aggregate plot for 529 Arabidopsis TFs (in C) and DAP-seq peak enrichments consolidated for TF families (in D) rather than direct evidence of functionally relevant TF binding at GATC motifs. Likewise, element interactions with target gene promoters were not tested or even inferred with existing 3D data.

The MPRA used in this manuscript does not distinguish between enhanced transcription versus enhanced RNA stability or other mechanisms that may increase RNA levels. Using published data, the authors show that polymerase II occupancy (Figure 4C) and active histone marks increase with the number of GATC motifs, and show a correlation with mRNA

synthesis but not mRNA half-life in Arabidopsis data. I agree that these results are consistent with the motif affecting RNA expression rather than RNA half-life; however, the motif may affect secondary structure, polymerase II pausing or processivity etc. rather than function as a bona fide enhancer (i.e. interact with a target gene promoter to enhance transcription). The strong claims made by the authors require direct evidence of enhancer-like function. Given the ease with which such experiments, including genetic manipulations, can be conducted in the model Arabidopsis, it is disappointing that such experiments have not been conducted.

Below are more detailed considerations, specific suggestions, and citations of prior work relevant to this manuscript.

Major issues:

1. The study lacks novelty: It is known that sequences in introns can increase mRNA levels both transcriptionally and post-transcriptionally (Callis, Fromm, and Walbot 1987; Laxa 2016; Shaul 2017; Le Hir, Nott, and Moore 2003). Notably, this is true in animals and plants. Rose and co-workers demonstrated that a GATC-like sequence motif increases expression in a dose-dependent manner (Alan B. Rose et al. 2016; Gallegos and Rose 2019) when residing within the transcribed sequence but not when residing upstream of the TSS (Gallegos and Rose 2017, 2019). A recent study showed that known plant enhancers and a highly active viral enhancer are inactive in the transcribed region in the MPRA design used in this manuscript (Jores et al. 2020). The authors need to discuss how their findings add to the rich existing knowledge. A promising avenue would be to generate mechanistic insights as to the function of the GATC motif with experiments geared towards identifying possibly bound proteins, and testing effects on transcription pre-initiation complex formation and on polymerase II pausing and processivity. Other possibilities include considerations of secondary structure, nucleosome position and supercoiling, all sequence-content- and position-dependent features that affect expression.

2. The authors imply that the expression-enhancing GATC motif functions as a transcriptional enhancer by binding GATA TFs (line 225). In contrast, earlier studies argue for an unknown, post-transcriptional mechanism (A. B. Rose and Last 1997; Samadder et al. 2008). The authors' suggestion is inconsistent with existing knowledge. GATA TFs are evolutionarily conserved zinc finger TFs that are found in plants, animals and fungi (Schwechheimer, Schröder, and Blaby-Haas 2022). GATA TFs are involved in brassinosteroid signaling and photomorphogenesis, processes controlling skotomorphogenic development, chlorophyll biosynthesis, chloroplast development, photosynthesis, and stomata formation, root, leaf, and flower development, seed dormancy and various stress responses. At least some of the

many plant GATA TFs must function by recognizing the GATA TF motif (Dap-seq AGATCT for GATA11, consensus motif W-G-A-T-A-R) upstream of the TSS to control target gene expression (O'Malley et al. 2016; Schwechheimer, Schröder, and Blaby-Haas 2022). However, the study under review found that the GATC motif is not active upstream of the TSS, making it considerably less likely that this motif is bound by GATA TFs and enhances transcription in this way. In stark contrast, Meng et al. shows that transcriptional enhancers derived from introns of Arabidopsis genes retain their activity when cloned upstream of the TSS (Meng et al. 2021), as expected for enhancers. Weigel, one of the senior authors, previously provided evidence for intronic enhancers, reporting binding sites for LEAFY and WUSCHEL, both AGAMOUS activators, in the 3 kb AGAMOUS intron (Hong et al. 2003). Taken together, the GATC motif is more likely to function via the post-transcriptional intron-mediated enhancement mechanism than as a transcriptional enhancer. Without additional mechanistic evidence, the authors should focus their discussion on the modes of GATC action that are consistent with existing data.

3. The authors use an intron for the "downstream MPRA" to "rule out effects due to altered sequence of the mature mRNA and thus minimizing effects on mRNA stability" (lines 89-90). This is a curious argument as many of these sequences presumably reside outside of introns in the RNA and exert their function in this context. In any case, the argument holds only true if the intron is spliced out equally efficiently. The introduction of foreign sequences into introns can lead to incorrect or inefficient splicing (Yofe et al. 2014). Such mis-splicing could have a strong effect on the levels of the corresponding RNAs and would lead to misinterpretation of the MPRA data. The authors should assess the splicing efficiency of their library, or do so for selected subsets (e.g., for sequences with high or low reporter gene expression).

4. While the experiments described in this study generated a large amount of data, much of this data remains unanalyzed. For example, species-specific differences are ignored even though they are clearly present. Results from the dicots Arabidopsis, tomato and tobacco appear more closely correlated with one another than with results from the monocot maize; this is especially noticeable in the "upstream MPRA". Given the strong differences in GC content between dicot and monocot genomes, GC content is a strong candidate for explaining these differences.

5. It is striking that the strongest effects on transcription are observed in the "upstream MPRA", yet the authors focus almost exclusively on the results of the "downstream MPRA". Further careful analysis, in particular of the "upstream MPRA" data, could identify sequence motifs that are associated with species-specific or position-dependent element activity. Such

additional analyses might considerably improve the novelty and hence the value of this study.

6. The authors show that the GATC motif is active downstream of the TSS but not upstream of the TSS in the Arabidopsis gene expression data. Does the same hold true in the MPRA data? In other words, is the GATC motif associated with higher (or lower) reporter gene expression in the "upstream MPRA"? I would be interested to see the results for the GATC motif in the "upstream MPRA" analogous to those shown in Figure 3 for the "downstream MPRA".

7. The figures need to be revised to improve clarity in order to allow readers to draw their own conclusions. Here are some suggestions for changes to the main Figures (corresponding changes should also be made to the Supplementary Figures):

- The sample size must be indicated for all box and violin plots and for the correlation scatter plots
- Axis labels should be more descriptive. Example of insufficient axis labels are: "Repeat 1" and "Repeat 2" (Fig. 2B), "upstream" and "downstream" (Fig 2D), "Tomato - p35S" and "N. be - pTRP1" (Fig. 3B)
- For easier comparisons, the authors should use the same scales for all sub-plots in Figs. 2B, 2E, and 3C
- Using partially transparent dots or a different plot type (e.g., a hexbin plot) would help to visualize the distribution in the correlation scatter plots (Fig. 2B and 2D)
- In Fig. 1D, the authors should use a linear color scale across the whole range (currently the range 0 to ~60 is all white)
- In Fig. 3B and 3C, it is not clear what the dot and lines represent (mean +/- sd?). This should be described clearly. Furthermore, converting the plots to violin plots could help better visualize the distribution of the (up to thousands of) data points
- In Fig. 4B, it is unclear whether the shaded area with GATC containing 6-mers applies to both the "Upstream" and "Downstream" categories or to just one of the two. Another way of highlighting the GATC-containing 6-mers would improve clarity.
- In Fig. 5, it is not described what the individual rows in the heatmap and the points in the boxplots represent. Are these different transcriptome data sets? This needs to be explained and the sample size must be indicated for each species.

Minor issues:

8. The title is not accurate for the reasons discussed above.

9. The introduction, just as the title, does not match the manuscript. Instead of discussing the evolution of gene regulation in plants and animals, it would be helpful to introduce concepts such as intron-mediated enhancement and previous findings on position-independent plant enhancers residing upstream of the TSS and in introns.

10. In the deep mutational scan, the authors find "additional sequences that do not include GATC motifs as important for enhancing activity" (lines 176-178). The authors should list and discuss these additional sequences.

11. It is unclear how the authors generated the GATC motif. The text "we combined the six 6-mers with the strongest effect into an 8-bp YVGATCBR motif" (lines 158-159) is insufficient. Which 6-mers were chosen and why? Why were only six 6-mers utilized? How were they combined? Did the authors try alternative methods, like searching for motifs enriched in the most active sequences? The authors should include a detailed description of how the motif was generated and ideally make the motif available in a suitable format (as a .meme file?) for use by others.

12. On lines 122-124, the authors state: "Enhancers were more effective in the CaMV 35S promoter construct, than in the TRP1 promoter construct, in which fragments insertions disrupted the TRP1 genomic sequence." Since the plots for the 35S and TRP1 promoter constructs in Fig. 2E and Fig. S9A are normalized to different control constructs, it is not possible for readers to directly compare the two. A plot for the direct comparison of reporter gene expression across the two different promoters should be included in the manuscript. Furthermore, if the disruption of the TRP1 genomic sequence is the cause for the observed differences, then the two promoters should behave similarly in the "downstream MPRA" context. This is not the case, though, and the authors should discuss this finding and possible explanations.

References:

- Callis, J., M. Fromm, and V. Walbot. 1987. "Introns Increase Gene Expression in Cultured Maize Cells." *Genes & Development* 1 (10): 1183–1200.
- Chung, Betty Y. W., Cas Simons, Andrew E. Firth, Chris M. Brown, and Roger P. Hellens. 2006. "Effect of 5'UTR Introns on Gene Expression in Arabidopsis Thaliana." *BMC Genomics* 7 (1): 120.
- Gallegos, Jenna E., and Alan B. Rose. 2017. "Intron DNA Sequences Can Be More Important than the Proximal Promoter in Determining the Site of Transcript Initiation." *The Plant Cell* 29 (4): 843–53.
- Gallegos, Jenna E., and Alan B. Rose. 2019. "An Intron-Derived Motif Strongly Increases Gene

- Expression from Transcribed Sequences through a Splicing Independent Mechanism in *Arabidopsis Thaliana*." *Scientific Reports* 9 (1): 13777.
- Hong, Ray L., Lynn Hamaguchi, Maximilian A. Busch, and Detlef Weigel. 2003. "Regulatory Elements of the Floral Homeotic Gene *AGAMOUS* Identified by Phylogenetic Footprinting and Shadowing." *The Plant Cell* 15 (6): 1296–1309.
- Jores, Tobias, Jackson Tonnie, Michael W. Dorrity, Josh T. Cuperus, Stanley Fields, and Christine Queitsch. 2020. "Identification of Plant Enhancers and Their Constituent Elements by STARR-Seq in Tobacco Leaves." *The Plant Cell* 32 (7): 2120–31.
- Laxa, Miriam. 2016. "Intron-Mediated Enhancement: A Tool for Heterologous Gene Expression in Plants?" *Frontiers in Plant Science* 7: 1977.
- Laxa, Miriam, Kristin Müller, Natalie Lange, Lennart Doering, Jan Thomas Pruscha, and Christoph Peterhänsel. 2016. "The 5'UTR Intron of *Arabidopsis* GGT1 Aminotransferase Enhances Promoter Activity by Recruiting RNA Polymerase II." *Plant Physiology* 172 (1): 313–27.
- Le Hir, Hervé, Ajit Nott, and Melissa J. Moore. 2003. "How Introns Influence and Enhance Eukaryotic Gene Expression." *Trends in Biochemical Sciences* 28 (4): 215–20.
- Meng, Fanli, Hainan Zhao, Bo Zhu, Tao Zhang, Mingyu Yang, Yang Li, Yingpeng Han, and Jiming Jiang. 2021. "Genomic Editing of Intronic Enhancers Unveils Their Role in Fine-Tuning Tissue-Specific Gene Expression in *Arabidopsis Thaliana*." *The Plant Cell* 33 (6): 1997–2014.
- O'Malley, Ronan C., Shao-Shan Carol Huang, Liang Song, Mathew G. Lewsey, Anna Bartlett, Joseph R. Nery, Mary Galli, Andrea Gallavotti, and Joseph R. Ecker. 2016. "Cistrome and Epicistrome Features Shape the Regulatory DNA Landscape." *Cell* 166 (6): 1598.
- Rose, A. B., and R. L. Last. 1997. "Introns Act Post-Transcriptionally to Increase Expression of the *Arabidopsis Thaliana* Tryptophan Pathway Gene *PAT1*." *The Plant Journal: For Cell and Molecular Biology* 11 (3): 455–64.
- Rose, Alan B. 2004. "The Effect of Intron Location on Intron-mediated Enhancement of Gene Expression in *Arabidopsis*." *The Plant Journal: For Cell and Molecular Biology* 40 (5): 744–51.
- Rose, Alan B., and Jason A. Beliakoff. 2000. "Intron-Mediated Enhancement of Gene Expression Independent of Unique Intron Sequences and Splicing." *Plant Physiology* 122 (2): 535–42.
- Rose, Alan B., Amanda Carter, Ian Korf, and Noah Kojima. 2016. "Intron Sequences That Stimulate Gene Expression in *Arabidopsis*." *Plant Molecular Biology* 92 (3): 337–46.
- Samadder, Partha, Elumalai Sivamani, Jianli Lu, Xianggan Li, and Rongda Qu. 2008. "Transcriptional and Post-Transcriptional Enhancement of Gene Expression by the 5' UTR Intron of Rice *Rubi3* Gene in Transgenic Rice Cells." *Molecular Genetics and Genomics: MGG* 279 (4): 429–39.
- Schmitz, Robert J., Erich Grotewold, and Maïke Stam. 2022. "Cis-Regulatory Sequences in Plants: Their Importance, Discovery, and Future Challenges." *The Plant Cell* 34 (2): 718–41.

Schwechheimer, Claus, Peter Michael Schröder, and Crysten E. Blaby-Haas. 2022. "Plant GATA Factors: Their Biology, Phylogeny, and Phylogenomics." *Annual Review of Plant Biology* 73 (1): 123–48.

Shaul, Orit. 2017. "How Introns Enhance Gene Expression." *The International Journal of Biochemistry & Cell Biology* 91 (Pt B): 145–55.

Strunz, Tobias, Felix Grassmann, Javier Gayán, Satu Nahkuri, Debora Souza-Costa, Cyrille Maugeais, Sascha Fauser, Everson Nogoceke, and Bernhard H. F. Weber. 2018. "A Mega-Analysis of Expression Quantitative Trait Loci (eQTL) Provides Insight into the Regulatory Architecture of Gene Expression Variation in Liver." *Scientific Reports* 8 (1): 5865.

Wang, Jianguo, Juhong Liu, and Zilong Guo. 2023. "Natural UORF Variation in Plants." *Trends in Plant Science*, August. <https://doi.org/10.1016/j.tplants.2023.07.005>.

Yofe, Ido, Zohar Zafrir, Rachel Blau, Maya Schuldiner, Tamir Tuller, Ehud Shapiro, and Tuval Ben-Yehzekel. 2014. "Accurate, Model-Based Tuning of Synthetic Gene Expression Using Introns in *S. Cerevisiae*." *PLoS Genetics* 10 (6): e1004407.

Reviewer 3 Attachment:

Figure 2

Characterisation of independent signal eQTL variants based on their genomic localisation. The distance to the nearest transcription start site (TSS) is plotted against the $-\log_{10} P$ Values of the most significant variant at each eQTL gene, including secondary signals (independent hits). Negative/positive distances denote that the variant is located upstream/downstream of the TSS with regard to the direction of transcription.

Author Rebuttal to Initial comments

Reviewer #1:

Remarks to the Author:

I found this to be a fascinating manuscript and enjoyed reading it. The authors wove together a compelling set of observations to demonstrate the role of sequences downstream of the TSS in influencing gene transcription and in particular highlight the role of a specific motif. The combination of using natural variation, large scale screens and deletion analyses are very useful in pulling together the observations. This work highlights unique aspects of cis-regulatory control of gene expression in plants and will open new avenues for research into cis-regulatory variation. I did not have major issues with the findings that were presented but found a number of details were hard to follow or missing. These are detailed below in the minor comments but the major ones for my understanding were as follows:

More clear description of the MPRA assay to document whether orientation of the synthesized oligos was controlled or not. I am quite curious if the motif has any orientation effects when present downstream of the sequence or not.

Fragments were inserted in their original orientation. We have now clarified this in the text. Because the GATC-motif is a palindrome, it presents the same forward and reverse: (CT)(ACG)GATC(CGT)(AG). Therefore, the core motif is not dependent on orientation.

Clearer description of the "GATC motif". Is this a 4-mer, a 6-mer or an 8-mer. It became difficult to follow this.

To be clearer, we have made explicit that whenever we mention the 'GATC-motif', we are specifically referring to the 8-bp (CT)(ACG)GATC(CGT)(AG) motif.

It is likely beyond the scope of this paper but I was left quite curious about some comparative genomics aspects of this work. Do orthologs in different species show any level of conservation for 5' UTR length or for GATC motif presence?

We found only one published instance of a direct comparison of 5' UTR lengths between orthologs in any eukaryotic species. The study examined 2,516 one-to-one orthologs between *S. cerevisiae* and *C. albicans* and observed a modest Spearman correlation of 0.2 in 5' UTR lengths (P value < 2.20×10^{-16}) (Lin and Li 2012).

To expand our understanding of the conservation of these features, we directly compared the 5' UTR lengths and presence of GATC motifs between orthologs of two species used in our MPRA analysis: *A. thaliana* and tomato. We identified 6,883 one-to-one orthologs with OrthoFinder (Emms and Kelly 2019). For 4,211 genes, a TSS had been identified in both species. The Spearman correlation between the TSS-to-ATG distance (including 5' UTR and introns in this region) was 0.42, higher than between *S. cerevisiae* and *C. albicans* (Lin and Li 2012). The higher correlation might reflect higher selective constraint, but it can also be due to the lower evolutionary distance (120 Mya vs. 235 Mya (Kumar et al. 2022)) or technical differences in the data quality. The presence of the GATC motif shows significant conservation between orthologs (p-value < 10^{-49}). This analysis is now presented in the newly added Fig. S26.

The frequency of the GATC motif in 5' UTR, coding sequences and introns was not all that clear. I am curious if the analyses in later portions of the paper would find similar effects for GATC motifs in all these contexts across species or not.

We thank the reviewer for this suggestion. The probability of finding the GATC motif in the 500 bp downstream of the TSS for introns, coding sequences and UTRs is the same as their background distribution (newly added Fig. S23B). A GATC motif in the UTR or intron is associated with higher gene expression than a GATC motif in the coding region (newly added Fig. S27). We speculate that this is either due to other sequence constraints around the motif or to a more open chromatin structure, which appears to be related to where the coding region starts relative to the TSS, as we show in Fig. S4, and also overall sequence characteristics distinguishing introns and exons (Schwartz and Ast 2010; Schwartz, Meshorer, and Ast 2009; Chodavarapu et al. 2010; Jabre et al. 2021). Although the effect is weaker in the coding region only, it is still statistically significant (Fig. S27).

The statistical association between GATC motif and gene expression also increases in all vascular plant species shown in Fig. 5 when only motifs located in introns or UTR regions are considered (newly added Fig. S32). This does not apply to the bryophytes tested, further suggesting that the mechanism driving transcription through the GATC motif is either much weaker or functions differently in bryophytes.

Our newly added experiments measuring genome-wide gene expression in response to transient overexpression of GATA TFs shed further light on the importance of genomic context for GATC motif activity. Indeed, we observe an increase in the expression of genes with GATC motifs in the three genomic contexts (newly added Fig. S25A).

Specific comments:

Lines 37-39; I know this analysis is using previously published datasets but it could be useful to mention the tissue type being used.

Added.

Lines 48-55: the authors do several tests to examine the potential bias in the locations of cis eQTL to assess the potential that these cis eQTL may influence transcript stability. It might be helpful to perform a test of the enrichment (or depletion) of cis eQTL within specific regions (ie 5' UTR, coding exon, intron and 3'UTR). By assessing the frequency of cis eQTL in these regions relative to the total length accounted for by these regions it should be possible to directly examine the contributions of variation in different types of regions. I am not saying this should replace the current analyses but it would be helpful for the reader to see the actual distribution.

We now include an analysis of the total signal of eQTLs identified within each transcript feature (of those detected within transcripts) against the total feature length within the corresponding genes (newly added Fig. S2D-E). Our analysis confirms that the overall presence of eQTL variants within introns is consistent with background expectations. There is a pronounced enrichment of exonic eQTL variants in the 5' UTR compared to coding sequences. This observation is consistent with the observed stronger eQTL signal near the TSS, which is more frequently associated with the 5' UTR than the CDS, as shown in newly added Fig. S2B.

Lines 54-59: I thought this was a very interesting finding. The propensity for having CREs downstream of the TSS seems highly influenced by the length of the 5' UTR. Just a quick clarification could be useful - how well characterized are the actual TSS sequences in Arabidopsis and do these exhibit any natural variation among accessions? This might be helpful to non-Arabidopsis focused researchers.

The positions of TSSs in the Arabidopsis reference are well established and have been progressively refined through successive updates of the reference genome annotation, as demonstrated by studies

assessing its accuracy (e.g. (Schon et al. 2018)). The fidelity of the TSS definition is confirmed by the analysis of genome-wide measurements in relation to the TSS. For example, the plot of DAP-Seq signals (Fig. 1C) shows a pronounced dip at the TSS location. Similarly, the examination of the genome-wide nucleotide composition adjacent to the TSS at bp resolution highlights the presence of the transcription initiator sequence (see Fig. 3A-B in (Hetzel et al. 2016)).

To the best of our knowledge, there is no literature on the variation in TSS position between different *Arabidopsis* accessions. Currently, our group is collaborating with Prof. Kosuke Hanada from the Kyushu Institute of Technology to investigate TSS variability among natural *Arabidopsis* accessions. Using CAGE-Seq (Cap Analysis of Gene Expression followed by Sequencing), our preliminary results indicate a high degree of conservation in TSS positions across accessions. Using the TSS definition described in (Ushijima et al. 2017), we found that of 15,700 genes with identifiable TSSs, 14,600 had a single, dominant TSS that was consistent across all 17 accessions studied. However, because these are preliminary results that will require further refinement, we consider this line of investigation to be beyond the focus of the current manuscript and expect to present these results in a separate publication.

One other question - given the observation that genes substantially vary for the position of CREs relative to the TSS given and that some TFs prefer to bind upstream of the TSS while others prefer downstream binding this suggests the UTR length may be a selected factor for the potential for downstream TFs to act on a gene. Is there much evidence that 5' UTR length has any conservation between plant species?

See our answer above and our newly added comparison between tomato and *A. thaliana* UTR length and GATC motif counts (Fig. S26). In brief, there is some conservation, but whether this reflects selective constraints is unclear.

Line 73: What happens to the plots in Figure S4 (or in 1B) if you center the plot on the ATG (rather than the TSS)?

We have created new figures for this purpose: the newly added Fig. S2G corresponds to Fig. 1B, and newly added Figs. S4C-D correspond to Figs. S4A-B, each showing the data centered around the ATG.

This re-centering reveals a pronounced enrichment of eQTL variants upstream of the ATG, particularly evident for the gene group with the broadest TSS-to-ATG range, where we observe an expanded eQTL signal. Notably, when examining H3.1 and H3.3 enrichment, aligning the data to the ATG results in a discernible concordance between the gene groups. These observations support the distinct biological function of the TSS-to-ATG region.

Lines 92-92: I could not tell from the description of the assay in the text whether the fragments were directionally inserted or whether the fragments could be in the sense and antisense orientation. If the latter is true did you assess whether orientation had equivalent effects for upstream and downstream sequences?

Fragments were inserted in their original orientation. We have now clarified this in the text.

Line 157-161: Do the sequences identified in this analysis match any of the predicted motifs for the TFs that have biased DAP-seq peaks for downstream regions (ie Figure 1D)?

If we take as an example the 35S promoter library in *Arabidopsis*, we find that the 10 most significant

6-mers identified all contain the GATC sequence, aligning with the DNA motif for GATA TFs as revealed in DAP-Seq data. This observation is in agreement with the preferential binding of the GATA TF family (“C2C2-GATA”) to regions downstream of the TSS, as depicted in Fig. 1D. Furthermore, among the subsequent 10 most significant 6-mers (each with a p-value < 10^{-35}), 7 do not include the sequence 'GATC'. These sequences either exhibit repetitive 'a' bases (e.g., 'AAAAAA', 'AAAATT') or the sequence 'ATCT'/ATC', a characteristic motif of the “G2-Like” TF family, also shown to bind downstream of the TSS in our analysis of DAP-Seq data. Additionally, several 6-mers containing 'CCG' or 'GGC' were notably significant (e.g., 'ACCGGC', 'AGTGGC', 'CCGGTA', 'ACCGCC', with p-values < 10^{-15}), resembling motifs recognized by the “AP2-EREBP” TF family, which similarly binds downstream of the TSS in our analysis of DAP-Seq data.

We have chosen not to explore these sequence motifs within the scope of this manuscript, because robust validation of these motifs' functional significance would require a comprehensive series of follow-up experiments that are beyond the scope of the present study.

Line 163 - on. I got a bit confused about how the authors were using the term “GATC motif”. Does this truly refer to the 4-mer GATC or are they using this for the 8-mer defined in line 159. This was not clear through the analyses and figures as the current usage seemed to imply a 4-mer but I wasn't sure that was how this was being used.

To be clearer, we have made explicit that whenever we mention the 'GATC-motif', we are specifically referring to the 8-bp (CT)(ACG)GATC(CGT)(AG) motif.

Line 204-207 - when you added these GATC motifs to these upstream and downstream fragments were they only tested in the downstream location or did this get test in both upstream and downstream locations?

The fragments in question were exclusively positioned downstream of the TSS. This detail is mentioned two paragraphs earlier, within the context of the follow-up experiment's description.

Lines 238-241 - why test 6-mers rather than the defined 8-bp GATC motif defined earlier?

The decision to utilize 6-bp sequences was driven by the aim to enhance the statistical robustness for each sequence occurrence. In our analyses, *k*-mers that include the GATC sequence emerged as the most significant, even when we explored 5-mers, 7-mers, and 8-mers, although these data are not presented in the manuscript.

Line 251-284: What is the overall expression of GATA TF family members in these tissues? Is there any correlation between overall GATA expression and the tissues with more activity for the GATC motif?

We appreciate the suggestion for further analysis, which has now been added as Fig. S31. The analysis reveals a strong Pearson's correlation ($r = 0.8$) between the average expression levels of all 30 GATA TFs family members and the GATC-motif effect across tissues. We further show that this correlation is not due to the GATA TFs themselves being part of the GATC regulation, as the correlation between GATA TF expression and GATC motif effect remains significant even when excluding the 5 TFs with a GATC motif within 500 bp downstream of their TSS. Additionally, we conducted separate analyses for each of the four GATA TF subfamilies, following the classifications by (Schwechheimer, Schröder, and Blaby-Haas 2022) and (Reyes, Muro-Pastor, and Florencio 2004). Our findings indicate that subfamily A, with 14 members, exhibits the most pronounced correlation. Notably, DAP-Seq data support this observation, predicting

binding of subfamily A members to the GATC motif downstream of the TSS, as depicted in the newly added Fig. S22.

Reviewer #2:

Remarks to the Author:

This manuscript is a tour de force study to investigate the important question of whether plant enhancers work similarly as animal enhancers, independently of orientation and position. The study used massively parallel reporter assays (MPRA) to demonstrate the regulatory role of sequences downstream of the transcription start site (TSS) in plant activation of transcription. The study identifies a specific motif (GATC) that functions in a dose-dependent fashion and appears to be associated with activation of transcription driven by downstream sequences and which doesn't function upstream of the TSS. MPRA studies were complemented with the re-analysis of published results to demonstrate that the effects of the GATC motif is not associated with mRNA stability, and that it is conserved across species. The hypothesis that the GATC motif acts as a regulatory module, operating as a rheostat in tuning gene expression between cell types, is attractive but largely derived from correlations (nevertheless, worth mentioning it).

1. Regarding the impact on mRNA stability: We now demonstrate directly that the effect on transcription when positioning sequences downstream in the MPRA setup is primarily due to mRNA synthesis (Fig. 2E and S10).
2. Regarding the GATC hypothesis: Our revised manuscript now provides two additional key pieces of evidence in support of our hypothesis. First, we provide new data demonstrating the upregulation of genes containing the GATC motif in response to overexpression of GATA TFs (detailed below in response to point 8), providing a mechanistic explanation for the effects of GATC motifs. Secondly, we present findings that the influence of the GATC motif—conceptualized as the "rheostat knob" in our analogy—is closely associated with the mean expression levels of GATA TFs, with particular emphasis on those belonging to the A/I subfamily (Reyes, Muro-Pastor, and Florencio 2004; Schwechheimer, Schröder, and Blaby-Haas 2022) (newly added Fig. S31). These additions to our manuscript lend further support to the notion of the GATC motif as a dynamic regulator of gene expression, aligning with the rheostat model.

Comments

1. The title sounds good as it suggests a generality in the findings, but the reality is that the study only focuses on putative enhancers containing a "GATC" motif and presumably recognized by GATA TFs. A more honest title (even if less spicy) would provide a better service to the community.

We agree that our claim of widespread position-dependent enhancers in plants cannot be based on the GATC motif effect alone. However, the conclusions about the GATC motif (with new experiments shown to be mediated by transcription rate, not transcript stability) are not the primary basis for the paper's title. Our results can be summarized as follows:

1. Regulatory elements inside transcribed regions play an important role in transcriptional regulation in flowering plants.
2. Unlike in animals, a large fraction of plant enhancer activity is position-dependent.
3. Finally, we discover and characterize a single motif, downstream of the TSS, with strong effects on thousands of genes.

Our claim of "widespread position-dependent enhancers" is first and foremost based on the first two findings. In animals, it does not matter on which side of the TSS an enhancer sequence is located because enhancers are position-independent. Plants are clearly different. Our most compelling evidence for this is shown in Figure 2D, which should be compared with the original figure from the seminal STARR-Seq paper (Arnold et al. 2013), a method that is widely accepted as the gold standard for genome-wide identification of enhancer elements. In animals, the effects of the same enhancer positioned upstream and downstream of a TSS are very highly correlated ($r^2 = 0.69$), whereas in plants this correlation is entirely absent ($r^2 = 0.02$):

It is important to note that from the 12,000 tested fragments shown in Figure 2D, only 1,300 have a GATC motif. Moreover, ~4,000 of the tested fragments originate from upstream of the TSSs of highly expressed genes in Arabidopsis. While these fragments do enhance expression level when positioned upstream of the TSS in the MPRA setup (Fig. 2F), they do not have this general tendency when positioned downstream, and, even more importantly, there is no correlation between their effects when positioned on either side of the TSS. Therefore, while placement does not matter at all for the vast majority of animal enhancers, plant enhancers normally located upstream of the TSS cannot simply be "transplanted" to a downstream position (or vice versa).

It is similarly important to note that: (1) The main effects on gene expression we measure when we position the sequence downstream of the TSS in the MPRA are due to mRNA synthesis (newly added Fig. 2E). (2) The observation that relevant TFs tend to bind to one side of the TSS (Fig. 1D & S5) further supports the asymmetric nature of regulatory sequences.

Finally, a previous MPRA study used genomic fragments in Arabidopsis and rice (Sun et al. 2019; Tan et al. 2023) in the 3' UTR, as in the original STARR-Seq configuration. That study concluded that enhancers are highly enriched in the gene body and especially at the 5' end of it, but depleted upstream of the TSS. This is in contrast to the same experiment in Drosophila or human cells (Arnold et al. 2013; Liu et al. 2017), where enhancers are enriched upstream of the TSS. Below is the original Fig. 6F from Tan et al. (2023) showing this comparison:

Patterns of enhancer distribution relative to gene bodies in four species.

This of course does not mean that regions upstream of the TSS cannot have strong transcriptional enhancing effects in plants, as has been shown in many studies focusing on single-gene transgenes, as well as in more systematic analyses (Jores et al. 2021). Our experimental system is able to systematically show that there is a difference in the logic of how transcription-enhancing sequences function on either side of the TSS, and we thereby explain the seemingly contradictory results of the previous studies. We now modify the discussion to better emphasize these points.

2. How much of the effect observed regarding the conclusion of position with regards to the TSS of the sequences tested could be an effect of interactions between the putative enhancers and core promoter elements? Perhaps the idea that the core promoter is “universal and largely exchangeable between genes” requires revisiting? For example, in Fig. 2C, how is the comparison of each enhancer with the two minimal promoters? For example, does 35S enhancer work better with a minimal 35S?

The 35S enhancer consistently exerts an enhancing effect on gene expression when positioned upstream of the TSS in all four species examined, using both core promoters. Notably, this relative enhancement effect is more pronounced with the 35S minimal promoter than with the *TRP1* promoter in Arabidopsis and tomato, whereas in maize and *N. benthamiana*, the enhancing effect is similar with both core promoters, as shown in the newly added Fig. S9A.

Furthermore, our analyses, supported by the correlation matrix shown in Fig. 2D and expanded in the newly added Figs. S9B-C, indicates that the effect of the 12,000 tested fragments is generally consistent between the 35S and *TRP1* core promoters. This suggests that while there are interactions between the core promoter and the enhancer fragments, they tend to produce comparable results.

Finally, we consistently compare the effects of inserted fragments relative to the TSS within the context of the same core promoter. Thus, it is difficult to see how the observed lack of correlation between insertion positions relative to the TSS could be attributed to specific enhancer-promoter interactions.

3. The value of conducting the MPRA experiments in four species is questionable – it certainly

complicates a straight-forward story and does not really add conceptually much. Moreover, it is concerning how the results should be interpreted in the light of what was found in the study by Jores et al (2021 Nature Plants) where they clearly show that core promoters behave differently in monocots and dicots, depending on the origin.

Our goal in conducting the MPRA experiment in different systems was to validate the generality of our results. When we first saw the difference in gene expression levels when positioning potential enhancers on either side of the TSS, we were worried that it might be an idiosyncrasy of the system used. Our first sanity check was to use different transformation approaches - protoplast transformation in *Arabidopsis* vs. leaf infiltration in *N. benthamiana*. Then, to verify the generality of the approach, we tested a monocot system (maize) in addition to the two dicots. We then added yet another dicot (tomato) using the protoplast system to further increase the generality of our results (and also because the *Arabidopsis* protoplast transformation had been adapted from a tomato protocol, reducing irrelevant technical differences (Ben-Tov et al. 2023)). For the same reasons we used two different core promoters. It is important to note that the two main results were reproducible in all four species and with both core promoters: (1) the lack of correlation when positioning potential enhancers on either side of the TSS, and (2) the strong association of gene expression with the GATC motif.

We believe that the results of Jores et al. (2021) are in accordance with what we see: first, as can be seen in the hierarchical clustering in Fig. 2D, in three of the four sub clusters (pTRP1 both up- and downstream, and 35S downstream), the monocot maize is separated from the three dicot species. Moreover, in Fig. 2F it can be seen that the effect of most fragments is negative when positioned upstream of the TSS in maize, in accordance with results from Jores et al. (2021). Nevertheless, the effect of the GATC motif is strong enough to be detectable in maize.

4. How many of the sequences tested in the Jores et al manuscript contained GATC motifs? It would be interesting to see if the hypothesis proposed here are supported by those experiments (expectation is that GATC should be significantly less represented in active core promoters).

In the study by Jores et al. (2021), 5.02% of the 170 bp sequences from *Arabidopsis* protein promoters contained the GATC motif. This is comparable to our own libraries, where 4.97% of the fragments originating upstream of the TSS of genes contained the GATC motif. Conversely, 13.73% of the fragments originating downstream of the TSS had the GATC motif.

Our re-analysis of the data from Jores et al. (2021) suggests that the presence of a GATC motif in regions upstream of the TSS is not associated with increased gene expression, as expected. We have illustrated this conclusion by splitting the Jores et al. (2021) data based on the presence or absence of the GATC motif:

Reviewer Figure R1: This figure shows the log2 gene expression data from MPRA experiments versus constructs containing only 35S minimal promoters, as reported by Jores et al. 2021 (Supplemental Table S2). The sequences shown are derived from *Arabidopsis* protein genes and are categorized based on the absence or presence of the GATC motif. Expression levels are shown under three specified conditions, both without (top row) and with the addition of the 35S enhancer (bottom row).

5. (Lines 148 – 155 and elsewhere) The idea that binding of a TF to a 6-mer sequence is sufficient to drive gene expression, particularly in a system in which the TF is not over-expressed is, in my opinion, a bit out of date. While true that 6 bp are often sufficient *in vitro* for TF binding, there is an extensive literature that demonstrates that *in vivo*, TFs recognize much longer sequences. Surely more activity is possible if repeats of the sequence are included, but that is not a very common situation for most TFs. Perhaps of even greater concern is that no TF works on its own with large complexes of TFs forming what is believed to be an active TF complex. I recommend some re-writing to make it clear that the situation is not as simple as written.

We fully agree with the reviewer's point, which is why the correlation between the relatively short GATC motif and gene expression in our results was particularly surprising to us. We have revised the section as suggested.

6. In Fig. S5, the bimodal distribution can easily be explained by nucleotide distribution bias around the TSS – indeed, *Inr* elements usually have a pretty conserved motif. Was this considered? And if it is, how does the result/interpretation change?

The *Inr* sequence does indeed have a strong effect on the nucleotide frequency around the TSS, as can be clearly seen in this figure from (Hetzel et al. 2016):

Fig. 3A Hetzel et al.

However, the effect is limited to about 5 bp on either side of the TSS. Therefore, the effect seen in Fig. S5, which is on the scale of 500 - 1000 bp, cannot be explained by this sequence bias.

7. Since the authors analyzed the published Arabidopsis DAP-seq results – when they identified GATC motifs downstream of the TSS, were they preferentially recognized methylated or unmethylated (irrespective of the TF)? Clearly GATC methylation would not fall into either GC or GNC methylation, might the A be methylated? And if so, would that be more suggestive of a role in transcription or post-transcription?

Thank you for suggesting this analysis. First, as can be seen by comparing Fig. S5A to Fig. S5B, DNA methylation mostly affects transcription factors binding downstream of the TSS, as the downstream binding peak is decreased relative to the upstream one in the samples where DNA was methylated. This difference is also consistent with the enrichment of methylation downstream of the TSS (e.g., (Zilberman et al. 2007)).

To specifically examine the differences in TF binding to the GATC motif, we re-analyzed the DAP-Seq data of the GATA TFs predicted to bind the GATC motif downstream of the TSS (newly added Supplementary Fig. S22). Of the 14 GATA TFs analyzed by DAP-Seq in (O'Malley et al. 2016), only four were also analyzed on non-methylated DNA (GATA1, GATA12, GATA19/HANL2, and GATA24/ZML1). Only three of these GATAs share a motif with the GATC sequence, of which one had more binding to methylated DNA (GATA12), one to non-methylated DNA (GATA1), and one had the same binding regardless of DNA (GATA19/HANL2). Therefore, the DAP-Seq data do not show a specific tendency to bind methylated GATC motifs.

Our newly added experimental data now demonstrate more directly that the action of the GATC motif is through transcription rate and not through post-transcriptional effects on mRNA stability by directly measuring mRNA synthesis (Fig. 2E) and showing that transient overexpression of GATA factors increases the expression of genes with a GATC motif (point 8 below).

8. Whether GATA TFs are responsible for recognizing the GATC motif described here is in many ways irrelevant. It is correct that some GATA TFs have been shown to bind GATC better than GATA, but the authors don't focus specifically on this aspect. Hence, I would recommend that the discussion of whether GATA TFs are responsible for binding the GATC motif should be eliminated (together with Note S1). As a word of caution, Zn-finger proteins (such as GATA TFs) are very abundant in plants and they often encode nucleic acid binding proteins (beyond TFs). GATC happens to be palindromic, also raising the question of whether it could be the core of a larger motif recognized by dimeric TFs.

We agree with the reviewer that the main claims of the manuscript are independent of whether GATA TFs are responsible for the recognition of the reported GATC motif. We decided to propose this based on the compelling DAP-Seq data indicating binding of GATA TFs to the GATC motif downstream of the TSS (Fig. S22), suggesting a mechanism for the strong effect of the GATC motif we observed.

Rather than omit this discussion, we decided during revision to further test our hypothesis experimentally. To this end, we transiently overexpressed three GATA TFs (GATA1, GATA4, and GATA6) and a GFP as a control in Arabidopsis leaf protoplasts for 8 hours followed by RNA-Seq to identify their targets. These specific TFs were selected based on their binding affinity to the GATC motif downstream of the TSS in the DAP-Seq data and their expression in leaf tissue, which was our source tissue of protoplasts. The eight-hour duration was chosen to focus on direct target effects, supported by a calibration experiment tracking fluorescence in GFP-expressing protoplasts, shown below:

Reviewer Figure R2: Percentage of protoplasts with nuclear GFP along an eight-hour time course after protoplast transformation. Arabidopsis leaf protoplasts were transformed with a plasmid with a double-35S promoter driving a GFP with a nuclear localization sequence (NLS), the same plasmid used as a control in the overexpression experiment.

The genes with increased expression in response to overexpression of each of the three GATAs are highly enriched in the GATC motif downstream of the TSS (newly added Fig. S24). Moreover, the increase in expression was proportional to the occurrence of the motif downstream of the TSS (newly added Fig. 3H). These results, together with a newly added re-analysis of published datasets correlating GATA TF expression with GATC motif effect in different tissues (newly added Fig. S31), support the role of GATA TFs as the upstream regulators of the GATC motif.

9. The title of Fig. 2 should read: "Position-dependent enhancers reside inside transcribe regions". What are the y-axis units for Figs. 2B and 2D (right inset)?

We appreciate your pointing out the omission of the units for the y-axis in Figures 2B and 2D (right inset). These units are $\log_2(\text{expression change})$, we have now added this information to the figure caption.

10. I found the activity of the known enhancers shown in Fig. 2C to be pretty modest. Is this because the activity of the minimal promoters was already high? If they were, it might be worth mentioning and comparing the two "minimal" promoters used (I understand that the TRP1 is not formally a minimal promoter). Unfortunately, only ratios are shown.

The effect we observed is relative to minimal promoters that are already active on their own; the observed level of increase is consistent with previous reports (Jores et al. 2020). In our current system the libraries with the TRP1 promoter or the 35S minimal promoter cannot be compared, as they were prepared using different 5' end primers for enrichment of the MPRA driven transcripts.

To compare the activity of the two "minimal" promoters, we now used an aliquot of RNA from our original 4 replicates of MPRA in Arabidopsis experiments. This RNA was used for RNA-Seq using the

SMART-Seq3 protocol (Hagemann-Jensen et al. 2020) and sequenced to deep coverage. In these libraries, we can identify the expression of all genes without primers specific to the MPRA setup, allowing us to compare the pTRP1- to the 35S-based libraries. Using this data, we observed that the expression of the minimal 35S promoter is about 4 times stronger than the "minimal" *TRP1* promoter. We have now described this in the Methods section.

11. In Fig. 3A, what is the correspondence of sequences when comparing the two core promoters? In other words, are the same sequences providing strong activation regardless of the core promoter used?

Comparing the results with two core promoters, there was a high degree of agreement in the sequences that were highly associated with transcription activity. To clearly illustrate this, we have included a detailed comparison in the newly added Fig. S13.

12. The authors should reexamine whether the sequences that they are analyzing (containing one or more GATC motifs) truly fit the definition of enhancers, and whether they have enough information to call them enhancers. While it is clear that plant and animal enhancers are probably different (several recent papers have discussed this, one of several that come to my mind includes <https://pubmed.ncbi.nlm.nih.gov/34918159/>, but there are certainly others).

In animals, enhancers are today usually considered clusters of multiple binding sites for transcription factors, which in turn often act in a cooperative manner. In our case, we do not know whether or not the transcription factors underlying the observed effects operate cooperatively. Nevertheless, considering the original definition of enhancers as sequences that increase activity of a core promoter (Banerji, Rusconi, and Schaffner 1981), we feel that the use of enhancers as a very general term is appropriate.

It is important to note that in our revised manuscript we demonstrate that the observed effects in the MPRA are mediated by transcription rather than mRNA stability. This is achieved by directly measuring the rate of mRNA synthesis in our MPRA (setup explained in Fig. S10A, and results in Fig. 2E & S10B). In combination with our new GATA overexpression experiments, we show that the downstream sequences are consistent with enhancer activity.

Reviewer #3:

Remarks to the Author:

The authors are asking an interesting question: What is the role of eQTL downstream of the transcription start site (TSS) in plants and how do the underlying sequences affect RNA levels? Unfortunately, they use a previously developed MPRA design that is ill suited to answer this question. The MPRA tests sequences for activity in only two ways: either as potential enhancers upstream of the TSS or inserted into a reporter gene intron at a fixed distance downstream of the TSS. This design fails to capture the contextual origin of the tested sequences (i.e. 5'UTR, intron in 5'UTR, exon, intron, etc.), ignoring the wealth of prior knowledge on how 5'UTRs, intron, introns in 5'UTRs and other transcribed sequences can affect RNA levels (Chung et al. 2006; Alan B. Rose and Beliakoff 2000; Alan B. Rose 2004; A. B. Rose and Last 1997; Laxa et al. 2016; Laxa 2016; Wang, Liu, and Guo 2023; Callis, Fromm, and Walbot 1987). Given this assay design, it is not surprising that the authors identify a known sequence motif that affects RNA levels when residing in introns. Although the authors present a large amount of carefully generated data from different expression systems, their strong claim of widespread position-dependent plant enhancers is not well supported by the presented evidence and the expression-enhancing motif GATC that is discussed as a major finding is not novel.

Our paper is not trying to answer a question about downstream eQTL—the eQTL study was only a tool to answer a more general question. To clarify, our study attempts to identify unique properties of transcriptional regulatory sequences in plants compared to animals. In multicellular organisms, the number of transcriptional states, which is larger than the number of cell types, greatly exceeds the number of TFs. With the advent of multicellularity, new mechanisms had to evolve that allowed for the generation of transcriptional complexity. Plants and animals became multicellular independently, and there is no reason to believe that evolution took the same path twice. Indeed, it has been shown that enhancers are neither position-independent in fungi, which are closer to animals than plants are (Struhl 1995; Hahn and Young 2011), nor in an outgroup of multicellular animals (Sebé-Pedrós et al. 2016), and therefore there is no evolutionary continuum for the mode of enhancer action between animals and plants. We started with an unbiased approach, and the identification of eQTL variants with the 1001 Genomes resource was only a "screen" to guide us to regulatory regions.

For the question we sought to answer, we believe our MPRA setup was the most comprehensive approach. By systematically testing the same sequence in both positions, we can determine whether regulatory sequences function in the same way or not. Using the intron as a landing site is the safest option because it does not alter the mature mRNA. Others have previously used the 3' UTR as a landing pad in STARR-Seq because this is easier experimentally, since it does not require the addition of a barcode to the mature mRNA. We chose an experimentally more challenging approach, because we wanted to make sure that the mature mRNA is not affected by stability differences that could be caused by different potential enhancers. We would also like to point out that our evidence goes well beyond the state of the art in the field, by interrogating multiple systems and combining insights from both experimental and genomic data. In the comments below, the reviewer raises several important concerns regarding the use of an intron as a landing site: (1) Is splicing affected? (2) Is mRNA stability affected? As we show below, our newly added experimental data show that splicing is not affected, and that the main effect of our regulatory sequences in the intron is on mRNA synthesis and not mRNA stability.

The reviewer further argues that our "*strong claim of widespread position-dependent plant enhancers is not well supported by the presented evidence*". We disagree because, as we explain in the next point below, we show that placing enhancers on different sides of the TSS gives completely different results. This is in stark contrast to similar experiments in animals.

Finally, we do not claim that the literature does not contain any evidence in support of our claims of (1) the presence of regulatory sequences in the region downstream of the TSS, (2) the position-dependent regulatory sequence, or (3) the effect of the GATC motif on transcription. We cite prior work throughout our manuscript. Indeed, given the generality of our results, it would be surprising if there were no prior results in support of our findings. However, the generality and the scope of our findings, including the very different modes of enhancer action in plants vs. animals, are far from being commonly accepted dogma in the field. Indeed, we believe that our results finally provide a unifying explanation for previous, often scattered and disparate observations, in particular the recurring identification of intron-mediated-enhancement in plants.

I am most puzzled by the insistence of the authors that they have identified a fundamental regulatory phenomenon specific to plants. eQTL in human also reside downstream of the TSS in about equal (or higher!) numbers – see figure reproduced from a published study (Strunz et al. 2018; Figure 2) in the review attachment. Although the scale differs on the x-axis compared to this manuscript's Figure 1A, the majority of signal in human is visible close to the TSS and gene size differs profoundly between human and Arabidopsis. Most assuredly, if the authors had conducted this experiment with human sequences, they would have found that these sequences also affect expression differently depending upon their position upstream or downstream of the TSS. While it is meaningful to consider regulatory features that

fundamentally differ between plants and animals – e.g., the presence of CTCF-dependent topological associated domains in animals (Schmitz, Grotewold, and Stam 2022) – the phenomenon investigated here is shared.

We do not claim that eQTLs in humans would not be downstream of the TSS. Moreover it is not clear to us why the reviewer asserts that “if the authors had conducted this experiment with human sequences, they would have found that these sequences also affect expression differently depending upon their position”. **As we explain below, experiments in animals obtained opposite results, as expected from the position independence of animal enhancers.**

We were inspired to use eQTLs for an exploration of transcriptional regulatory sequences by research in humans (Veyrieras et al. 2008). The study by Veyrieras et al. (2008), which we cite, reported an eQTL distribution around 100 Kb on both sides of the TSS, as seen in the later studies as the one mentioned by the reviewer (Strunz et al. 2018). This distribution of eQTLs is consistent with the propensity of enhancers to drive transcription, as recently seen in mouse cells (Zuin et al. 2022):

Fig. 1H from Zuin et al. - Enhancer effect at different positions relative to TSS

Thus, eQTL distribution follows the regions potentially affected by enhancers. By comparison, a screen that randomly positioned a strong enhancer across the genome in *Arabidopsis* detected effects only in nearby regions (up to 3.6 Kb) (Weigel et al. 2000), despite its genome size being comparable to that of the fruit fly, which relies largely on long-range enhancers (Small and Arnosti 2020). This highlights the fundamental difference between the species.

Another clear difference, which we mention in the manuscript, is that when the authors studied eQTL distribution in humans, they found an enrichment of eQTL variants in exons vs. introns, which they associated with an effect related to mRNA stability (Veyrieras et al. 2008). In our analysis of eQTLs in *Arabidopsis*, we found no such enrichment (also with the new analysis now shown in Fig. S2D-E).

Most importantly, similar experiments in humans or *Drosophila* have produced very similar results when positioning sequences on either side of the TSS, in contrast to our findings. This was done in the original STARR-Seq paper (Arnold et al. 2013), with more sequences tested for S2 *Drosophila* cells, but also for human cells. We show here the three plots for comparison:

Therefore, it largely does not matter where animal enhancers are placed, but plant enhancers from upstream of the TSS cannot simply be "transplanted" to a downstream position (or vice versa).

Another example of how different results from animals and plants look when using the same experimental setup can be found in the use of STARR-Seq in plant vs. animal systems. Reports that examined the activity of genomic sequences in Arabidopsis and rice (Sun et al. 2019; Tan et al. 2023) by positioning them in the 3' UTR (using the original STARR-Seq configuration), identified enhancers that are highly enriched in the gene body and especially at its 5' end (in agreement with our positional dependency results), while they are depleted upstream of the TSS. This is in contrast to the results using a similar approach in Drosophila or human cells (Arnold et al. 2013; Liu et al. 2017), where enhancers are enriched upstream of the TSS. Below is the original Fig. 6F from Tan et al. (2023) for comparison:

It is important to emphasize that our results support a more complex regulatory logic than enhancers only working on one side of the TSS. Rather, some enhancer sequences are more active when positioned upstream than downstream of the TSS, some are more active when positioned downstream, and some sequences are equally active (or inactive) on either side of the TSS. Therefore, the enhancers are read differently in these two genomic environments (up- and downstream of the TSS), which we believe is consistent with the results by (Meng et al. 2021) that the reviewer refers to below (as explained in the

answer to point 2).

The authors' claim that their study identifies widespread position-dependent enhancers in plants is particularly irksome. Enhancers are defined as position-independent sequence elements that enhance transcription of their target gene in a condition- or tissue-specific manner, mediated by transcription factor binding at both enhancer and promoter, often involving looping that allows for long-distance interactions (Schmitz, Grotewold, and Stam 2022). Are these position-dependent elements bound by TFs in the respective genomes and is this binding relevant for their function? The DAP-seq data (O'Malley et al. 2016) presented in Figure 1 are an aggregate plot for 529 Arabidopsis TFs (in C) and DAP-seq peak enrichments consolidated for TF families (in D) rather than direct evidence of functionally relevant TF binding at GATC motifs. Likewise, element interactions with target gene promoters were not tested or even inferred with existing 3D data.

Enhancers were initially defined as sequence elements that enhance transcription from a core promoter in a position-independent manner (Banerji, Rusconi, and Schaffner 1981), and today it is often implied that they included multiple binding sites for TFs. It is in addition true that condition specificity, tissue specificity, looping, etc. are common features of animal enhancers, but they are not part of the definition of enhancers. In animals, a general "rule" appears to be that sequences that enhance transcription from the core promoter do so in a position-independent manner; **what we report here is categorically different from what is known in animals**. Therefore, we define them as position-dependent enhancers, as opposed to the position-independent enhancers that constitute the vast majority of enhancers in animal genomes.

Regarding TF binding, the DAP-Seq data for many individual TFs indicate binding downstream of the TSS, as we show in Fig. S5C and the newly added Fig. S22. We also directly demonstrate, using transient overexpression of GATA TFs and RNA-seq, that the binding identified by DAP-Seq is indeed functional and accurate, as we describe further below.

The MPRA used in this manuscript does not distinguish between enhanced transcription versus enhanced RNA stability or other mechanisms that may increase RNA levels. Using published data, the authors show that polymerase II occupancy (Figure 4C) and active histone marks increase with the number of GATC motifs, and show a correlation with mRNA synthesis but not mRNA half-life in Arabidopsis data. I agree that these results are consistent with the motif affecting RNA expression rather than RNA half-life; however, the motif may affect secondary structure, polymerase II pausing or processivity etc. rather than function as a bona fide enhancer (i.e. interact with a target gene promoter to enhance transcription). The strong claims made by the authors require direct evidence of enhancer-like function. Given the ease with which such experiments, including genetic manipulations, can be conducted in the model Arabidopsis, it is disappointing that such experiments have not been conducted.

To demonstrate that the observed effects in the MPRA are through transcription rate rather than mRNA stability, we have now directly measured the rate of synthesis in our MPRA setup, where fragments are positioned downstream of the TSS (setup explained in Fig. S10A). The results, shown in the new Fig. 2E, show a strong correlation between total mRNA levels and newly synthesized mRNA, confirming that transcription is the primary mechanism affected. Notably, fragments containing GATC motifs have a pronounced effect on mRNA synthesis, slightly stronger than on total mRNA levels (newly added Fig. S10B).

Regarding the reviewer's specific suggestion for alternative mechanisms, an effect on mRNA secondary structure would affect mRNA levels through mRNA stability, which we show is not the case with this new

MPRA-mRNA synthesis data. RNA polymerase pausing has not been detected in Arabidopsis (Hetzel et al. 2016). Finally, RNA polymerase processivity could not be the cause, as this would require RNA polymerase occupancy to strongly decrease in the first 500 bp downstream of the TSS (at least as much as the fold increase we see as a function of GATC motif number), but GRO-Seq data indicate a similar level of RNA polymerase II occupancy in these regions (Fig. 1E in (Hetzel et al. 2016)).

Finally, as we explain below, the mechanism is more straightforward: our new overexpression experiments of GATA factors further demonstrate that this transcriptional enhancement is mediated by transcription factors, consistent with enhancer activity.

Below are more detailed considerations, specific suggestions, and citations of prior work relevant to this manuscript.

Major issues:

1. The study lacks novelty: It is known that sequences in introns can increase mRNA levels both transcriptionally and post-transcriptionally (Callis, Fromm, and Walbot 1987; Laxa 2016; Shaul 2017; Le Hir, Nott, and Moore 2003). Notably, this is true in animals and plants. Rose and co-workers demonstrated that a GATC-like sequence motif increases expression in a dose-dependent manner (Alan B. Rose et al. 2016; Gallegos and Rose 2019) when residing within the transcribed sequence but not when residing upstream of the TSS (Gallegos and Rose 2017, 2019). A recent study showed that known plant enhancers and a highly active viral enhancer are inactive in the transcribed region in the MPRA design used in this manuscript (Jores et al. 2020). The authors need to discuss how their findings add to the rich existing knowledge. A promising avenue would be to generate mechanistic insights as to the function of the GATC motif with experiments geared towards identifying possibly bound proteins, and testing effects on transcription pre-initiation complex formation and on polymerase II pausing and processivity. Other possibilities include considerations of secondary structure, nucleosome position and supercoiling, all sequence-content- and position-dependent features that affect expression.

We do not agree with this assessment. That sequences in introns can increase mRNA levels is indeed well known, however, our main results are different, and can be summarized as follows:

1. The region downstream of the TSS is a hub of transcription regulatory sequences.
2. Unlike in animals, the identified plant enhancers activity is position-dependent.
3. We identify the GATC-motif, downstream of the TSS, as having a strong effect on the expression of thousands of genes.

These results and conclusions are far from being an accepted dogma. It is true that evidence from prior studies of single regulatory sequences/genes can now be interpreted as being in agreement with our work, and we cite such studies. However, none of these studies have provided a unified, generalizable framework, as we do. Specifically, the GATC motif was analyzed by Alan Rose using only a single promoter and found to be conserved in introns by (Back and Walther 2021). In contrast, our comprehensive analysis shows that the GATC motif exerts the strongest influence on transcriptional enhancement by a very large number of regulatory sequences, even upstream of the TSS. This is true even when compared to the results of the comprehensive study of upstream regulatory sequences by Jores et al. (2021). Furthermore, we are the first to demonstrate the broad relevance of the GATC motif by its strong effect on thousands of genes, which has not been shown before.

In addition, our newly added experimental data, included in the revised manuscript, show that the mode of action of these downstream enhancers is through transcription, not RNA stability, and that the GATC motif

acts through transcriptional regulation mediated by GATA transcription factors. Our discovery opens exciting avenues for further research and provides a comprehensive framework to link the diverse phenotypic outcomes of GATA TF mutations to their target genes with a GATC motif downstream of the TSS.

2. The authors imply that the expression-enhancing GATC motif functions as a transcriptional enhancer by binding GATA TFs (line 225). In contrast, earlier studies argue for an unknown, post-transcriptional mechanism (A. B. Rose and Last 1997; Samadder et al. 2008). The authors' suggestion is inconsistent with existing knowledge. GATA TFs are evolutionarily conserved zinc finger TFs that are found in plants, animals and fungi (Schwechheimer, Schröder, and Blaby-Haas 2022). GATA TFs are involved in brassinosteroid signaling and photomorphogenesis, processes controlling skotomorphogenic development, chlorophyll biosynthesis, chloroplast development, photosynthesis, and stomata formation, root, leaf, and flower development, seed dormancy and various stress responses. At least some of the many plant GATA TFs must function by recognizing the GATA TF motif (Dap-seq AGATCT for GATA11, consensus motif W-G-A-T-A-R) upstream of the TSS to control target gene expression (O'Malley et al. 2016; Schwechheimer, Schröder, and Blaby-Haas 2022). However, the study under review found that the GATC motif is not active upstream of the TSS, making it considerably less likely that this motif is bound by GATA TFs and enhances transcription in this way. In stark contrast, Meng et al. shows that transcriptional enhancers derived from introns of Arabidopsis genes retain their activity when cloned upstream of the TSS (Meng et al. 2021), as expected for enhancers. Weigel, one of the senior authors, previously provided evidence for intronic enhancers, reporting binding sites for LEAFY and WUSCHEL, both AGAMOUS activators, in the 3 kb AGAMOUS intron (Hong et al. 2003). Taken together, the GATC motif is more likely to function via the post-transcriptional intron-mediated enhancement mechanism than as a transcriptional enhancer. Without additional mechanistic evidence, the authors should focus their discussion on the modes of GATC action that are consistent with existing data.

The reviewer argues that, based on published results, the GATC motif most likely functions via post-transcriptional mechanisms rather than as a transcriptional enhancer. We disagree with the reviewer's interpretation of the literature, and will explain why below. However, before doing so, we note that we have added direct experimental evidence that the GATC motif does function as a transcriptional enhancer, making the whole argument academic.

Early work argued that the intron-mediated enhancement (IME) found in plants is due to a post-transcriptional mechanism specifically related to the splicing process (A. B. Rose and Last 1997). Later results caused the same authors to move away from this view, arguing that splicing per se is not required for IME (A. B. Rose and Beliakoff 2000) and that the mechanism is likely to be DNA-based (Alan B. Rose et al. 2011). **It is important to note that Alan Rose, who favored the post-transcriptional mechanism in his early work (A. B. Rose and Last 1997), has later on supported a mechanism based on transcription (Gallegos and Rose 2015).**

Regarding the GATA TFs: first, while GATA TFs are important both in animals and in plants, there is no conservation between kingdoms outside the DNA binding domain (Lowry and Atchley 2000; Schwechheimer, Schröder, and Blaby-Haas 2022). Second, as shown in the newly added Fig. S22, all systematic data created with DAP-Seq (O'Malley et al. 2016) on GATA TFs in Arabidopsis show no preference to bind G-A-T-A. Rather almost all GATA TFs bind downstream of the TSS and a large fraction of them bind a G-A-T-C motif. We also show in Fig. S23C the same to be true for GATA12 in maize, based on a re-analysis of published in vivo data (Tu et al. 2020). More generally, as we review in Supplementary note S1, GATA TFs in eukaryotes bind either a G-A-T-A motif or a G-A-T-C motif. (Note: Some Myb-related TFs in the DAP-Seq data show binding to G-A-T-A, which might explain the presence

of the GATA motif, but this is not connected to the GATA TFs (O'Malley et al. 2016; Zenker et al. 2023)).

To show that the binding identified by DAP-Seq is indeed functional, we experimentally identified the target genes of three GATA TFs using transient overexpression analysis. To this end, we transiently overexpressed three GATA TFs (GATA1, GATA4, and GATA6), along with GFP as control, in Arabidopsis leaf protoplasts for 8 hours followed by RNA-Seq for target identification. These specific TFs were selected based on their binding affinity to the GATC motif downstream of the TSS in the DAP-Seq data and their expression in leaf tissue, which was the source tissue for our protoplasts. The 8-hour time window was chosen to focus on direct effects, supported by a calibration experiment tracking fluorescence in GFP-expressing protoplasts, shown below:

Reviewer Figure R2: Percentage of protoplasts with nuclear GFP along an eight-hour time course after protoplast transformation. Arabidopsis leaf protoplasts were transformed with a plasmid with a double-35S promoter driving a GFP with a nuclear localization sequence (NLS), the same plasmid used as a control in the overexpression experiment.

The genes with increased expression in response to overexpression of each of the three GATAs are highly enriched in the GATC motif downstream of the TSS (newly added Fig. S24D). Moreover, the increase in expression was proportional to the occurrence of the motif downstream of the TSS (newly added Fig. 3H). These results, together with a new analysis of published datasets correlating GATA TF expression with the GATC motif effect across tissues (newly added Fig. S31), support the role of GATA TFs as the upstream regulators of the GATC motif.

As the reviewer mentioned, GATA TFs are involved in many biological processes in plants, which is consistent with the broad effect of the GATC motif on different tissues we identified (Fig. 4).

Regarding Meng et al. (2021), we first apologize for the oversight, we now cite the paper in our discussion. Note that our data show no correlation between the strength of the transcriptional effect from the two sides of the TSS, rather than a negative correlation, as expected if most sequences would be specialized to act on one side of the TSS. In this regard, we find the results of Meng et al. (2021) particularly interesting.

Meng et al. (2021) used DNA hypersensitivity (DHS) from either flowers or seedlings to identify open chromatin as a marker for enhancers in Arabidopsis introns. They then selected a few potential enhancers and positioned them upstream of the TSS of a GUS reporter and looked for tissue-specific expression patterns. A previous study by the same group (Zhu et al. 2015) used the same methodology and the same data set to identify intragenic enhancers, and they also positioned them upstream of a GUS gene to look for tissue-specific expression patterns. Thus, it is interesting to compare these two studies because, similar to our approach, they evaluate the activity of enhancers originating both upstream and downstream of the TSS for their effects upstream of the TSS.

In the intragenic enhancer paper (Zhu et al. 2015), the enhancers were found to function in the same tissue where the DHS signal was found (from the abstract: “the tissue specificity of the putative enhancers can be precisely predicted based on DNase I hypersensitivity data sets developed from different plant tissues”). However, when enhancers were identified in the intronic region in Meng et al. (2021), the link between the tissue in which the enhancer was identified and where the enhancer drove expression was

lost (Reviewer Figure R3). Therefore, the two papers provide a control on the tissue specificity of enhancers: when enhancers were identified upstream, the DHS is predictive of activity, whereas when the enhancers were identified downstream, there is no predictability. In other words, enhancers up- and downstream of the TSS appear to work in different ways. This result aligns with the lack of correlation in enhancer activity when sequences were placed up- or downstream of the TSS in our dataset, particularly from the perspective of a specific tissue as measured in the MPRA.

Reviewer Figure R3 highlights the tissue specificity of enhancers according to data from Table S2 in Meng et al. Intronic enhancers identified via DHS in seedlings or flower buds were placed upstream of the TSS of a GUS gene. Notably, enhancers identified in the flower DHS, but not in seedlings, did not show increased expression in flowers, in contrast to previous findings where intragenic enhancers on the same side of the TSS predicted tissue specificity.

Lastly, Meng et al. (2021) observed a higher propensity for DHS-defined enhancers to be located in first introns, consistent with positional dependence. This is in agreement with our findings of increased signals at the beginning of genes for eQTL, TF binding, and the presence of the GATC motif.

To summarize, our data in addition to the DAP-Seq data fully support a mode of action of GATA TF binding to the GATC motif downstream of the TSS and increasing transcription from the core promoter.

3. The authors use an intron for the "downstream MPRA" to "rule out effects due to altered sequence of the mature mRNA and thus minimizing effects on mRNA stability" (lines 89-90). This is a curious argument as many of these sequences presumably reside outside of introns in the RNA and exert their function in this context. In any case, the argument holds only true if the intron is spliced out equally efficiently. The introduction of foreign sequences into introns can lead to incorrect or inefficient splicing (Yofe et al. 2014). Such mis-splicing could have a strong effect on the levels of the corresponding RNAs and would lead to misinterpretation of the MPRA data. The authors should assess the splicing efficiency of their library, or do so for selected subsets (e.g., for sequences with high or low reporter gene expression).

We acknowledge the concern raised by the reviewer and have taken steps to address it. To minimize the possibility of missplicing, we initially removed acceptor and donor splice sites in our synthetic fragment design. In addition, as suggested by the reviewer, we have now estimated the splicing efficiency in our library. To do this, we used an aliquot of RNA from our original four replicates of MPRA in Arabidopsis. This RNA was used for RNA-Seq with the SMART-Seq3 protocol (Hagemann-Jensen et al. 2020) and sequenced to deep coverage. In these libraries, we can identify splice junctions and estimate splicing efficiency (details in the Methods section). From all four experiments together we were able to identify 7,405 reads for which we can identify correct splicing against merely 100 reads that indicate an unspliced event. Therefore, we estimate that 98.7% of the library produces correct splicing events, or 99.39%, 98.91%, 98.55%, 98.34% for each of the four repeats. Therefore, we believe that mis-splicing does not significantly affect our results.

4. While the experiments described in this study generated a large amount of data, much of this data

remains unanalyzed. For example, species-specific differences are ignored even though they are clearly present. Results from the dicots *Arabidopsis*, tomato and tobacco appear more closely correlated with one another than with results from the monocot maize; this is especially noticeable in the "upstream MPRA". Given the strong differences in GC content between dicot and monocot genomes, GC content is a strong candidate for explaining these differences.

We agree with the reviewer's assessment regarding the difference between monocots and dicots and the likely effect of GC content in this. This is consistent with the work of Jores et al. (2021). Crucially, our main results are consistent across species, i.e., the effect of the GATC motif was strong enough to be detected in maize, and the lack of correlation between the effect of upstream and downstream positioned fragments in all four species.

While we acknowledge the presence of species-specific differences in the data, further investigation of these differences is beyond the scope of this study. We hope that the dataset generated from our experiments can serve as a valuable resource for the scientific community to explore such questions.

5. It is striking that the strongest effects on transcription are observed in the "upstream MPRA", yet the authors focus almost exclusively on the results of the "downstream MPRA". Further careful analysis, in particular of the "upstream MPRA" data, could identify sequence motifs that are associated with species-specific or position-dependent element activity. Such additional analyses might considerably improve the novelty and hence the value of this study.

While we acknowledge the potential for further analysis of the upstream region, this is beyond the scope of this work. We envision that our dataset will serve as a valuable resource, complementing the comprehensive datasets of Jores et al. (2021) and facilitating future investigations of upstream regulatory sequences and their species-specific differences.

6. The authors show that the GATC motif is active downstream of the TSS but not upstream of the TSS in the *Arabidopsis* gene expression data. Does the same hold true in the MPRA data? In other words, is the GATC motif associated with higher (or lower) reporter gene expression in the "upstream MPRA"? I would be interested to see the results for the GATC motif in the "upstream MPRA" analogous to those shown in Figure 3 for the "downstream MPRA".

We appreciate the reviewer's suggestion and have included it in our revised manuscript. In the newly added Fig. S15, we show the association of gene expression with the presence of the GATC motif in all eight MPRA setups, comparing the upstream and downstream configurations side by side. As expected, the analysis shows different effects between the upstream and downstream configurations. While some systems show a modest increase in average expression in the upstream configuration, this effect is significantly weaker compared to the pronounced effect observed in the downstream configuration. Furthermore, as can be seen in response to a question from reviewer #2 above (Fig. R1), there is also no association between the upstream GATC motif and gene expression in the data from Jores et al. (2021).

7. The figures need to be revised to improve clarity in order to allow readers to draw their own conclusions. Here are some suggestions for changes to the main Figures (corresponding changes should also be made to the Supplementary Figures):

- The sample size must be indicated for all box and violin plots and for the correlation scatter plots

Sample sizes were added to the graphs or captions.

- Axis labels should be more descriptive. Example of insufficient axis labels are: "Repeat 1" and "Repeat 2" (Fig. 2B), "upstream" and "downstream" (Fig 2D), "Tomato - p35S" and "N. be - pTRP1" (Fig. 3B)

Information added to the captions.

- For easier comparisons, the authors should use the same scales for all sub-plots in Figs. 2B, 2E, and 3C

In Figures 2B, 2E, and 3C, we use different scales to effectively highlight specific aspects of our data: reproducibility in Figure 2B, within-position variation in Figure 2E, and within-species differences in Figure 3C. We believe that these different scales better illustrate the relevant biological phenomena.

- Using partially transparent dots or a different plot type (e.g., a hexbin plot) would help to visualize the distribution in the correlation scatter plots (Fig. 2B and 2D)

Done.

- In Fig. 1D, the authors should use a linear color scale across the whole range (currently the range 0 to ~60 is all white)

Done.

- In Fig. 3B and 3C, it is not clear what the dot and lines represent (mean +/- sd?). This should be described clearly. Furthermore, converting the plots to violin plots could help better visualize the distribution of the (up to thousands of) data points

Error bars in B-C represent mean \pm 1 standard deviation; this information was added to the captions. Since the figure is already very crowded and adding more might make it difficult to understand, we prefer not to change the graphs to violin plots.

- In Fig. 4B, it is unclear whether the shaded area with GATC containing 6-mers applies to both the "Upstream" and "Downstream" categories or to just one of the two. Another way of highlighting the GATC-containing 6-mers would improve clarity.

Thanks for pointing this out. The shaded area in Figure 4B was intended to represent only the 'Downstream' category. To improve clarity, we have now encircled these dots.

- In Fig. 5, it is not described what the individual rows in the heatmap and the points in the boxplots represent. Are these different transcriptome data sets? This needs to be explained and the sample size must be indicated for each species.

These are different RNA-seq samples; details have been added.

Minor issues:

8. The title is not accurate for the reasons discussed above.

9. The introduction, just as the title, does not match the manuscript. Instead of discussing the evolution of gene regulation in plants and animals, it would be helpful to introduce concepts such as intron-mediated enhancement and previous findings on position-independent plant enhancers residing upstream of the TSS and in introns.

We address points 8 and 9 together in this response. Our main findings focus on the regulatory significance of the region downstream of the TSS, the position-dependent nature of enhancers within this region compared to the region upstream of the TSS, and the distinct role of the GATC motif in the downstream region. We believe that these key findings provide a novel perspective on gene regulation in plants. We hope that our response to the previous concerns, together with the inclusion of new experimental data demonstrating the effect of mRNA synthesis rate rather than mRNA stability, the

confirmation of correct intron splicing in our system, and the elucidation of the effect of the GATC motif by GATA TFs, further strengthens the validity and significance of our findings.

10. In the deep mutational scan, the authors find "additional sequences that do not include GATC motifs as important for enhancing activity" (lines 176-178). The authors should list and discuss these additional sequences.

With this sentence we wanted to indicate that not only mutations in the GATC motif but also in other regions affect gene expression. To clearly demonstrate which sequences are sensitive to mutation, we have included a summary of all single-bp mutation results, shown as sequence logos in the newly added Fig. S20. In this figure, the size of each nucleotide visually represents the average effect of its mutation on expression levels. This points to additional sequences critical for the regulation of gene expression.

11. It is unclear how the authors generated the GATC motif. The text "we combined the six 6-mers with the strongest effect into an 8-bp YVGATCBR motif" (lines 158-159) is insufficient. Which 6-mers were chosen and why? Why were only six 6-mers utilized? How were they combined? Did the authors try alternative methods, like searching for motifs enriched in the most active sequences? The authors should include a detailed description of how the motif was generated and ideally make the motif available in a suitable format (as a .meme file?) for use by others.

We have added details on the construction of the motif in the captions of Fig. S14, as follows: "The six 6-mers (CAGatc, AgatcT, AgatcC, gatcTA, gatcCA, and AgatcG) with the highest median effect were used to establish the consensus YVGATCBR sequence of the GATC motif, which was then used in downstream analyses". In addition, we explicitly define YVGATCBR as the 'GATC motif' in the main text for clarity.

The choice of six top 6-mers was arbitrary. As shown in Fig. S14, other 6-mers also show an association with gene expression. In addition, our deep mutation scanning shows that any mutation to the 4-bp GATC core significantly reduces expression, with less constraint observed in the 2-bp adjacent to the GATC core. In practice, different approaches yielded consistent results due to the strong effect of the motif. For example, HOMER motif analysis on fragments originating downstream of the TSS, comparing the 500 fragments associated with the strongest gene expression levels in the MPRA setup to all 8,000 downstream-derived fragments as background, consistently highlighted the GATC motif as the top identified motif. Here are the top results from this analysis using Arabidopsis MPRA data in the 35S promoter-based MPRA:

* - possible false positive

Rank	Motif	P-value	log P-value	% of Targets	% of Background	STD(Bg STD)	Best Match/Details	Motif File
1		1e-105	-2.423e+02	85.40%	36.42%	43.7bp (62.4bp)	GAT3/MA0301.1/Jaspar(0.968) More Information Similar Motifs Found	motif file (matrix)
2 *		1e-4	-9.585e+00	4.60%	1.73%	30.0bp (44.2bp)	LEC2/MA0581.1/Jaspar(0.738) More Information Similar Motifs Found	motif file (matrix)
3 *		1e-3	-8.874e+00	10.40%	5.93%	47.5bp (59.3bp)	GATA20/MA1324.1/Jaspar(0.738) More Information Similar Motifs Found	motif file (matrix)
4 *		1e-3	-7.444e+00	7.80%	4.32%	45.0bp (59.6bp)	SeqBias: polyA-repeat(0.928) More Information Similar Motifs Found	motif file (matrix)
5 *		1e-1	-4.339e+00	1.00%	0.26%	34.9bp (40.5bp)	SeqBias: TA-repeat(0.866) More Information Similar Motifs Found	motif file (matrix)

For the known motif results in HOMER we get GATA TFs from DAP-Seq in all top results:

Rank	Motif	Name	P-value	log P-value	q-value (Benjamini)	# Target Sequences with Motif	% of Targets Sequences with Motif	# Background Sequences with Motif	% of Background Sequences with Motif	Motif File	SVG
1		GATA12(C2C2gata)/col-GATA12-DAP-Seq(GSE60143)/Homer	1e-79	-1.824e+02	0.0000	307.0	61.40%	1448.2	20.63%	motif file (matrix)	svg
2		GATA4(C2C2gata)/col-GATA4-DAP-Seq(GSE60143)/Homer	1e-69	-1.591e+02	0.0000	344.0	68.80%	2038.0	29.04%	motif file (matrix)	svg
3		GATA14(C2C2gata)/col-GATA14-DAP-Seq(GSE60143)/Homer	1e-68	-1.572e+02	0.0000	295.0	59.00%	1488.2	21.20%	motif file (matrix)	svg

And for the pTRP1-based MPRA:

* - possible false positive

Rank	Motif	P-value	log P-value	% of Targets	% of Background	STD(Bg STD)	Best Match/Details	Motif File
1		1e-64	-1.482e+02	74.20%	35.48%	43.9bp (57.6bp)	GAT4/MA0302.1/Jaspar(0.979) More Information Similar Motifs Found	motif file (matrix)
2*		1e-4	-1.128e+01	6.20%	2.50%	38.1bp (56.4bp)	MATR3(RRM)/Homo_sapiens-RNCMPT00037-PBM/HughesRNA(0.685) More Information Similar Motifs Found	motif file (matrix)
3*		1e-4	-1.118e+01	14.60%	8.58%	44.5bp (49.0bp)	prd/dmmpmm(Down)/fly(0.741) More Information Similar Motifs Found	motif file (matrix)

And known motifs are again GATA TFs from the DAP-Seq data:

Rank	Motif	Name	P-value	log P-value	q-value (Benjamini)	# Target Sequences with Motif	% of Targets Sequences with Motif	# Background Sequences with Motif	% of Background Sequences with Motif	Motif File	SVG
1		GATA14(C2C2gata)/col-GATA14-DAP-Seq(GSE60143)/Homer	1e-45	-1.040e+02	0.0000	244.0	48.80%	1435.5	19.33%	motif file (matrix)	svg
2		GATA12(C2C2gata)/col-GATA12-DAP-Seq(GSE60143)/Homer	1e-44	-1.035e+02	0.0000	239.0	47.80%	1385.9	18.66%	motif file (matrix)	svg
3		GATA4(C2C2gata)/col-GATA4-DAP-Seq(GSE60143)/Homer	1e-38	-8.851e+01	0.0000	276.0	55.20%	1962.7	26.43%	motif file (matrix)	svg

Since the motif is defined combinatorially, we believe that a PWM file is unnecessary.

12. On lines 122-124, the authors state: "Enhancers were more effective in the CaMV 35S promoter construct, than in the TRP1 promoter construct, in which fragments insertions disrupted the TRP1 genomic sequence." Since the plots for the 35S and TRP1 promoter constructs in Fig. 2E and Fig. S9A are normalized to different control constructs, it is not possible for readers to directly compare the two. A plot for the direct comparison of reporter gene expression across the two different promoters should be included in the manuscript. Furthermore, if the disruption of the TRP1 genomic sequence is the cause for the observed differences, then the two promoters should behave similarly in the "downstream MPRA" context. This is not the case, though, and the authors should discuss this finding and possible explanations.

We meant the relative effect of the enhancer compared to the no insertion control. We now rewrite the text as "Enhancers were relatively more effective in the CaMV 35S ...". We do not compare the different core promoters in the manuscript because in our current system the libraries with the *TRP1* promoter or the 35S minimal promoter cannot be compared directly, as they were prepared with different 5'-end primers for enrichment of MPRA-derived transcripts.

We do not argue that the reason for all the differences between the two core promoters is due to the pTRP1 interference, we only suggest this is the reason for the lower relative effect on gene expression. As the reviewer wrote, the correlation between downstream libraries with the two promoters suggests a dependence of the relative enhancer effect on the core promoter.

Finally, as requested by reviewer #2, we now provide an estimate of the relative mRNA levels of the two no-insert constructs calculated using the same SMART-Seq3 libraries used to estimate splicing efficiency. Using these data, we observed that the expression of the minimal 35S promoter is about 4 times stronger than the "minimal" TRP1 promoter. We have now described this in the Methods section.

References:

- Callis, J., M. Fromm, and V. Walbot. 1987. "Introns Increase Gene Expression in Cultured Maize Cells." *Genes & Development* 1 (10): 1183–1200.
- Chung, Betty Y. W., Cas Simons, Andrew E. Firth, Chris M. Brown, and Roger P. Hellens. 2006. "Effect of 5'UTR Introns on Gene Expression in *Arabidopsis Thaliana*." *BMC Genomics* 7 (1): 120.
- Gallegos, Jenna E., and Alan B. Rose. 2017. "Intron DNA Sequences Can Be More Important than the Proximal Promoter in Determining the Site of Transcript Initiation." *The Plant Cell* 29 (4): 843–53.
- Gallegos, Jenna E., and Alan B. Rose. 2019. "An Intron-Derived Motif Strongly Increases Gene Expression from Transcribed Sequences through a Splicing Independent Mechanism in *Arabidopsis Thaliana*." *Scientific Reports* 9 (1): 13777.
- Hong, Ray L., Lynn Hamaguchi, Maximilian A. Busch, and Detlef Weigel. 2003. "Regulatory Elements of the Floral Homeotic Gene *AGAMOUS* Identified by Phylogenetic Footprinting and Shadowing." *The Plant Cell* 15 (6): 1296–1309.
- Jores, Tobias, Jackson Tonnie, Michael W. Dorrity, Josh T. Cuperus, Stanley Fields, and Christine Queitsch. 2020. "Identification of Plant Enhancers and Their Constituent Elements by STARR-Seq in Tobacco Leaves." *The Plant Cell* 32 (7): 2120–31.
- Laxa, Miriam. 2016. "Intron-Mediated Enhancement: A Tool for Heterologous Gene Expression in Plants?" *Frontiers in Plant Science* 7: 1977.
- Laxa, Miriam, Kristin Müller, Natalie Lange, Lennart Doering, Jan Thomas Pruscha, and Christoph Peterhänsel. 2016. "The 5'UTR Intron of *Arabidopsis* GGT1 Aminotransferase Enhances Promoter Activity by Recruiting RNA Polymerase II." *Plant Physiology* 172 (1): 313–27.
- Le Hir, Hervé, Ajit Nott, and Melissa J. Moore. 2003. "How Introns Influence and Enhance Eukaryotic Gene Expression." *Trends in Biochemical Sciences* 28 (4): 215–20.
- Meng, Fanli, Hainan Zhao, Bo Zhu, Tao Zhang, Mingyu Yang, Yang Li, Yingpeng Han, and Jiming Jiang. 2021. "Genomic Editing of Intronic Enhancers Unveils Their Role in Fine-Tuning Tissue-Specific Gene Expression in *Arabidopsis Thaliana*." *The Plant Cell* 33 (6): 1997–2014.
- O'Malley, Ronan C., Shao-Shan Carol Huang, Liang Song, Mathew G. Lewsey, Anna Bartlett, Joseph R. Nery, Mary Galli, Andrea Gallavotti, and Joseph R. Ecker. 2016. "Cistrome and Epicistrome Features Shape the Regulatory DNA Landscape." *Cell* 166 (6): 1598.
- Rose, A. B., and R. L. Last. 1997. "Introns Act Post-Transcriptionally to Increase Expression of the *Arabidopsis Thaliana* Tryptophan Pathway Gene *PAT1*." *The Plant Journal: For Cell and Molecular Biology* 11 (3): 455–64.
- Rose, Alan B. 2004. "The Effect of Intron Location on Intron-mediated Enhancement of Gene Expression in *Arabidopsis*." *The Plant Journal: For Cell and Molecular Biology* 40 (5): 744–51.
- Rose, Alan B., and Jason A. Beliakoff. 2000. "Intron-Mediated Enhancement of Gene Expression Independent of Unique Intron Sequences and Splicing." *Plant Physiology* 122 (2): 535–42.
- Rose, Alan B., Amanda Carter, Ian Korf, and Noah Kojima. 2016. "Intron Sequences That Stimulate Gene Expression in *Arabidopsis*." *Plant Molecular Biology* 92 (3): 337–46.
- Samadder, Partha, Elumalai Sivamani, Jianli Lu, Xianggan Li, and Rongda Qu. 2008. "Transcriptional and Post-Transcriptional Enhancement of Gene Expression by the 5' UTR Intron of Rice *Rubi3* Gene in Transgenic Rice Cells." *Molecular Genetics and Genomics: MGG* 279 (4): 429–39.
- Schmitz, Robert J., Erich Grotewold, and Maïke Stam. 2022. "Cis-Regulatory Sequences in Plants: Their Importance, Discovery, and Future Challenges." *The Plant Cell* 34 (2): 718–41.
- Schwechheimer, Claus, Peter Michael Schröder, and Crysten E. Blaby-Haas. 2022. "Plant GATA Factors: Their Biology, Phylogeny, and Phylogenomics." *Annual Review of Plant Biology* 73 (1): 123–48.
- Shaul, Orit. 2017. "How Introns Enhance Gene Expression." *The International Journal of Biochemistry & Cell Biology* 91 (Pt B): 145–55.
- Strunz, Tobias, Felix Grassmann, Javier Gayán, Satu Nahkuri, Debora Souza-Costa, Cyrille Maugeais, Sascha Fauser, Everson Nogoceke, and Bernhard H. F. Weber. 2018. "A Mega-Analysis of Expression Quantitative Trait Loci (EQL) Provides Insight into the Regulatory Architecture of Gene Expression Variation in Liver." *Scientific Reports* 8 (1): 5865.
- Wang, Jiangen, Juhong Liu, and Zilong Guo. 2023. "Natural UORF Variation in Plants." *Trends in Plant Science*, August. <https://doi.org/10.1016/j.tplants.2023.07.005>.
- Yofe, Ido, Zohar Zafir, Rachel Blau, Maya Schuldiner, Tamir Tuller, Ehud Shapiro, and Tuval Ben-Yehzekel. 2014. "Accurate, Model-Based Tuning of Synthetic Gene Expression Using Introns in *S. Cerevisiae*." *PLoS Genetics* 10 (6):

References:

- Arnold, Cosmas D., Daniel Gerlach, Christoph Stelzer, Łukasz M. Boryń, Martina Rath, and Alexander Stark. 2013. "Genome-Wide Quantitative Enhancer Activity Maps Identified by STARR-Seq." *Science* 339 (6123): 1074–77.
- Back, Georg, and Dirk Walther. 2021. "Identification of Cis-Regulatory Motifs in First Introns and the Prediction of Intron-Mediated Enhancement of Gene Expression in Arabidopsis Thaliana." *BMC Genomics* 22 (1): 390.
- Banerji, J., S. Rusconi, and W. Schaffner. 1981. "Expression of a Beta-Globin Gene Is Enhanced by Remote SV40 DNA Sequences." *Cell* 27 (2 Pt 1): 299–308.
- Ben-Tov, Daniela, Fabrizio Mafessoni, Amit Cucuy, Arik Honig, Cathy Melamed-Bessudo, and Avraham A. Levy. 2023. "Uncovering the Dynamics of Precise Repair at CRISPR/Cas9-Induced Double-Strand Breaks." *bioRxiv*. <https://doi.org/10.1101/2023.01.10.523377>.
- Chodavarapu, Ramakrishna K., Suhua Feng, Yana V. Bernatavichute, Pao-Yang Chen, Hume Stroud, Yanchun Yu, Jonathan A. Hetzel, et al. 2010. "Relationship between Nucleosome Positioning and DNA Methylation." *Nature* 466 (7304): 388–92.
- Emms, David M., and Steven Kelly. 2019. "OrthoFinder: Phylogenetic Orthology Inference for Comparative Genomics." *Genome Biology* 20 (1): 238.
- Gallegos, Jenna E., and Alan B. Rose. 2015. "The Enduring Mystery of Intron-Mediated Enhancement." *Plant Science: An International Journal of Experimental Plant Biology* 237 (August): 8–15.
- Hagemann-Jensen, Michael, Christoph Ziegenhain, Ping Chen, Daniel Ramsköld, Gert-Jan Hendriks, Anton J. M. Larsson, Omid R. Faridani, and Rickard Sandberg. 2020. "Single-Cell RNA Counting at Allele and Isoform Resolution Using Smart-seq3." *Nature Biotechnology* 38 (6): 708–14.
- Hahn, Steven, and Elton T. Young. 2011. "Transcriptional Regulation in *Saccharomyces Cerevisiae*: Transcription Factor Regulation and Function, Mechanisms of Initiation, and Roles of Activators and Coactivators." *Genetics* 189 (3): 705–36.
- Hetzel, Jonathan, Sascha H. Duttke, Christopher Benner, and Joanne Chory. 2016. "Nascent RNA Sequencing Reveals Distinct Features in Plant Transcription." *Proceedings of the National Academy of Sciences of the United States of America* 113 (43): 12316–21.
- Jabre, Ibtissam, Saurabh Chaudhary, Wenbin Guo, Maria Kalyna, Anireddy S. N. Reddy, Weizhong Chen, Runxuan Zhang, Cornelia Wilson, and Naeem H. Syed. 2021. "Differential Nucleosome Occupancy Modulates Alternative Splicing in Arabidopsis Thaliana." *The New Phytologist* 229 (4): 1937–45.
- Jores, Tobias, Jackson Tonnies, Michael W. Dorrity, Josh T. Cuperus, Stanley Fields, and Christine Queitsch. 2020. "Identification of Plant Enhancers and Their Constituent Elements by STARR-Seq in Tobacco Leaves." *The Plant Cell* 32 (7): 2120–31.
- Jores, Tobias, Jackson Tonnies, Travis Wrightsman, Edward S. Buckler, Josh T. Cuperus, Stanley Fields, and Christine Queitsch. 2021. "Synthetic Promoter Designs Enabled by a Comprehensive Analysis of Plant Core Promoters." *Nature Plants* 7 (6): 842–55.
- Kumar, Sudhir, Michael Suleski, Jack M. Craig, Adrienne E. Kasproicz, Maxwell Sanderford, Michael Li, Glen Stecher, and S. Blair Hedges. 2022. "TimeTree 5: An Expanded Resource for Species Divergence Times." *Molecular Biology and Evolution* 39 (8). <https://doi.org/10.1093/molbev/msac174>.
- Lin, Zhenguo, and Wen-Hsiung Li. 2012. "Evolution of 5' Untranslated Region Length and Gene Expression Reprogramming in Yeasts." *Molecular Biology and Evolution* 29 (1): 81–89.
- Liu, Yuwen, Shan Yu, Vineet K. Dhiman, Tonya Brunetti, Heather Eckart, and Kevin P. White.

2017. "Functional Assessment of Human Enhancer Activities Using Whole-Genome STARR-Sequencing." *Genome Biology* 18 (1): 219.
- Lowry, J. A., and W. R. Atchley. 2000. "Molecular Evolution of the GATA Family of Transcription Factors: Conservation within the DNA-Binding Domain." *Journal of Molecular Evolution* 50 (2): 103–15.
- Meng, Fanli, Hainan Zhao, Bo Zhu, Tao Zhang, Mingyu Yang, Yang Li, Yingpeng Han, and Jiming Jiang. 2021. "Genomic Editing of Intronic Enhancers Unveils Their Role in Fine-Tuning Tissue-Specific Gene Expression in Arabidopsis Thaliana." *The Plant Cell* 33 (6): 1997–2014.
- O'Malley, Ronan C., Shao-Shan Carol Huang, Liang Song, Mathew G. Lewsey, Anna Bartlett, Joseph R. Nery, Mary Galli, Andrea Gallavotti, and Joseph R. Ecker. 2016. "Cistrome and Epicistrome Features Shape the Regulatory DNA Landscape." *Cell* 165 (5): 1280–92.
- Reyes, José C., M. Isabel Muro-Pastor, and Francisco J. Florencio. 2004. "The GATA Family of Transcription Factors in Arabidopsis and Rice." *Plant Physiology* 134 (4): 1718–32.
- Rose, A. B., and J. A. Beliakoff. 2000. "Intron-Mediated Enhancement of Gene Expression Independent of Unique Intron Sequences and Splicing." *Plant Physiology* 122 (2): 535–42.
- Rose, A. B., and R. L. Last. 1997. "Introns Act Post-Transcriptionally to Increase Expression of the Arabidopsis Thaliana Tryptophan Pathway Gene PAT1." *The Plant Journal: For Cell and Molecular Biology* 11 (3): 455–64.
- Rose, Alan B., Shahram Emami, Keith Bradnam, and Ian Korf. 2011. "Evidence for a DNA-Based Mechanism of Intron-Mediated Enhancement." *Frontiers in Plant Science* 2 (December): 98.
- Schon, Michael A., Max J. Kellner, Alexandra Plotnikova, Falko Hofmann, and Michael D. Nodine. 2018. "NanoPARE: Parallel Analysis of RNA 5' Ends from Low-Input RNA." *Genome Research* 28 (12): 1931–42.
- Schwartz, Schraga, and Gil Ast. 2010. "Chromatin Density and Splicing Destiny: On the Cross-Talk between Chromatin Structure and Splicing." *The EMBO Journal* 29 (10): 1629–36.
- Schwartz, Schraga, Eran Meshorer, and Gil Ast. 2009. "Chromatin Organization Marks Exon-Intron Structure." *Nature Structural & Molecular Biology* 16 (9): 990–95.
- Schwechheimer, Claus, Peter Michael Schröder, and Crysten E. Blaby-Haas. 2022. "Plant GATA Factors: Their Biology, Phylogeny, and Phylogenomics." *Annual Review of Plant Biology* 73 (May): 123–48.
- Sebé-Pedrós, Arnau, Cecilia Ballaré, Helena Parra-Acero, Cristina Chiva, Juan J. Tena, Eduard Sabidó, José Luis Gómez-Skarmeta, Luciano Di Croce, and Iñaki Ruiz-Trillo. 2016. "The Dynamic Regulatory Genome of Capsaspora and the Origin of Animal Multicellularity." *Cell* 165 (5): 1224–37.
- Small, Stephen, and David N. Arnosti. 2020. "Transcriptional Enhancers in Drosophila." *Genetics* 216 (1): 1–26.
- Struhl, K. 1995. "Yeast Transcriptional Regulatory Mechanisms." *Annual Review of Genetics* 29: 651–74.
- Strunz, Tobias, Felix Grassmann, Javier Gayán, Satu Nahkuri, Debora Souza-Costa, Cyrille Maugeais, Sascha Fauser, Everson Nogoceke, and Bernhard H. F. Weber. 2018. "A Mega-Analysis of Expression Quantitative Trait Loci (eQTL) Provides Insight into the Regulatory Architecture of Gene Expression Variation in Liver." *Scientific Reports* 8 (1): 5865.
- Sun, Jialei, Na He, Longjian Niu, Yingzhang Huang, Wei Shen, Yuedong Zhang, Li Li, and Chunhui Hou. 2019. "Global Quantitative Mapping of Enhancers in Rice by STARR-Seq." *Genomics, Proteomics & Bioinformatics* 17 (2): 140–53.
- Tan, Yongjun, Xiaohao Yan, Jialei Sun, Jing Wan, Xinxin Li, Yingzhang Huang, Li Li, Longjian Niu, and Chunhui Hou. 2023. "Genome-Wide Enhancer Identification by Massively Parallel

- Reporter Assay in Arabidopsis." *The Plant Journal: For Cell and Molecular Biology* 116 (1): 234–50.
- Tu, Xiaoyu, María Katherine Mejía-Guerra, Jose A. Valdes Franco, David Tzeng, Po-Yu Chu, Wei Shen, Yingying Wei, et al. 2020. "Reconstructing the Maize Leaf Regulatory Network Using ChIP-Seq Data of 104 Transcription Factors." *Nature Communications* 11 (1): 5089.
- Ushijima, Tomokazu, Kousuke Hanada, Eiji Gotoh, Wataru Yamori, Yutaka Kodama, Hiroyuki Tanaka, Miyako Kusano, et al. 2017. "Light Controls Protein Localization through Phytochrome-Mediated Alternative Promoter Selection." *Cell* 171 (6): 1316–25.e12.
- Veyrieras, Jean-Baptiste, Sridhar Kudaravalli, Su Yeon Kim, Emmanouil T. Dermitzakis, Yoav Gilad, Matthew Stephens, and Jonathan K. Pritchard. 2008. "High-Resolution Mapping of Expression-QTLs Yields Insight into Human Gene Regulation." *PLoS Genetics* 4 (10): e1000214.
- Weigel, D., J. H. Ahn, M. A. Blázquez, J. O. Borevitz, S. K. Christensen, C. Fankhauser, C. Ferrándiz, et al. 2000. "Activation Tagging in Arabidopsis." *Plant Physiology* 122 (4): 1003–13.
- Zenker, Sanja, Donat Wulf, Anja Meierhenrich, Sarah Becker, Marion Eisenhut, Ralf Stracke, Bernd Weisshaar, and Andrea Bräutigam. 2023. "Transcription Factors Operate on a Limited Vocabulary of Binding Motifs in Arabidopsis Thaliana." *bioRxiv*. <https://doi.org/10.1101/2023.08.28.555073>.
- Zhu, Bo, Wenli Zhang, Tao Zhang, Bao Liu, and Jiming Jiang. 2015. "Genome-Wide Prediction and Validation of Intergenic Enhancers in Arabidopsis Using Open Chromatin Signatures." *The Plant Cell* 27 (9): 2415–26.
- Zilberman, Daniel, Mary Gehring, Robert K. Tran, Tracy Ballinger, and Steven Henikoff. 2007. "Genome-Wide Analysis of Arabidopsis Thaliana DNA Methylation Uncovers an Interdependence between Methylation and Transcription." *Nature Genetics* 39 (1): 61–69.
- Zuin, Jessica, Gregory Roth, Yinxiu Zhan, Julie Cramard, Josef Redolfi, Ewa Piskadlo, Pia Mach, et al. 2022. "Nonlinear Control of Transcription through Enhancer-Promoter Interactions." *Nature* 604 (7906): 571–77.

Decision Letter, first revision:**Message:** 21st Jun 2024

Dear Yoav,

Your Letter, "Widespread position-dependent transcriptional regulatory sequences in plants" has now been seen by 2 of the original 3 referees. You will see from their comments below that while they find your work of interest, some important points are raised. We are interested in the possibility of publishing your study in Nature Genetics, but would like to consider your response to these concerns in the form of a revised manuscript before we make a final decision on publication.

The previously-critical Reviewer #3 was, unfortunately, not able to review the revision, and so we asked the other referees to comment on your responses to the critiques of #3.

In brief, the two reviewers both appreciate the improvements made in revision and sound supportive of an eventual publication. Importantly, both also found your responses to Referee #3 convincing. Reviewer #2 raises a few new, but minor points that will need some additional analysis; to our reading they seem very reasonable and would not require substantial expansion of the work in its current form.

To guide the scope of the revisions, the editors discuss the referee reports in detail within the team, including with the chief editor, with a view to identifying key priorities that should be addressed in revision and sometimes overruling referee requests that are deemed beyond the scope of the current study. We hope that you will find the prioritized set of referee points to be useful when revising your study. Please do not hesitate to get in touch if you would like to discuss these issues further.

We therefore invite you to revise your manuscript taking into account all reviewer and editor comments. Please highlight all changes in the manuscript text file. At this stage we will need you to upload a copy of the manuscript in MS Word .docx or similar editable format.

*1) Include a “Response to referees” document detailing, point-by-point, how you addressed each referee comment. If no action was taken to address a point, you must provide a compelling argument. This response will be sent back to the referees along with the revised manuscript.

*2) If you have not done so already please begin to revise your manuscript so that it conforms to our Letter format instructions, available here.

*3) Include a revised version of any required Reporting Summary:

Please be aware of our guidelines on digital image standards.

[Redacted]

We hope to receive your revised manuscript within four to eight weeks. If you cannot send it within this time, please let us know.

Nature Genetics is committed to improving transparency in authorship. As part of our efforts in this direction, we are now requesting that all authors identified as ‘corresponding author’ on published papers create and link their Open Researcher and Contributor Identifier (ORCID) with their account on the Manuscript Tracking System (MTS), prior to acceptance.

ORCID helps the scientific community achieve unambiguous attribution of all scholarly contributions. You can create and link your ORCID from the home page of the MTS by clicking on 'Modify my Springer Nature account'. For more information please visit please visit www.springernature.com/orcid.

Sincerely,

Michael Fletcher, PhD
Senior Editor, Nature Genetics
ORCID: 0000-0003-1589-7087

Reviewers' Comments:

Reviewer #1:

Remarks to the Author:

I appreciated the authors careful and comprehensive response to the reviews. Many of my comments were driven by lack of clarity in understanding some aspects of the work. The revisions (and responses) were helpful in clarifying details of this complex study. I appreciate that the authors were willing to undertake additional analyses (comparative genomics of 5' UTR length, re-centering plots on the AUG, etc). These answered curiosities that I had and I hope the authors find them useful. It was important to provide more clear description of the key motif and the additions now provide that. I felt that my comments and suggestions were fully addressed.

The editor also requested an evaluation of the responses to reviewer 3. I can see the perspective of reviewer 3 but also understand the perspective of the authors. I appreciate the authors willingness to engage in a dialogue and to provide details to support their perspective. A key criticism of reviewer 3 was a lack of novelty, a more nuanced description of the findings – especially relative to animal systems and potential limitations of the key assays. In each case I thought that the authors provided suitable responses to alleviate the majority of the concern. While some of the findings are related to earlier work, the scale and details of the understanding from this study are sufficiently novel relative to prior works. It is also difficult to broadly compare fundamental properties in systems such as plants and

animals but I think in this case the authors are able to highlight how some of the findings in this study do not agree with the broad assumptions and findings in animal systems. I thought this was a useful comparison / contrast. No assay is ideal and each has limitations. The reviewer highlights some of the limitations for this assay. The authors responded to clarify and provide data about how they attempt to reduce the potential limitation of the assay and in some cases clarify in the text alternative interpretations. In my opinion, I felt that the authors satisfactorily addressed the key concerns of this reviewer or acknowledge limitations in interpretations in the revised manuscript.

The revised manuscript represents a subtle improvement over the original version and I do not have any remaining concerns or issues.

Reviewer #2:

Remarks to the Author:

The revised manuscript is significantly enhanced, addressing the issues raised by the reviewers and include significant additional experimentation. The manuscript would provide a significant contribution in the field, once published.

I have only a few comments left:

- 1) While it is true that a vast majority of Arabidopsis TSSs are known, there are still genes in which the TSS has not been experimentally determined (but show in the annotation). How might this affect the conclusions? One easy way to address is to indicate what evidence supports the determination of the TSS for key genes used to conclude the importance of the regulatory sequences downstream of the TSS. Also important to keep in mind that RNA pol II "stutters" and that for most genes the TSS does not represent a single position.
- 2) While not directly relevant, I am puzzled by the finding that TSS position does not vary between ecotypes given what has been found in maize when comparing inbred lines.
- 3) I don't understand Fig. S2A. What exactly is it showing? Wouldn't it be much easier to say distance from TSS to first intron? Given that coding sequences are all exonic, it seems that the figure is unnecessarily complicated. Same with Fig. S2F - the departure from 100% is just intron? And in Fig. S2G, this is all on genomic distance, not mRNA, correct?
- 4) In Fig. S3, the TES is determined how? is this the polyadenylation site or the true transcription end site? determined how?

Minor comments:
Authors in Ref 42 need to be revised

Author Rebuttal, first revision:

Reviewer #1:

Remarks to the Author:

I appreciated the authors careful and comprehensive response to the reviews. Many of my comments were driven by lack of clarity in understanding some aspects of the work. The revisions (and responses) were helpful in clarifying details of this complex study. I appreciate that the authors were willing to undertake additional analyses (comparative genomics of 5' UTR length, re-centering plots on the AUG, etc). These answered curiosities that I had and I hope the authors find them useful. It was important to provide more clear description of the key motif and the additions now provide that. I felt that my comments and suggestions were fully addressed.

The editor also requested an evaluation of the responses to reviewer 3. I can see the perspective of reviewer 3 but also understand the perspective of the authors. I appreciate the authors willingness to engage in a dialogue and to provide details to support their perspective. A key criticism of reviewer 3 was a lack of novelty, a more nuanced description of the findings – especially relative to animal systems and potential limitations of the key assays. In each case I thought that the authors provided suitable responses to alleviate the majority of the concern. While some of the findings are related to earlier work, the scale and details of the understanding from this study are sufficiently novel relative to prior works. It is also difficult to broadly compare fundamental properties in systems such as plants and animals but I think in this case the authors are able to highlight how some of the findings in this study do not agree with the broad assumptions and findings in animal systems. I thought this was a useful comparison / contrast. No assay is ideal and each has limitations. The reviewer highlights some of the limitations for this assay. The authors responded to clarify and provide data about how they attempt to reduce the potential limitation of the assay and in some cases clarify in the text alternative interpretations. In my opinion, I felt that the authors satisfactorily addressed the key concerns of this reviewer or acknowledge limitations in interpretations in the revised manuscript.

The revised manuscript represents a subtle improvement over the original version and I do not have any remaining concerns or issues.

Reviewer #2:

Remarks to the Author:

The revised manuscript is significantly enhanced, addressing the issues raised by the reviewers and include significant additional experimentation. The manuscript would provide a significant contribution in the field, once published.

I have only a few comments left:

1) While it is true that a vast majority of Arabidopsis TSSs are known, there are still genes in which the TSS has not been experimentally determined (but show in the annotation). How might this affect the conclusions? One easy way to address is to indicate what evidence supports the determination of the TSS for key genes used to conclude the importance of the regulatory sequences downstream of the TSS. Also

important to keep in mind that RNA pol II "stutters" and that for most genes the TSS does not represent a single position.

Our primary conclusions regarding the positional dependence of regulatory sequences, the importance of the region downstream of the TSS, and the role of the GATC motif are supported by the MPRA experimental systems. In the MPRA, sequences are positioned relative to a specific core promoter. Therefore, potential inaccuracies in the Arabidopsis TSS annotation should not undermine our results.

The genomic analysis, presented primarily in Fig. 1 and accompanying Supplementary Figures, would only be compromised by a systematic imprecision in the TSS annotations, as only averages of hundreds/thousands of genes are presented. This seems unlikely, as evidenced by the existing literature (e.g. Schon et al. 2018; Thieffry et al. 2020) as well as the genomic analysis of signals around the TSS, such as the DAP-Seq dip in Fig. 1C and the nucleotide composition around the TSS in Figure 3A-B in Hetzel et al. 2016. These data strongly suggest the reliability of the TSS definitions used.

Arabidopsis TSS annotations have been iteratively refined through genomic measurements (including RNA-Seq and proteomics) and manual curation (Swarbreck et al. 2008; Lamesch et al. 2012). Thus, it is challenging to identify evidence for individual gene annotations, but the aggregate data supports their accuracy.

The phenomenon of RNA polymerase II "stuttering" occurs on a smaller scale than the phenomena we report. For example, in Thieffry et al. Fig. 1E shows that the CAGE-Seq data, when using TAIR10 annotations, is focused on the TSS and mostly a few tens of base pairs around it, whereas the effects we report are observed at the scale of hundreds of base pairs. This difference in scale suggests that RNA polymerase II stuttering does not significantly affect our conclusions.

Following the reviewers' suggestions, we have included new analyses (Fig. S2C) demonstrating experimental support for the TSS annotations of key gene groups - those with cis-eQTLs and those used in our oligonucleotide pool design for the MPRA. In the re-analysis of the CAGE-Seq data from Thieffry et al., 99.6% and 99.1% of the genes in these groups, respectively, have a CAGE-Seq signal within 500 bp of the annotated TSS on the corresponding strand. The average CAGE-Seq data around the TSS of these genes is presented in the newly added Fig. S2C, which illustrates a sharp peak at the defined TSS in the TAIR10 annotation.

Finally, the genes selected for oligonucleotide pool design were filtered for ones with a unique TSS in the TAIR10 annotation, as described in the Methods section.

2) While not directly relevant, I am puzzled by the finding that TSS position does not vary between ecotypes given what has been found in maize when comparing inbred lines.

We were also surprised by the small variation in TSS positions between ecotypes and note that these results are preliminary. Regarding maize inbred lines, Mejía-Guerra et al. 2015 found that 757 and 347 genes have a shifted TSS between maize ecotypes B73 and Mo17 in roots and shoots, respectively. This is <5% of the 17,409 genes with identified TSS in the study. Therefore, even in maize, a large proportion of genes have a similar TSS position. In any case, further analysis is certainly needed, but we note that the conclusions of this paper do not depend on it.

3) I don't understand Fig. S2A. What exactly is it showing? Wouldn't it be much easier to say distance from TSS to first intron? Given that coding sequences are all exonic, it seems that the figure is unnecessarily complicated. Same with Fig. S2F - the departure from 100% is just intron? And in Fig. S2G, this is all on genomic distance, not mRNA, correct?

We appreciate the reviewer's insights and have revised the figure captions for clarity. Unfortunately, due to space limitations, we were unable to fully discuss these control graphs in the main text.

Fig. S2A: This figure shows the exon distribution specifically within the 500 bp downstream of the TSS. This region is relevant because this is where we show eQTL enrichment (e.g. Fig. 1A-B). In addition, this figure illustrates the distribution of gene cluster sizes used in Fig. S2D.

Fig. S2G (formerly S2F): This figure illustrates the proportion of the genomic distance from the TSS to the ATG that is exonic, essentially representing the total exon length of the 5' UTR. Since introns can be present within the 5' UTR, this metric differs from simply measuring the distance to the first intron. The figure supports the analyses in Fig. 1B and S2H, where we plot eQTLs for different gene groups categorized by TSS-to-ATG distance. Although not essential for our claim, it gives a sense of the genomic composition of the TSS-to-ATG region, which is central to our analysis.

Regarding Fig. S2H (formerly S2G): This figure plots genomic distances relative to the ATG, not the TSS, and was added in response to a suggestion from reviewer #1 during the revision of the paper.

4) In Fig. S3, the TES is determined how? is this the polyadenylation site or the true transcription end site? determined how?

The eQTLs in Fig. S3 are aligned around the ends of the transcripts. The definition is based on the TAIR10 annotation of the transcript end. This definition, derived from mRNA sequencing, represents the polyadenylation site. We have clarified this in the figure caption and written that it is the transcript 3' end. Thank you for bringing this to our attention.

Minor comments:

Authors in Ref 42 need to be revised

Corrected. Thank you for noticing this!

References:

- Hetzl, Jonathan, Sascha H. Duttke, Christopher Benner, and Joanne Chory. 2016. "Nascent RNA Sequencing Reveals Distinct Features in Plant Transcription." *Proceedings of the National Academy of Sciences of the United States of America* 113 (43): 12316–21.
- Lamesch, Philippe, Tanya Z. Berardini, Donghui Li, David Swarbreck, Christopher Wilks, Rajkumar Sasidharan, Robert Muller, et al. 2012. "The Arabidopsis Information Resource (TAIR): Improved Gene Annotation and New Tools." *Nucleic Acids Research* 40 (Database issue): D1202–10.
- Mejía-Guerra, María Katherine, Wei Li, Narmer F. Galeano, Mabel Vidal, John Gray, Andrea I. Doseff, and Erich Grotewold. 2015. "Core Promoter Plasticity Between Maize Tissues and Genotypes Contrasts with Predominance of Sharp Transcription Initiation Sites." *The Plant Cell* 27 (12): 3309–20.
- Schon, Michael A., Max J. Kellner, Alexandra Plotnikova, Falko Hofmann, and Michael D. Nodine. 2018. "NanoPARE: Parallel Analysis of RNA 5' Ends from Low-Input RNA." *Genome Research* 28 (12): 1931–42.
- Swarbreck, David, Christopher Wilks, Philippe Lamesch, Tanya Z. Berardini, Margarita Garcia-Hernandez, Hartmut Foerster, Donghui Li, et al. 2008. "The Arabidopsis Information Resource (TAIR): Gene Structure and Function Annotation." *Nucleic Acids Research* 36 (Database issue): D1009–14.
- Thieffry, Axel, Maria Louisa Vigh, Jette Bornholdt, Maxim Ivanov, Peter Brodersen, and Albin Sandelin. 2020. "Characterization of Arabidopsis Thaliana Promoter Bidirectionality and Antisense RNAs by Inactivation of Nuclear RNA Decay Pathways." *The Plant Cell* 32 (6): 1845–67.

Decision Letter, second revision:

Message: Our ref: NG-LE63565R3

10th Jul 2024

Dear Yoav,

Thank you for submitting your revised manuscript "Widespread position-dependent transcriptional regulatory sequences in plants" (NG-LE63565R3).

We have made an editorial check of the changes made in response to the last round of reviews, and we are satisfied with them such that we think no further review is required. Therefore, we'll be happy in principle to publish it in Nature Genetics, pending minor revisions to satisfy the referees' final requests and to comply with our editorial and formatting guidelines.

Sincerely,

Michael Fletcher, PhD
Senior Editor, Nature Genetics
ORCID: 0000-0003-1589-7087

Final Decision Letter

13th Aug 2024

Dear Yoav,

I am delighted to say that your manuscript "Widespread position-dependent transcriptional regulatory sequences in plants" has been accepted for publication in an upcoming issue of *Nature Genetics*.

Over the next few weeks, your paper will be copyedited to ensure that it conforms to *Nature Genetics* style. Once your paper is typeset, you will receive an email with a link to choose the appropriate publishing options for your paper and our Author Services team will be in touch regarding any additional information that may be required.

Your paper will be published online after we receive your corrections and will appear in print in the next available issue. You can find out your date of online publication by contacting the Nature Press Office (press@nature.com) after sending your e-proof corrections.

Before your paper is published online, we shall be distributing a press release to news organizations worldwide, which may very well include details of your work. We are happy for your institution or funding agency to prepare its own press release, but it must mention the embargo date and *Nature Genetics*. Our Press Office may contact you closer to the time of publication, but if you or your Press Office have any enquiries in the meantime, please contact press@nature.com.

Please note that *Nature Genetics* is a Transformative Journal (TJ). Authors may publish their research with us through the traditional subscription access route or make their paper immediately open access through payment of an article-processing charge (APC). Authors will not be required to make a final decision about access to their article until it has been accepted. Find out more about Transformative Journals

Authors may need to take specific actions to achieve compliance with funder and institutional open access mandates. If your research is supported by a funder that requires

immediate open access (e.g. according to Plan S principles) then you should select the gold OA route, and we will direct you to the compliant route where possible. For authors selecting the subscription publication route, the journal's standard licensing terms will need to be accepted, including <https://www.nature.com/nature-portfolio/editorial-policies/self-archiving-and-license-to-publish>. Those licensing terms will supersede any other terms that the author or any third party may assert apply to any version of the manuscript.

If you have not already done so, we strongly recommend that you upload the step-by-step protocols used in this manuscript to [protocols.io](https://www.protocols.io). [protocols.io](https://www.protocols.io) is an open online resource that allows researchers to share their detailed experimental know-how. All uploaded protocols are made freely available and are assigned DOIs for ease of citation. Protocols can be linked to any publications in which they are used and will be linked to from your article. You can also establish a dedicated workspace to collect all your lab Protocols. By uploading your Protocols to [protocols.io](https://www.protocols.io), you are enabling researchers to more readily reproduce or adapt the methodology you use, as well as increasing the visibility of your protocols and papers. Upload your Protocols at <https://www.protocols.io>. Further information can be found at <https://www.protocols.io/help/publish-articles>.

Sincerely,

Michael Fletcher, PhD
Senior Editor, Nature Genetics
ORCID: 0000-0003-1589-7087